



# Variable tree rooting strategies improve tropical productivity and evapotranspiration in a dynamic global vegetation model

Boris Sakschewski[1], Werner von Bloh[1], Markus Drüke[1], Anna A. Sörensson[2, 3], Romina Ruscica[2, 3], Fanny Langerwisch[4, 5], Maik Billing[1], Sarah Bereswill[6], Marina Hirota[7, 8], Rafael S. Oliveira[8], Jens Heinke[1], Kirsten Thonicke[1]

[1]Potsdam Institute for Climate Impact Research, Potsdam, 14473, Germany
[2]Universidad de Buenos Aires - Consejo Nacional de Investigaciones Científicas y Técnicas, Centro de Investigaciones del Mar y la Atmósfera (CIMA/UBA-CONICET), Buenos Aires, Argentina.
[3]Institut Franco-Argentin d'Etudes sur le Climat et ses Impacts, Unité Mixte Internationale (UMI-IFAECI/CNRS-CONICET-UBA), Argentina
[4]Czech University of Life Sciences Prague, Department of Water Resources and Environmental Modeling, 165 00 Praha 6 – Suchdol, Czech Republic
[5]Palacký University Olomouc, Department of Ecology and Environmental Sciences, 78371 Olomouc, Czech Republic
[6]University of Potsdam, Potsdam, 14469, Germany
[7]Federal University of Santa Catarina (UFSC), Campus Universitário Reitor João David Ferreira Lima Trindade – Florianópolis – SC, CEP: 88040-900, Santa Catarina, Brazil
[8]University of Campinas (UNICAMP) Cidade Universitária "Zeferino Vaz" CEP 13083-970, Campinas-SP, Sao Paulo, Brazil

*Correspondence to*: Boris Sakschewski (boris.sakschewski@pik-potsdam.de)

**Abstract.** Tree water access via roots is crucial for forest functioning and therefore forests have developed a vast variety of rooting strategies across the globe. However, Dynamic Global Vegetation Models (DGVMs), which are increasingly used to simulate forest functioning, often condense this variety of tree rooting strategies into biome-scale averages, potentially under- or overestimating forest response to intra- and inter-annual variability in precipitation. Here we present a new approach of implementing variable rooting strategies and dynamic root growth into the LPJmL4.0 DGVM and apply it to tropical and sub-tropical South-America under contemporary climate conditions. We show how competing rooting strategies which underlie the trade-off between above- and below-ground carbon investment lead to more realistic simulated intra-annual productivity and evapotranspiration, and consequently forest cover and spatial biomass distribution. We find that climate and soil depth determine a spatially heterogeneous pattern of mean rooting depth and belowground biomass across the study region.

## 1 Introduction

Tropical evergreen forest is the naturally dominant biome type in South-America over a large climatic range including regions with a marked dry season (Hirota et al., 2011; Xiao et al., 2006). To withstand seasonal shortages of precipitation and sustain productivity, trees with evergreen phenology often gain access to deep soil water via deep roots (Brum et al., 2019; Canadell et al., 1996; Johnson et al., 2018; Kim et al., 2012; Markewitz et al., 2010). Consequently, recent studies suggest a heterogeneous spatial pattern of maximum rooting depth across tropical forest biomes in South-America which differs over the order of magnitudes depending on local groundwater, soil and climate conditions (Canadell et al., 1996; Fan et al., 2017). Therefore, tree rooting depth is regarded as a crucial variable to explain the geographical distribution of main phenology strategies such as "evergreen" and "deciduous", as well as the observed local to continental pattern of productivity, biomass storage, evapotranspiration (ET) and consequently moisture recycling (Fan et al., 2017; Jobbágy and Jackson, 2000; Kleidon and Heimann, 2000; Langan et al., 2017; Nepstad et al., 1994; Stahl et al., 2013). While these variables and processes are in the focus of model-based earth system sciences projecting the development of vegetation formation and ecosystem functioning worldwide (Huntingford et al., 2013; Liu et al., 2018; Weber et al., 2009), most DGVMs and land-surface models (LSMs) still do not represent the diversity of rooting depth or tree rooting strategies (Warren et al., 2015a). In general these models condense the diversity of such functional plant traits to biome-scale averages,





to simulate so called plant functional types (PFTs) which reflect average plant individuals of a biome. Here, often a shallow
rooting depth for tree PFTs is assumed, i.e. most roots are distributed downwards to a few meters of depths at maximum
(Arora and Boer, 2003; Best et al., 2011; Guimberteau et al., 2017; Lawrence et al., 2011; Ostle et al., 2009; Schaphoff et al.,
2018; Smith et al., 2014). By ignoring natural local adaptations of rooting depth, DGVMs and LSMs in the past had
problems reproducing the extent of South-America's tropical evergreen forests, as well as its seasonal productivity and ET
especially in regions with seasonal rainfall (Baudena et al., 2014; Liu et al., 2018, 2017; Restrepo-Coupe et al., 2017).
So far different attempts were carried out trying to solve this problem in DGVMs and similar models by allowing for
variable rooting strategies. More than 20 years ago a pioneering study by Kleidon and Heimann (1998) systematically
searched for rooting strategies which yield highest net primary productivity over South America in a DGVM to explain intra-
annual rates of ET and vegetation cover. Follow up studies further underlined the importance of deep roots for the climate
system of South America (Kleidon and Heimann, 2000). Lee *et al.* (2005) found that allowing for deep roots and hydraulic
redistribution of water in the soil column in a general circulation model enhances Amazon forest productivity and
evapotranspiration (ET) in the dry season. Baker *et al.* (2008) came to similar results when introducing deep roots in a land
surface model. Ichii *et al.*, (2007) found that constraining rooting depth across the Amazon based on satellite data yields
similar results in a terrestrial ecosystem model. More recently, Langan, Higgins and Scheiter (2017) showed for the same
study area how diverse rooting strategies in a tree individual and trait-based DGVM can improve simulated intra-annual
productivity and ET and better explain patterns of different tropical biome types and biomass in connection with fire.
While these studies are important steps to acknowledge the diversity of tree rooting strategies and its effects, some
assumptions of the underlying models might decrease the liability of their results. These assumptions are related to 1)
resource investment, 2) timing and 3) physical constraints of rooting depth. 1) Most models so far do not account for coarse
roots (Warren et al., 2015a) even though they can make up the majority of total root biomass (Xiao et al., 2003). This
approach may be sufficient when employing shallow tree rooting strategies only, but with increasing rooting depth costs for
coarse roots increases substantially. Since the amount of resources trees can allocate to their processes and structures is
finite, a local adaptation of tree rooting depth must follow a trade-off between above- and below-ground resource investment
(Nikolova et al., 2011). Generally above-ground investments into leaf and stem growth can increase light absorption and
$CO_2$ uptake, while below-ground investments can increase the uptake of water and nutrients. Depending on local
environmental and competitive conditions one or the other direction might be more advantageous, eventually leading to
substantial regional variation in the mean below-ground to above-ground biomass ratios (Leuschner et al., 2007; Mokany et
al., 2006). Therefore, the simulated spectrum of tree rooting strategies which can survive and co-exist should be in
accordance with this crucial trade-off. 2) In contrast to above-ground stem growth, most DGVMs so far do not simulate
gradual root growth (Warren et al., 2015a). Instead PFTs are assigned a constant relative distribution of fine roots throughout
the soil column at any point in space and time (Best et al., 2011; Lawrence et al., 2011; Schaphoff et al., 2018; Smith et al.,
2014). As under the above mentioned simplification under 1), this approach may be sufficient when accounting for shallow
rooting strategies only, but when the maximum tree rooting depth of PFTs strongly diverges, it is questionable that the time
needed to reach this depth is negligible, especially when accounting for PFT competition. Rooting depth increases rather
gradually and non-linear over a tree's lifetime with a velocity driven by a mix of plastic optimization and allometric
determination (Brum et al., 2019; Brunner et al., 2015; Nikolova et al., 2011; Poorter et al., 2012; Warren et al., 2015b).
While the knowledge base for a mechanistic bottom-up modelling approach of plastic optimization is very sparse (Jenik,
2010; Poorter et al., 2012; Warren et al., 2015b), knowledge on certain allometric rules (Brum et al., 2019; Eshel and
Grünzweig, 2013; Mokany et al., 2006) seems enough to be applied in DGVMs. 3) Most DGVMs so far do not account for a
location dependent soil depth, but apply a constant soil depth across the globe (Best et al., 2011; Guimberteau et al., 2017;
Lawrence et al., 2011; Ostle et al., 2009; Schaphoff et al., 2018; Smith et al., 2014). Again this approach may be sufficient
when accounting for shallow rooting strategies only, but allowing for deep tree rooting strategies should go in parallel with





their potential physical barriers. Recent data products on global soil depth now enable to better constrain rooting depth in
DGVMs (Pelletier et al., 2016).
Here we overcome the above mentioned limitations and present a new approach of diversifying tree rooting strategies in the
DGVM LPJmL4.0 (Lund-Potsdam-Jena managed Lands; Schaphoff *et al.*, 2018) which increases the ecological liability
with the following aspects: 1) Maximum rooting depth is restricted to a recent global product of soil depth, 2) simulated tree
rooting strategies were chosen to represent a wide range of maximum rooting depth between 0.5 and 18 m, 3) this spectrum
of tree rooting strategies grows in competition and tree performance determines dominance, 4) dominance is supported by
best performing rooting strategies producing more offspring, 5) trees have to invest into coarse roots as well, i.e.
acknowledging the trade-off between growing deeper roots and allocating available carbon to other compartments (stem and
leaf growth), and 6) tree roots are growing deeper over time depending on tree height. The objectives of this study are to
describe an approach of how to diversify tree rooting strategies in a DGVM and to evaluate its effect on simulated
evapotranspiration, productivity, biomass and spatial distribution of evergreen and deciduous tropical forests using different
sources of validation data.

## 2 Materials and Methods

In the method sections below we describe the implementation of a new tree rooting scheme in LPJmL4.0 (Sect. 2.2) where
maximum rooting depth is constrained by a recent map on maximum soil/sediment thickness (Sect. 2.3). We apply the model
to several historical climate input data (Sect. 2.7) with details of the simulation protocol described in Sect. 2.9. The
Evaluation of the new model version is described in Sect. 2.10.
All data processing and statistical analysis described in the methods sections were performed with the commercial software
MATLAB® (MATLAB and Statistics Toolbox Release 2012b; The MathWorks, Inc., Natick, MA, USA).

### 2.1 The LPJmL4.0 model

LPJmL4.0 is a process-based Dynamic Global Vegetation Model (DGVM) which simulates the surface energy balance,
water fluxes, carbon fluxes and stocks of the global land (Schaphoff et al., 2018). Plant productivity is modelled on the basis
of leaf-level photosynthesis responding to climatic and environmental conditions, atmospheric $CO_2$ concentration, canopy
conductance, autotrophic respiration, phenology and management intensity. The model employs 11 plant functional types
(PFTs), 3 bioenergy types (BFTs) and 12 crop functional types (CFTs), to represent average plants of biomes, bioenergy
plantations and agriculture, respectively. The standard spatial resolution is a 0.5° x 0.5° grid. For each grid cell the fractional
coverage of bioenergy and agricultural BFTs and CFTs follows a prescribed land-use data set, whereas in the remaining area
natural PFTs grow in competition.

### 2.2 A new tree rooting scheme for LPJmL4.0

In this section we describe the new basic scheme for soil layer partitioning, the new tree rooting scheme, the simulation of
belowground carbon investment, and how different tree rooting schemes compete. All changes made to LPJmL4.0 described
in the methods below result in a new sub-version of LPJmL4.0 we call LPJmL4.0-VR hereafter (where "VR" stands for
variable roots).

### 2.2.1 Scheme for soil layer partitioning

LPJmL4.0 employs a globally universal soil depth of 3 m. For LPJmL4.0-VR we extended the general maximum soil depth
to 20 m (but restrict it to local soil depth information at spatial model resolution; Sect. 2.3.2). We applied the same basic
scheme for soil layer partitioning from LPJmL4.0 (Schaphoff et al., 2018), but continue this scheme down to 20 m (Tab. 1).





We chose a maximum of 20 m soil depth to considerably increase the maximum soil depth compared to constant 3 m in
LPJmL4.0, while keeping the increment of computational intensiveness connected to adding more soil layers within an
acceptable range. As for LPJmL4.0, a general soil texture information is applied to the whole soil column (Schaphoff et al.,

132   2018).

**2.2.2 Diversifying general tree rooting strategies in LPJmL4.0-VR**

In LPJmL4.0 the tree rooting strategy of a PFT is reflected by a certain prescribed vertical distribution of fine roots
throughout the soil column. Each soil layer $l$ is assigned a PFT specific relative amount of fine roots $rootdist_l$:
$rootdist_l = rootdist(z_l) - rootdist(z_{l-1})$                                                                 Eq. (1)
where $z_l$ is the soil layer boundary depth in cm of each soil layer $l$ and $rootdist(z_l)$ is the relative amount of fine roots between
the forest floor and the boundary of soil layer $l$. The function $rootdist(z)$ is defined following Jackson *et al.* (1996):
$roodist(z) = \frac{1-\beta^z}{1-\beta^{z_{bottom}}}$                                                             Eq. (2)
where $\beta$ is a constant parameter shaping the vertical distribution of fine roots and therefore determining the tree rooting
strategy and $z_{bottom}$ is the maximum soil depth in cm. In LPJmL4.0 each PFT is assigned a different $\beta$-value reflecting the
average tree rooting strategy on this broad biome scale (Schaphoff et al., 2018).
To quantify the maximum rooting depth of PFTs that actually results from this approach (Eq. 1&2) we here calculate the
depth which is reached by 95% of the fine root biomass ($D_{95\_max}$) as follows:
$D_{95\_max} = \frac{log(1-0.95\cdot(1-\beta^{z_{bottom}}))}{log(\beta)}$                                        Eq. (3)
In LPJmL4.0 the $\beta$-values of tropical tree PFTs are set to 0.962 for the evergreen PFT and 0.961 for the deciduous PFT.
According to Eq. 3 both PFTs have a $D_{95\_max}$ smaller than 1 m. For LPJmL4.0-VR we extended this representation of tree
rooting strategies by splitting both tropical tree PFTs into 10 sub-PFTs and assigned each with a different $\beta$-value. These
values were chosen to cover a range of different $D_{95\_max}$ values between 0.5 and 18m (Tab. 2). Fig. 1 shows the new
maximum distribution of fine roots throughout the soil column according to the chosen $\beta$-values (Tab. 2).

**2.2.3 Belowground carbon investment**

Tropical trees can avoid water stress under seasonally dry climate by growing relatively deep roots (Brum et al., 2019; Fan et
al., 2017) which goes along with increased below-ground carbon investment. Thus, the need for deep water access creates a
trade-off between below-ground and above-ground carbon investment. Therefore, a new tree rooting scheme for LPJmL4.0-
VR was necessary to account for this trade-off in order to reproduce observed local to regional patterns and distributions of
tree rooting strategies instead of prescribing them. Therefore, we introduced two new carbon pools in LPJmL4.0-VR, namely
root sapwood and root heartwood. Like stem sapwood in LPJmL4.0, also root sapwood in LPJmL4.0-VR needs to satisfy the
assumptions of the pipe model (Shinozaki et al., 1964; Waring et al., 1982). The pipe model describes, that for a certain
amount of leaf area a certain amount of water conducting tissue must be available. In LPJmL4.0 the cross-sectional area of
stem sapwood needs to be proportional to the leaf area $LA_{ind}$ as follows:
$LA_{ind} = k_{la:sa} \cdot SA_{ind}$                                                                           Eq. (4)
where $k_{la:sa}$ is a constant describing the ratio of leaf area and stem sapwood cross-sectional area ($SA_{ind}$). In LPJmL4.0-VR we
also apply the pipe model to root sapwood. Root sapwood cross-sectional area must be proportional to stem sapwood cross-
sectional area, but is decreasing with soil depth, depending on the relative amount of fine roots in each soil layer (Fig. 2).
Root sapwood is turned into root heartwood at an equal rate as stem sapwood is turned into stem heartwood, i.e. 5% per year
as in LPJmL4.0 (see Schaphoff *et al.*, 2018).



**2.2.4 Root growth**
In LPJmL4.0 (Schaphoff et al., 2018) no vertical root growth is simulated, thus the relative distribution of fine roots over the
soil column is constant over space and time. It means that PFTs starting from bare ground in a sapling stage display the same
relative distribution of fine roots throughout the soil column as a full-grown forest which contradicts the principles of
dynamic root growth over a tree's lifetime. Applied to LPJmL4.0-VR, the belowground biomass of an initialized deep
rooting-strategy sub-PFT would exceed its aboveground biomass (AGB) by order of magnitudes when acknowledging
coarse roots. Consequently, deep rooting strategies would always be disadvantageous, calling for modelling gradual root
growth in LPJmL4.0-VR. Unfortunately, little is known about how roots of tropical trees grow over time, given the fact that
this research field is strongly time and resource demanding, and at the same time the variety of tree species, rooting
strategies and environmental conditions are large (Jenik, 2010). A recent promising study by Brum *et al.* (2019) was able to
capture the effective functional rooting depth (EFRD) of different size classes of 12 dominant tree species in a seasonal
Amazon forest where tree roots grow considerably deep with maximum values reaching below 30m. To our knowledge this
is the only study capturing the relation between the size of tropical trees and their maximum rooting depth in a high spatial
resolution covering sufficient tree-height classes in order to derive a function. Following the findings of Brum *et al.* (2019),
we here implemented a logistic root growth function, which calculates a general maximum conceivable tree rooting depth $D$
depending on tree height:
$D = \frac{S}{e^{-kSh}} \cdot \left( \frac{S}{D_0} - 1 \right)$                                             Eq. (5)
where $S$ is the maximum soil depth in the model (20 m), $k$ is the growth rate (set to 0.02), $h$ is the tree height in m and $D_0$ is
the initial rooting depth of tree saplings (set to 0.1 m; tree saplings in LPJmL4.0-VR are initialized with a height of 0.45 m as
in LPJmL4.0). The distribution of fine root biomass of each sub-PFT in the soil column is then adjusted according to $D$ each
time step, by restricting $z_{bottom}$ in Eq. 2. Every time $D$ crosses a specific soil layer boundary (Sect. 2.2.1) $z_{bottom}$ is assigned the
value of the next soil layer boundary. Thus, $z_{bottom}$ increases in discrete steps. Consequently, each tree rooting strategy
allowed for in this study (2.2.2) shows a logistic growth of rooting depth dependent on tree height which saturates towards
its specific maximum rooting depth (Fig. 3). Therefore, limitations of aboveground tree growth due to below-ground carbon
investment of different tree rooting strategies (Sect. 2.2.3) are equal in the tree sapling phase of all sub-PFTs) and start to
diverge with increasing tree height. In the case $D$ exceeds the grid cell specific local soil depth (as prescribed by the soil
thickness input, see Sect. 2.3.2) all the respective fine root biomass exceeding this grid cell specific soil depth is transferred
to the last soil layer which matches this soil depth (see also Fig. 2 right panel and Supplementary Video 1 for a visualization
of root growth under http://www.pik-potsdam.de/~borissa/LPJmL4_VR/Supplementary_Video_1.pptx).
The parameter $k$ in Eq. 5 was chosen to preserve the slope of the 75%ile function describing the relation between tree height
and effective functional rooting depth (EFRD) as found in Brum *et al.* (2019). We could not implement any of the original
functions as suggested in Brum *et al.* (2019) since they deliver unrealistic low values of rooting depth (between 0 and 10cm)
for trees <= 10 m, which results in a strong competitive disadvantage against herbaceous PFTs in LPJmL4.0-VR. We
decided for the slope of the 75%ile function since we wanted to apply root growth rates close to the maximum which also
allows for the largest $D_{95\_max}$ values in this study (Sect. 2.2.1) to be reached.
Note that Brum *et al.* (2019) originally propose a relation between tree diameter at breast height (*DBH*) and EFRD. For our
purposes we related rooting depth to tree height (h), which is calculated from DBH in in LPJmL4.0 according to (Huang et
al., 1992):
$h = k_{allom2} \cdot DBH^{k_{allom3}}$                                             Eq. (6)
where $k_{allom2}$ and $k_{allom3}$ are constants set to 40 and 0.67, respectively (Schaphoff et al., 2018).



### 2.2.5 Competition of rooting strategies

In each grid-cell all sub-PFTs of the evergreen and deciduous tree PFTs compete for light and water following LPJmL4.0's approach to simulate plant competition. To allow for environmental filtering of tree rooting strategies which are best adapted to local environmental conditions, we changed the tree establishment scheme of LPJmL4.0-VR. In LPJmL4.0, the number of new PFT saplings per unit area ($est_{PFT}$ in ind m$^{-2}$ a$^{-1}$) which are established each year is proportional to a maximum establishment rate $k_{est}$ and to the sum of foliage projected cover (FPC; a relative number between 0 and 1) of all tree PFTs present in a grid cell ($FPC_{TREE}$). It declines in proportion to canopy light attenuation when the sum of woody FPCs exceeds 0.95, thus simulating a decline in establishment success with canopy closure (Prentice et al., 1993):

$$est_{PFT} = k_{est} \cdot (1 - e^{(-5 \cdot (1-FPC_{TREE})}) \cdot \frac{1 - FPC_{TREE}}{n_{est_{TREE}}} \qquad \text{Eq. (7)}$$

where $n_{est_{TREE}}$ is the number of established tree individuals per m² per year. In LPJmL4.0-VR, establishment rates of sub-PFTs ($est_{sub\_PFT}$) are additionally weighted by local dominance of each sub-PFT as follows:

$$est_{sub\_PFT} = k_{est} \cdot \left(1 - e^{-5 \cdot (1-FPC_{TREE})}\right) \cdot \frac{1 - FPC_{TREE}}{n_{est_{TREE}}} \cdot \frac{FPC_{sub\_PFT}}{FPC_{TREE}} \cdot n_{est_{TREE}} \qquad \text{Eq. (8)}$$

where $FPC_{sub\_PFT}$ is the FPC of each sub-PFT. The new term allows productive sub-PFTs to establish more offspring relative to their spatial dominance and vice versa, without changing the overall establishment rate as set by (Prentice et al., 1993). This function has the effect that non-viable sub-PFTs are outcompeted over time.

### 2.2.5 Background mortality

In LPJmL4.0 background mortality is modelled by a fractional reduction of biomass, which depends on growth efficiency (Schaphoff et al., 2018). This annual rate of mortality is limited by a constant maximum mortality rate of 3% of tree individuals per year which is applied to all tree PFTs. In other words the fastest total biomass loss of a tree PFT due to low growth efficiency can happen within $1/0.03 \sim 33$ simulation years. In general, this maximum mortality rate can be regarded as a global tuning parameter of biomass accumulation as it caps the maximum biomass loss. Since many mechanisms influencing tree mortality in the real world, e.g. hydraulic failure (Johnson et al., 2018), are not yet implemented in most DGVMs including LPJmL4.0 (Allen et al., 2015), the parameterization of a background tree mortality remains a challenging topic. Under the current model status of LPJmL4.0 maximum mortality rates are a necessary feature, while future model development must overcome the concept of applying a maximum mortality rate by refining and implementing most important mechanisms that influence tree mortality.

The new features of LPJmL4.0-VR head in this direction. Here tree PFTs can access water in soil depths which were formally inaccessible. This enhances the general growth efficiencies of tree PFTs and consequently decreases their overall background mortality. Since global biomass pattern simulated with LPJmL4.0 are already in an acceptable range, we increased the maximum background mortality in LPJmL4.0-VR to 7% in order to counter-balance increased survival rates and therefore biomass accumulation. This value keeps simulated mortality rates in real world boundaries, as a recent study comprising data of 167 forest plots finds that actual annual stem mortality rates generally do not exceed 6% across Amazonia (Johnson et al., 2016). We regard increasing the maximum mortality rate as a step into the right direction as its value can eventually be set close to 100% when model development progresses.

### 2.3 Model input data

### 2.3.1 Climate and land-use input data

All versions of LPJmL used in this study (see Sect. 2.4) were forced with 4 different climate inputs each based on single or multiple available data products delivering the climate variables air temperature, precipitation, long-wave and shortwave downward radiation at daily or monthly resolution:





1) WATCH Forcing Data (WFD) + WATCH Forcing Data methodology applied to ERAInterim data. A combination of the
WATCH data set (Weedon et al., 2011) and the WFDEI data set (Weedon et al., 2014) as used in the ISIMIP project
(https://www.isimip.org/gettingstarted/input-data-bias-correction/details/5/). This input data set is called WATCH+WFDEI
hereafter.
2) Global Soil Wetness Project Phase 3 (GSWP3) (Kim *et al.*, no date; http://hydro.iis.u-tokyo.ac.jp/GSWP3/index.html).
3) NOAH Global Land Assimilation System version 2.0 (GLDAS, Rodell *et al.*, 2004).
4) Climate forcing as in Schaphoff *et al.* (2018) with monthly precipitation provided by the Global Precipitation Climatology
Centre (GPCC Full Data Reanalysis version 7.0; (Becker et al., 2013), daily mean temperature from the Climate Research
Unit (CRU TS version 3.23, University of East Anglia Climatic Research Unit, 2015; Harris *et al.*, 2014), shortwave
downward radiation and net downward radiation reanalysis data from ERA-Interim (Dee et al., 2011), and number of wet
days from (New et al., 2000) used to allocate monthly precipitation to individual days.
This input data set is called CRU hereafter.

### 2.3.2 Soil and sediment thickness

We regridded a global 1 x 1 km soil and sediment thickness product (Pelletier et al., 2016) to the 0.5° x 0.5° spatial
resolution of LPJmL4.0-VR, set the global maximum value to 20 m according to the maximum soil depth chosen for
LPJmL4.0-VR (Sect. 2.2.1), and used the resulting map as grid cell specific model input (Fig. 4).

### 2.4 Model versions and simulation protocol

In order to investigate the impact of simulating variable rooting strategies and root growth, we employ 3 model versions of
LPJmL in this study: 1) LPJmL4.0, 2) LPJmL4.0-VR, and 3) LPJmL4.0-VR-base with the same settings as LPJmL4.0-VR
but without variable rooting strategies, i.e. using the $\beta$-values of LPJmL4.0 for the tropical evergreen PFT ($\beta = 0.962$) and
the tropical deciduous PFT ($\beta=0.961$) for all the respective 10 sub-PFTs. We regard the latter model version as a baseline
model of this study, because comparisons to LPJmL4.0-VR enable to investigate differences which are caused by the amount
of considered tree rooting strategies only.
Each simulation was initialized with 5000 simulation years of spin up from bare ground without land-use by randomly
recycling the first 30 years of respective climate data (1901-1930 for WATCH+WFDEI, GSWP3, CRU and 1948-1977 for
GLDAS) and a pre-industrial atmospheric $CO_2$ level of 278ppm, in order to ensure that carbon pools and local distribution of
PFTs and sub-PFTs are in equilibrium with climate. This first spin-up phase was followed by another spin up phase of 390
years using the same climate data, but employing historical land-use data (reshuffling the first 30 years 1851-1880). Land-
use input and routines were carried out according to the standard settings of LPJmL 4.0 as described in (Schaphoff et al.,
2018). This second spin-up phase was followed by transient simulations (1901-2010 for WATCH+WFDEI, GSWP3, CRU
and 1948-2010 for GLDAS) with respective land-use change and changing levels of atmospheric $CO_2$ concentration.
At the beginning of the spin-up phase, all sub-PFTs in LPJmL4.0-VR and LPJmL4.0-VR-base have the same chance to
establish, i.e. tree rooting strategies are uniformly distributed. During the spin-up simulation local environmental conditions
lead to environmental filtering supported by competition and PFT-dominance dependent establishment rates (Sect. 2.2.4).
Therefore, the following transient simulations already start with distinct distributions of tree rooting strategies.

### 2.5 Model validation

### 2.5.1 Validation data

*Regional biomass pattern*



For evaluation of simulated regional pattern of AGB we compare the results of all LPJmL model versions used in this study
to two remote sensing based biomass maps (Avitabile et al., 2016a; Saatchi et al., 2011) regridded to the spatial resolution of
the LPJmL models.
*Inventory-based biomass*
Because of the contradicting spatial pattern of currently available AGB maps, we also perform a direct comparison of our
modelled AGB patterns to inventory-based biomass estimates provided by (Brienen et al., 2015). The general problem of
such a comparison is that AGB estimates from DGVMs represent large-scale (0.5 x 0.5 degree) averages, while inventory-
based AGB estimates are representative for forest plots of a typical size of ~1 ha. Because of the smaller spatial scale, plot
estimates are affected by spatial variability and random measurement errors (Chave et al., 2004), which causes plot estimates
to differ from large-scale average AGB. Thus, even a simulated AGB pattern that perfectly matches the real large-scale
pattern would not yield a correlation coefficient of one when compared to small-scale plot observations. To address this
problem, we apply the method from Rammig *et al.* (2018), which was specifically developed to compare spatial patterns of
simulated large-scale ecosystem properties ($Y$) to ground-based observations ($X$). The method assumes that a small scale
"point" measurement consists of two components: the large-scale average and a normally-distributed random component
originating from small-scale variability and measurement error. The standard deviation of the random component can be
estimated from the data by analyzing differences among neighboring observation point, and then be used to obtain an
estimate of the standard deviation of the underlying large-scale AGB pattern $\sigma_{x,LS}$ and to calculate a modified correlation
coefficient $r_{LS}$ that accounts for differences in the large-scale patterns by removing the diminishing effect of the random
component in point observations. The subscript $LS$ for $\sigma_{x,LS}$ and $r_{LS}$ indicates that they represent estimates of the true large-
scale variability and the true correlation coefficient of the large-scale patterns. The uncertainty ranges for these two
properties as well as for the pattern average $\underline{x}$ (which does not require a correction and therefore no differentiation of 'large-
scale') are estimated by bootstrapping. For further details on the underlying methodology see (Rammig et al., 2018).
For the evaluation of the modeled large-scale AGB pattern ($Y$) against inventory-based biomass estimates ($X$) we employ
three metrics to detect deviations in important pattern properties: 1. The ratio of means ($\overline{y}/\overline{x}$) as a measure for the agreement
of pattern average. 2. The ratio of standard deviations of large-scale AGB patterns ($\sigma_y/\sigma_{x,LS}$) as a measure for the agreement
of pattern amplitude (the differences between grid cells). 3. The modified 'large-scale' Pearson correlation coefficient ($r_{LS}$)
as a measure for the agreement of large-scale pattern shape (the location of maxima and minima).
*Local scale evapotranspiration and productivity*
To evaluate the performance of simulated local ET and net ecosystem exchange (NEE) of the LPJmL versions used in this
study, we compare Fluxnet eddy covariance measurements of ET at 7 sites and NEE at 3sites  across the study region (Bonal
*et al.*, 2008; Saleska *et al.*, 2013, table 3) to respective simulated rates of local ET and NEE. Fluxnet data was downloaded
from https://fluxnet.fluxdata.org (under DOI: 10.18140/FLX/1440032 and DOI: 10.18140/FLX/1440165) in October 2017
and from https://daac.ornl.gov/LBA/guides/CD32_Brazil_Flux_Network.html in November 2019.
*Continental scale gridded evapotranspiration products and selection of regions*
To evaluate the ET over large regions and during a long period (1981-2010), we use three global gridded datasets: Global
Land Data Assimilation System Version 2 (Rodell et al., 2004), ERA-Interim/Land (ERAI-L, Balsamo *et al.*, 2015) and
Global Land Evaporation Amsterdam Model v3.2 (GLEAM, Miralles *et al.*, 2011; Martens *et al.*, 2017).
GLDAS and ERAI-L are land-reanalysis products, meaning that they are land surface models forced with meteorological
data that has been corrected with observations to give better estimates of land surface variables. The selection of these two
products is based on the study of Sörensson and Ruscica (2018), who found that they have a better performance over South
America than other reanalysis and satellite-based ET products. GLDAS uses the land surface model Noah (Ek et al., 2003)
forced by Princeton meteorological dataset version 2.2  (Sheffield et al., 2006). The soil depth of Noah is 2 m and the model
uses four soil layers and vegetation data from University of Maryland (http://glcf.umd.edu/data/landcover/). ERAI-L uses the





land surface model HTESSEL (Hydrology-Tiled ECMWF Scheme for Surface Exchanges over Land, Balsamo *et al.*, 2009)
forced by ERA-Interim atmospheric data with a GPCP based correction of monthly precipitation. The soil depth of ERAI-L
is 2.89 m, the model uses four soil layers and vegetation data from ECOCLIMAP (Masson et al., 2003).
GLEAM uses the Priestley-Taylor equation to estimate the potential ET and a set of algorithms with meteorological and
vegetation satellite data as input to calculate the actual ET. The version used here, GLEAMv3.2a (Martens *et al.*, 2017,
downloaded from https://www.gleam.eu/#downloads) uses precipitation input from MSWEP v1.0 (Beck et al., 2017),
vegetation cover from the MODIS product MOD44B, remotely sensed Vegetation Optical Index from CCI-LPRM (Liu et
al., 2013) and assimilates soil moisture from both remote sensing (ESA CCI SM v2.3, Liu *et al.*, 2012) and land-reanalysis
(GLDAS Noah, Rodell *et al.*, 2004).
For the temporal analysis of ET we used five climatological regions across the study area called Northern South America
(NSA), Equatorial Amazon West (EQ W), Equatorial Amazon East (EQ E), Southern Amazon (SAMz), and South American
Monsoon System region (SAMS) (see Figure 11f). These regions result from a K-means clustering analysis of the annual
cycles of the main drivers of ET: precipitation and surface net radiation (for details see Sörensson and Ruscica, 2018). For
the purpose of this study we divided the large EQ region used by Sörensson and Ruscica (2018) in two smaller (EQ W and
EQ E) at 60ºW, since this is the approximate division between regimes that have a maximum climatological water deficit
(MCWD) of around -200 mm per year (EQ W), and of around -500 mm per year (EQ E). MCWD is an indicator of seasonal
water stress (see section 2.5.3).
The original spatio-temporal resolution of GLDAS and GLEAM is 0.25º x 0.25º while for ERAI-L it is 0.75º x 0.75º.
Monthly time series were calculated from daily values for the three datasets. Hereafter, we use the short names GLDAS,
ERAI-L and GLEAM for the described reference datasets.
*Spatial distribution of vegetation types*
To evaluate the regional distribution of simulated biome types in all LPJmL versions we compare our results to satellite-
derived vegetation composition maps from ESA Land cover CCI V2.0.7 (Li et al., 2018) reclassified to the PFTs of LPJmL
from Forkel *et al.* (2014). In this dataset PFT dominance is indicated by foliage projected cover (FPC) which is also a
standard output variable of all LPJmL models enabling a direct comparison of model results.
*Spatial pattern of rooting depth*
We compare regional patterns of mean rooting depth simulated with LPJmL4.0-VR to a maximum depth of root water
uptake map (Fan et al., 2017) regridded to the 0.5° x 0.5° spatial resolution of LPJmL4.0-VR. This product was inversely
modelled by taking the dynamically interacting variables soil water supply and plant water demand into account. Here,
supply was based on climate, soil properties and topography and demand on plant transpiration deduced from satellite based
reanalysis of atmospheric water fluxes and leaf area index (LAI) data.
**2.5.2 Validation metrics**
Except for inventory biomass all statistical evaluations of model results were based on 1) Pearson Correlation and 2)
normalized mean squared error (NME; Kelley *et al.*, 2013). NME is calculated as:
$$NME = \frac{\sum_{i=1}^{N} |y_i - x_i|}{\sum_{i=1}^{N} |x_i - \bar{x}|}$$   Eq. (9)
where $y_i$ is the simulated and $x_i$ the reference value in the grid cell or time step $i$. $\bar{x}$ is the mean reference value. NME takes
the value 0 at perfect agreement, 1 when the model performs as well as the reference mean and values > 2 indicate complete
disagreement.

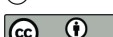


**2.5.3 Maximum cumulative water deficit as indicator of seasonal water stress**

For this study we use the maximum cumulative water deficit (MCWD) as an explanatory variable, since it is a widely used indicator for seasonal water stress for studies in South America (Aragão et al., 2007; Lewis et al., 2011; Malhi et al., 2009). MCWD captures the seasonal difference of ET and precipitation in a cumulative way and therefore reveals dry season strength and duration. Here we calculate MCWD on a monthly basis. Therefore, we first calculate the cumulative water deficit $CWD_n$ of each month $n$ as:

$$CWD_n = CWD_{n-1} - PET_n + P_n \qquad \text{Eq. (10)}$$

where PET is the potential monthly ET and P the monthly sum of precipitation. CWD is constrained to values <=0 and is set to 0 at the end of each hydrological year, here the last day of September, as in Lewis *et al.* (2011). We use $P$ from climate input used for model forcing (Sect. 2.3.1) and PET as it is simulated by LPJmL4.0 (Schaphoff et al., 2018) which is only dependent on net surface radiation and air temperature, therefore remaining an explanatory variable independent of vegetation dynamics. We chose this PET over using the commonly used constant ET of 100 mm/month to calculate CWD (Aragão et al., 2007; Lewis et al., 2011; Malhi et al., 2009), because in this way, the CWD better corresponds to the actual climatological conditions in the different LPJmL model versions used in this study (see Sect. 2.4). MCWD is then calculated as:

$$MCWD_y = min \left(CWD_{October,y-1}, \dots, CWD_{September,y}\right) \qquad \text{Eq. (11)}$$

where $y$ indicates the calendrical year.

**3 Results**

**3. 1 Local and regional pattern of tree rooting strategies**

The results of LPJmL4.0-VR show a high variation in dominance and compositions of tree rooting strategies across the study region. The contribution of each tree rooting strategy to the overall net primary productivity (NPP) appears highly dependent on local environmental conditions. Comparisons at the local scale show that shallow-rooted (deep-rooted) sub-PFTs contribute more to the overall NPP under generally wetter (drier) and less (more) seasonal climate conditions (Fig. 5). At the Fluxnet site MAN K34, which exhibits a mean annual precipitation (MAP) of 2609 mm and a mean MCWD of -222 mm under CRU climate input (2001-2010), the sub-PFT with a maximum rooting depth ($D_{95\_max}$) of 0.5 m  contributes most to overall NPP and the whole distribution of NPP weighted $D_{95\_max}$ classes shows a mean of 1.52 m (Fig. 5a). At the Fluxnet site STM K67, which exhibits a lower MAP of 2144 mm and a stronger dry season reflected in a mean MCWD of -465 mm, the NPP weighted distribution of $D_{95\_max}$ shows a peak at 10 m and a corresponding mean of 10.26 m (Fig. 5b). Since both sites have a soil thickness of 20 m (according to the soil depth input; Sect. 2.3.2) differences in rooting strategy compositions must emerge from climatic differences. It is important to note that $D_{95\_max}$ values in Fig. 5 do not necessarily reflect the true achieved rooting depth of each sub-PFT, but the maximum value. For reasons of visual clarity for this figure we kept the bins of the x-axes as chosen in Tab. 1.

Based on this NPP information of each sub-PFT in each grid cell we derived maps of mean rooting depth over the whole study region for the time span 2001-2010 for each climate input used in this study (Fig. 6). In contrast to Fig. 5 we computed the mean of the actually achieved $D_{95}$ of each sub-PFT (evergreen and deciduous combined) weighted by the respective relative NPP contribution of each sub-PFT to total forest NPP (we call $\overline{D_{95}}$). The regional pattern of $\overline{D_{95}}$ is a result of environmental filtering and sub-PFT competition, reflecting the effects of climate and sediment thickness. A general East to West gradient of $\overline{D_{95}}$ over the Amazon region follows climatic gradients of precipitation and MCWD (Fig. S1-2) while soil depth (Fig. 4) constrains $\overline{D_{95}}$ especially in the South-eastern Amazon (compare Fig. 4 & 6). In general, areas with higher





mean annual rainfall and weaker dry season show lower $\overline{D_{95}}$ and vice versa. This pattern holds true under all climate inputs,
with some minor local differences.
Focussing on the climatological clusters (see Sect. 2.5.1 and Fig. 9f) under CRU climate input, the western Amazon (EQ-W),
with a MAP of 2708 mm and MCWD of -163 mm, displays an overall mean $\overline{D_{95}}$ of 1.14 m and a maximum of 5.47 m,
despite considerably deeper soils present. In the Northern, Western and Southern Amazon clusters (NSA, EQ E, SAMz) with
lower MAP of 2299, 2190 and 2035 mm and considerably lower MCWD of -488, -438 and -497 mm (meaning higher
seasonality), respectively, mean $\overline{D_{95}}$ increases to 2.32, 3.20 and 2.68 m, respectively. Here maximum $\overline{D_{95}}$ values reach 11.97,
11.27 and 9.04 m. In the monsoon dominated region (SAMS) displaying the lowest MAP of 1449 mm and MCWD of -649
mm, mean $\overline{D_{95}}$ decreases to 1.37 m. The maximum $\overline{D_{95}}$ of this region reaches 11.17 m located at the border to SAMz.
Comparing our results to an inversely modelled global gridded product of maximum depth of root water uptake (MDRU;
Fan et al. 2017) we find considerable absolute differences to simulated $\overline{D_{95}}$ while overall patterns coincide (Fig. 7). As $\overline{D_{95}}$
(Fig. 6) also the original product by Fan et al. (2017) regridded to LPJmL4.0-VR's spatial resolution (Fig. 7a) shows a
northwest to southeast gradient of MDRU across the Amazon region. Lowest mean MDRU is found in cluster EQ W with
1.38 m, followed by NSA with 2.98 m, SAMz with 5 m, SAMS with 5.47 m and EQ E with 5.88 m. All cluster have a
maximum MDRU > 20 m with the highest value found in SAMS with 64.4 m. Fig. 7c shows a difference map between
MDRU and simulated $\overline{D_{95}}$ using CRU climate input. Largest differences are found over a wide area (most pronounced in EQ
E, SAMz and SAMS) especially where MDRU exceeds $\overline{D_{95}}$. It appeared that for many grid cells in this area MDRU even
exceeds the soil depth input used in this study (2.3.2) substantially. To overcome this technical bias we set MDRU to our soil
depth input values in cases where MDRU exceeded them (Fig. 7b) to make MDRU and $\overline{D_{95}}$ more comparable. The
differences between this adjusted MDRU and $\overline{D_{95}}$ are more likely caused by model architecture than prescribed abiotic
limits, enabling for a more meaningful comparison. After this adjustment mean and maximum values of MDRU in the
clusters converge to results of LPJmL4.0-VR by decreasing to 1.85 and 14.28 m for NSA, 1.26 and 17.95 m for EQ W, 2.84
and 13.47 m for EQ E, 3.28 and 16.57 m for SAMz, and 2.61 and 49.37 m for SAMS. Consequently, the geographical
pattern of $\overline{D_{95}}$ and adjusted MDRU shows a better agreement (Fig. 7d). Largest differences remain in north-western NSA,
eastern EQ W, along the Amazon River in EQ E and in eastern SAMz where $\overline{D_{95}}$ exceeds MDRU. On the other hand MDRU
substantially exceeds $\overline{D_{95}}$ in south-western SAMz and south-western SAMS. These differences might easily emerge from
different model settings and assumptions, e.g. related to differences in spatial model resolution, simulated water percolation
and underlying vegetation features.
The regional validation of $\overline{D_{95}}$ now allows us to generalize which tree rooting strategies occupy which climate space. Using
MCWD and MAP to define a climate space we find a clear separation of $\overline{D_{95}}$ (Fig. 8). A core region with deep-rooted forests
(mean $\underline{D_{95}}$ > 4 m) is found where MCWD ranges between -1300 and -400 and where MAP is at least 1500 mm (see maps of
MCWD and MAP in Fig. S1-2) if soil depth allows for it. This core region is surrounded by a small band of medium rooting
depth forests (mean $\overline{D_{95}}$ ~ 2-4 m) forming a crescent shape. Rather shallow-rooted forests (mean $\overline{D_{95}}$ < 2 m) are found where
MAP is less than 1000 mm and MCWD is below -500 mm, i.e. in increasing seasonally dry climates with MAP at the edge
to support closed tropical evergreen forest. Shallow-rooted forests are also simulated in very wet conditions where MCWD is
greater than -300 mm and MAP is 1200 mm or higher.
**3.2 Evapotranspiration rates and productivity**
**3.2.1 Local evapotranspiration**
Differences of intra-annual ET rates between LPJmL4.0, LPJmL4.0-VR and LPJmL4.0-VR-base are most pronounced at
Fluxnet sites showing a high seasonality of rainfall (Fig. 9b, e, g and Fig. 10b, e, g). Here, the results of LPJmL4.0-VR show
how variable tree rooting strategies lead to a major improvement of matching measured Fluxnet NEE and ET expressed in

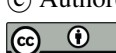



reduced NME and increased r²-values (Table 4 and 5). This improvement arises from an important new model behaviour:
Whereas, LPJmL4.0-VR-base and LPJmL4.0 simulate decreasing ET and increasing NEE during dry seasons at these sites,
which is anticorrelated to Fluxnet measurements, LPJmL4.0-VR shows the opposite and follows the Fluxnet signals. This
means LPJmL4.0-VR's variable rooting strategies buffer precipitation deficits by usage of deep water. Together with a
generally lower mean cloud cover during the dry season, this leads to an increase of productivity and ET as suggested by
numerous studies (Nemani et al., 2003; da Rocha et al., 2004). Most pronounced improvements are found at STM K67 and
STM K83 where NME of ET and NEE drop below or close to 1 and r²-values considerably increase compared to the other 2
model versions (Table 4-5). For STM K67 r² of NEE is higher under LPJmL4.0 and LPJmL4.0-VR-base, but this refers to a
significant anti-correlation. At STM K77 (Fig. 9f) the influence of variable rooting strategies is reversely demonstrated. This
former rainforest site was converted to pasture before Eddy covariance measurements began. This local land-use at STM
K77 is not representative for the respective 0.5° grid cell, and all 3 LPJmL model versions simulate natural forest. In this
case, the shallow rooting systems of LPJmL4.0 and LPJmL4.0-VR-base show a better match to ET measurements.
Nevertheless, at STM K83, a selectively logged primary forest site which shares the same model grid cell as STM K77 due
to their geographical proximity, LPJmL4.0-VR is the only model reproducing increased ET and decreased NEE during the
dry season. At sites with weaker to no dry season (Fig. 9c, d, h) differences between model versions become less
pronounced, as water availability is more stable throughout the year leading to less variability in ET. Generally, all models
show a better match with ET than with NEE, most likely explainable by the fact that DGVMs a) miss or underestimate
important mechanisms driving seasonal productivity and respiration and b) have a coarse spatial resolution and therefore
miss site specific environmental factors. The latter might also explain why LPJmL4.0-VR overestimates ET at GF GUY in
the dry season. Here the soil depth input for the corresponding grid cell most likely exceeds the soil depth at this site, thus
the model overestimates rooting depth and resulting ET.

**3.2.2 Continental Evapotranspiration**

Results of regional ET are in line with results of site-specific ET. The climatological clusters within the Amazon region
which undergo the strongest dry season (EQ E and SAMz) show the largest differences between LPJmL4.0-VR and the other
2 models and also here LPJmL4.0-VR shows a higher agreement with validation data (Fig. 11c, d and Table 6).
Improvement is largest for EQ E where NME and r² show values of 0.62 and 0.91, respectively, whereas the other 2 models
show values of NME >= 1.92 and r² <= 0.21. As expected in NSA and EQ W model differences become less pronounced as
annual precipitation deficits are lower and deep rooting systems play a lesser role, but still there is noticeable improvement
e.g. in NSA between January and April, where monthly precipitation is lower than during the rest of the year. In the
monsoon dominated cluster outside the Amazon region (SAMS) model differences are least pronounced, since shallow
rooting forests in LPJmL4.0-VR dominate this area (Fig. 5) which are similar to the forests with constant rooting systems in
the other 2 model versions.

**3.3 Biome distributions**

The simulated relative dominance of tropical tree PFTs across the study area differs substantially between model versions
(Fig. 12). More than half of the grid cells of LPJmL4.0 show the evergreen and deciduous PFTs equally dominant (Fig. 11g-
h). Only in areas outside tropical moist climate regions the model shows a clear dominance of the deciduous PFT, whereas
e.g. in the Amazon region evergreen and deciduous PFTs co-exist in almost equal abundance. These patterns strongly differ
to satellite-derived vegetation composition maps (Fig. 12a-b) and therefore yield in respective comparisons the highest NME
values among all models (Table 7). In contrast LPJmL4.0-VR and LPJmL4.0-VR-base show clear dominance patterns of
both tropical tree PFTs across the study area (Fig. 12c-f). This can be attributed to the dominance dependent PFT
establishment introduced in this study (Sect. 2.2.5) and applied to LPJmL4.0-VR and LPJmL4.0-VR-base, which makes it





possible that one PFT (or sub-PFT) can fully outcompete others. Nevertheless, differences between LPJmL4.0-VR and
LPJmL4.0-VR-base are quite substantial. In LPJmL4.0-VR-base the tropical evergreen PFT dominates the North-Western
Amazon region only, negligibly extending further than the borders of climatological clusters NSA and EQ W combined.
Beyond these borders the tropical deciduous PFT is dominating. In contrast, in LPJmL4.0-VR (Fig. 12e-f) the evergreen tree
PFT dominance extends closer to its observed borders including EQ E and SAMz, and the deciduous PFT is pushed towards
drier and more seasonal climate (including parts of SAMS). Therefore, LPJmL4.0-VR yields lowest NME values in
comparison to satellite-derived vegetation composition maps (Table 7).

### 3.4 Aboveground biomass (AGB)

### 3.4.1 Regional AGB pattern

The simulated mean AGB pattern (2001-2010) of LPJmL4.0-VR (Fig. 13c) shows how deep water access produces a
contiguous high biomass over the Amazon region. Especially towards the borders of the South-Eastern Amazon region in the
climatological clusters EQ E and SAMz AGB values appear rather homogenous in contrast to the other 2 model versions
(Fig. 13d-e). In connection with the significantly improved underlying vegetation composition (Fig. 12) it is clear that
LPJmL4.0-VR is the only model version capable of simulating high AGB evergreen rainforests across the climatic gradient
of the Amazon region (Fig. S1-2). This pattern is suggested by one satellite derived AGB  product chosen for evaluation of
our model results (Saatchi *et al.*, 2011; Fig 12b) which yields a corresponding NME close to 0 (Table 8), even though this is
true for all model versions. Surprisingly, for the other AGB validation product (Avitabile *et al.*, 2016b; Fig. 12a) LPJmL4.0-
VR-base yields a smaller NME than LPJmL4.0-VR. Taking into account the significantly less accurate underlying
vegetation composition of LPJmL4.0-VR-base (Fig. 12) we regard the comparison as obsolete in this context. The same
holds true for LPJmL4.0. A known problem with AGB maps for South America is their poor overall agreement especially in
the Amazon region (Mitchard et al., 2014), making it hard to interpret such geographical evaluations. The divergence
between the 2 AGB evaluation products chosen for this study clearly displays this problem (Fig. 13a-b). Therefore, we also
conducted a site specific AGB comparison with results in the following section (Sect. 3.4.2).

### 3.4.2 AGB at specific sites

For site specific comparisons of simulated and observed AGB we calculated 3 indicators, 1) the ratio of means ($\overline{y}/\overline{x}$) as a
measure for the agreement of pattern average, 2) the ratio of standard deviations of large-scale AGB patterns ($\sigma_y/\sigma_{x,LS}$) as a
measure for the agreement of pattern amplitude (the differences between grid cells), and 3) the modified 'large-scale'
Pearson correlation coefficient ($r_{LS}$) as a measure for the agreement of large-scale pattern shape (the location of maxima and
minima).
Fig. 14 shows a site-specific AGB comparison for LPJmL4.0, LPJmL4.0-VR and LPJmL4.0-VR-base for the four climate
input data sets used in this study against inventory data from Brienen *et al.* (2015). We find that for all climate datasets,
LPJmL4.0 tends to overestimate and LPJmL4.0-VR-base tends to underestimate average AGB across forest plots in the
Amazon region. Except for GLDAS, average AGB from LPJmL4.0-VR lies between these two cases, showing the closest
match with average AGB derived from forest plots. However, uncertainties in average AGB from forest plots is quite large
(as indicated in spread of violine) so that for all but two cases (LPJmL4.0-VR-base with GSWP3 and WATCH+WFDEI)
$\overline{y}/\overline{x} = 1$ falls within the 95 % confidence interval of $\overline{y}/\overline{x}$.
With regard to the pattern's amplitude ($\sigma$), we find that for all climate datasets all model versions tend to overestimate AGB
differences across the Amazon, but only for LPJmL4.0 with GSWP3 and WATCH+WFDEI unity is outside the 95 %
confidence interval of $\sigma_y/\sigma_{x,LS}$. In other words the spatial difference between grid cell biomass is generally larger than
observations imply. Nevertheless, pattern amplitude decreases with increasing model complexity (from LPJmL4.0 over

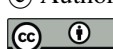



LPJmL4.0-VR-base to LPJmL4.0-VR) so that for LPJmL4.0-VR unity falls within the interquartile range of $\sigma_y/\sigma_{x,LS}$ for all
climate datasets. Note, however, that for GLDAS median $\sigma_y/\sigma_{x,LS}$ for LPJmL4.0-VR is slightly larger than for LPJmL4.0-
VR-base but the 25 % percentile is lower for LPJmL4.0-VR due to the wider uncertainty distribution.
Evaluation of the shape of the large-scale average AGB pattern shows that median $r_{LS}$ increases with increasing model
complexity. In other words LPJmL4.0-VR matches large scale maxima and minima of biomass across the Amazon forests
best. Highest median $r_{LS}$ are found for LPJmL4.0-VR with 0.43 for CRU, GSWP3, and WATCH+WFDEI and 0.51 for
GLDAS (upper bounds of the 95 % interval are 0.61 for CRU, 0.68 for GLDAS, and 0.48 for GSWP3 and
WATCH+WFDEI).
In summary, we conclude that LPJmL4.0-VR reproduces the 'observed' large scale AGB pattern in the Amazon in all three
relevant aspects (pattern mean, amplitude and shape) better than either of the two other model versions. Still LPJmL4.0-VR
cannot completely reproduce all features of the large-scale AGB pattern in the Amazon, which points to bias in model input
connected to climate and soil depth as well as insufficient representation of other important processes for modelling carbon
dynamics in tropical forests such as tree mortality (Pillet et al., 2018), gap dynamics (Espírito-Santo et al., 2014), and
nutrient limitation (Quesada et al., 2012). However, it is important to acknowledge that AGB estimates derived from
inventory plots may be subject to large errors and spatial biases themselves (Saatchi et al., 2015).
**3.5 Belowground biomass**
Simulations with LPJml4.0-VR enable an unprecedented analysis of root carbon pools due to the implementation of
belowground carbon investment into tree coarse root structures (Sect. 2.2.3). Fig. 15 shows the mean sum (2001-2010) of
coarse and fine root carbon pools of tropical evergreen and deciduous tree PFTs under CRU climate over the study region.
As expected the pattern follows simulated mean rooting depth (Fig. 6) as coarse root carbon investment increases
accordingly. In the Amazon region drier and more seasonal climate selects for sub-PFTs with deeper tree rooting strategies
which comes with higher investments into below-ground root structures, implying lower growth rates of these forests
compared to wetter and less seasonal regions.
**4 Discussion**
This study demonstrates a generalizable approach to improve the representation of tree root system diversity in a DGVM by
employing gradual root growth and a trade-off between below- and aboveground carbon investment. A major advance of the
new sub-model version LPJmL4.0-VR is that simulations start with a uniform input distribution of tree rooting strategies for
each PFT (tropical evergreen and deciduous) in each location, thus ensuring that all tree rooting strategies have the same
chance to establish. This uniform distribution then shapes into a local distribution of abundance driven by local
environmental conditions and competition (Fig. 5). Therefore, these distributions are not a pre-selected input, but a model
output, enabling to investigate patterns like mean rooting depth over the study region (Fig. 6, 7). Since the simulated
vegetation can now adjust its root systems to environmental conditions, the quality of simulated biome distributions (Fig. 12)
and subsequently the quality of simulated ET and NEE fluxes (Fig. 9-11) and state variables like AGB (Fig. 13-14) is
considerably increased.
**4.1 Climate and soil determine tree rooting strategies**
Simulated $\overline{D_{95}}$ (Fig. 6) clearly follows climate gradients and soil depth of the study region (Fig. 4, Fig. S1). Our findings are
in line with the general ecological expectation and former studies that seasonal water depletion of upper soil layers, as a
combination of annual precipitation sums and dry season length and strength, is positively correlated with the rooting depth
of tropical evergreen trees (Baker et al., 2009; Ichii et al., 2007; Kleidon and Heimann, 1998, 1999). We also find lower





thresholds for MAP and MCWD were $\overline{D_{95}}$ strongly decreases again which can be explained by different mechanisms leading
to a regime shift from the evergreen to the deciduous growing strategy as discussed below (see 4.2).
In this study, we focus on the NPP weighted mean rooting depth ($\overline{D_{95}}$) to detect the tree rooting strategies which are most
important for water and carbon fluxes (ET, NEE) as well as biomass. The comparisons of mapped mean MDRU of Fan *et al.*
(2017) to $\overline{D_{95}}$ (Fig. 9) should be treated with caution as the latter does not necessarily reflect the productivity nor the
dominance of certain tree rooting strategies. Fan *et al.* (2017) back-calculate the necessary water uptake depth to meet
observed plant productivity derived from satellites while our results are based on DGVM simulations which yield
communities of different tree rooting systems. A number of additional general differences of both approaches and underlying
assumptions could have easily led to the observed mismatches: 1) Fan *et al.* (2017) use a different soil depth input, i.e.
assuming a different physical boundary of maximum rooting depth. Even though we adjusted MDRU of Fan *et al.* (2017) to
the soil depth input used in this study (Fig. 8b), this adjustment was only for cases where MDRU exceeds our soil depth
input. Therefore, cases where adjusted MDRU exceeds simulated $\overline{D_{95}}$ in Fig. 8d, e.g. in western SAMz, could be caused by a
higher soil depth input assumed in Fan *et al.* (2017) for the respective grid cells. 2) LPJmL4.0-VR simulates the growth and
competition of (sub-)PFTs on the basis of leaf level photosynthesis and allocation of accumulated carbon. Their traits, such
as the rooting strategy, determine performance and subsequently competitiveness. Therefore, competition could lead to a
different $\overline{D_{95}}$ as would be expected when considering water supply and demand of each (sub-)PFT alone. 3) Satellite derived
productivity of tropical vegetation can be biased, e.g. due to strong cloud cover all year round, potentially leading to biased
plant water demand and deduced MDRU. 4) Different water percolation schemes and soil textures in both models lead to
different seasonal plant water supply determining MDRU and $\overline{D_{95}}$. 5) LPJmL4.0-VR does not employ a ground-water model
or static ground-water table. By considering ground-water aquifers the simulated dominance of tree rooting strategies and
consequently $\overline{D_{95}}$ could locally shift towards lower values, e.g. in the EQ-E and SAMz cluster, if ground-water depth would
be lower than the simulated $\overline{D_{95}}$. Applying a ground-water model in LPJmL4.0-VR is in the focus of future studies. 6) The
tropical deciduous PFT of LPJmL4.0-VR avoids water stress during the dry season by shedding its leaves. Therefore the
need for deeper roots to withstand a dry season is relatively low. Generally, areas where the deciduous PFT dominates, e.g.
the most southwestern part of the study region (Fig. 6), display a low $\overline{D_{95}}$ whereas this area shows amongst the highest
values of MDRU in Fan *et al.* (2017). Since deciduous tree types dominate this area also in reality (Fig. 12b), MDRU values
might be overestimated. 7) LPJmL4.0-VR does account for tropical tree PFTs only. Bush and shrub PFT types which might
be evergreen and gain access to deep water while stem size remains relatively small (Oliveira et al., 2005) are not accounted
for. Implementing more general PFTs into LPJmL4.0-VR is in the focus of future studies.
For this study we checked the data availability on maximum rooting depth across South America in the TRY database
(Kattge *et al.*, 2020; data downloaded September 2019). As it is also shown in Fan *et al.* (2017; Fig. 2) we found the number
of sites within the TRY data base where maximum rooting depth has been measured in South America to be very low.
Moreover, the number of data entries per site appeared very small, where 33 TRY sites falling within our study area showed
a mean of ~9 and a median of 6 data entries, while 15 sites showed <=5 data entries. Therefore, we decided to not include
site specific comparisons of rooting depth as it is not clear how representative these measurements are for the local forest
communities. Nevertheless, as shown in Fan *et al.* (2017; Fig. 2) measured site-specific maximum rooting depth across the
Amazon region seems to follow the expected climatic gradient and gives confidence to our results. More measurements
gathered in openly available databases like TRY will help to evaluate future simulation results more sophisticly.
**4.2 Rooting depth influences the distribution of tropical biomes and biomass**
Seasonal water deficit and annual precipitation are the main determinants of which tree rooting strategies perform best, are
able co-exist and outcompete others in LPJmL4.0-VR (Fig. 8). Avoiding seasonal drought stress due to deep roots broadens
the geographical extent of simulated tropical evergreen forest. This vegetation type appears to be competitive over a



substantially wider climatic range than anticipated when employing the tree rooting strategies of LPJmL4.0. With
LPJmL4.0-VR, drier and more seasonal environments now appear suited for the evergreen PFT (Fig. 10). Below certain
thresholds of annual precipitation (around 1000 mm) and of MCWD (around -500 mm) mean $\overline{D_{95}}$ decreases again, indicating
a transition from the evergreen to the deciduous growing strategy or more open grasslands (Fig. 8). Whether this transition
for each of those thresholds is mainly caused by (a) environmental filtering of deep tree rooting strategies, (b) their
competitive exclusion by shallow rooted deciduous tree types together with grass PFTs, (c) fire feedbacks or most probably,
a combination of all is yet to be determined and in the focus of further studies.
The climatic thresholds of vegetation types we find are comparable to thresholds between evergreen forests and savannah
found by e.g. Malhi *et al.* (2009) at an annual precipitation of 1500 mm and at an MCWD of -300 mm. The substantially
lower MCWD value found in this study can be explained by the differences in calculating CWD. While Malhi *et al.* (2009)
assume a constant rate of ET per month of 100 mm, we use the monthly variable PET (Sect. 2.5.3). Since PET often is
significantly higher than 100 mm our monthly CWD and therefore MCWD values are respectively lower. Similarly to Malhi
*et al.* (2009), Staver, Archibald and Levin (2011) find that evergreen tree cover appears to be bi-modal within a range of
MAP of 1000-2500 mm and ascribe this to climate-fire-vegetation feedbacks. Many recent studies investigating potential
forest-savanna bi-stability and tipping points of the Amazon region rely solely on such climatic ranges of tropical biomes
(Hirota *et al.*, 2011; Wuyts, Champneys and House, 2017; Zemp *et al.*, 2017; Staal *et al.*, 2018; Ciemer *et al.*, 2019). The
results of LPJmL4.0-VR show that knowledge on local tree root adaptations is another important explanatory variable of
vegetation cover reducing the uncertainty and width of anticipated climatic ranges where evergreen tree cover can be bi-
modal. This will help future studies to quantify climate-fire-vegetation feedbacks, forest resilience and potential individual
tipping points of forests in the Amazon region in a new way.
Especially the current and potential extent of evergreen forests into drier and more seasonal environments can be better
explained when considering local adaptations of tree rooting strategies. In these environments an evergreen growing strategy
requires deeper root systems to access deep water. Deeper roots require higher BGB investments (Fig. 12) which on the one
hand side has a negative effect on productivity, because during growth the allocation of assimilated carbon shifts towards
respiring BGB, while investments into productive AGB need to be reduced. On the other hand drier and more seasonal
environments show less cloud cover during the dry season (Nemani et al., 2003), enhancing photosynthesis which increases
productivity as long as water access is assured (Costa et al., 2010; Wu et al., 2016). The trade-off between AGB and BGB
investment most probably leads to a more homogenous AGB pattern across the Amazon region with similar values over a
wide climatic range (compare EQ E and SAMz in Fig. 13c-e). This effect is also visible in lower amplitudes and higher
correlation in the large scale AGB pattern from different evaluation sites (Fig. 14).
In fact comparisons of biomass pattern between all model versions of this study and different biomass products are difficult,
since only LPJmL4.0-VR shows a reasonable geographical distribution of underlying biome types across the study area (Fig.
12, Table 7). Therefore, differences in biomass are not solely the consequence of different productivities directly related to
diversity in tree rooting strategies, but also the consequence of simulated biome type which can be regarded as an indirect
effect of diversity in tree rooting strategies. In LPJmL4.0-VR the evergreen growing strategy dominates the entire Amazon
region, which is more productive and accumulates more biomass than the deciduous growing strategy. The latter dominates
EQ E and SAMz in LPJmL4.0-VR-base and is equally abundant throughout the Amazon region in LPJmL4.0.
Concentrating on LPJmL4.0-VR only, the model matches substantially better with the gridded biomass product of Saatchi et
al. (2011b), since this product shows generally higher biomass values across the Amazon region which are more similar to
LPJmL4.0-VR (Table 8). Therefore, the differences in NME are mainly caused by mean biomass values of rainforests across
the whole study area rather than pattern divergence. Thus, we argue lowering overall biomass values in LPJmL4.0-VR
would improve its match with (Avitabile et al., 2016b) which is a matter of adjusting overall maximum tree mortality rates
(see Sect. 2.2.5). Differences to site-specific measurements (Fig. 14) are rather caused by additional factors, such as a) the



coarse model resolution leading to a different climate and soil information input than found at specific sites and b)
insufficient representation of important processes forcing carbon dynamics in tropical forests such as tree mortality (Pillet et
al., 2018), gap dynamics (Espírito-Santo et al., 2014), and nutrient limitation (Quesada et al., 2012).

### 4.3 Diverse tree rooting strategies improve simulated evapotranspiration and productivity

In LPJmL4.0-VR variable tree rooting strategies decrease the intra-annual variability of ET and maintain high rates of NEE
and ET during the dry season in accordance with the intra-annual trends suggested by evaluation data (Fig. 9-11). More than
that simulated rates of ET and productivity peak during the dry season in EQ E which is explained by increased solar
radiation while trees having access to deep water in the model and in reality (Costa et al., 2010; Wu et al., 2016). While
recent parameter optimization against FAPAR data (Forkel et al., 2015) tried to improve the simulated productivity by
adjusting phenology pattern in LPJmL4.0, the seasonal offset in simulated ET for Fluxnet sites in the Amazon region
remained a challenge (Schaphoff et al., 2018). In this study we can show for the first time on the regional scale how PFTs
with variable tree rooting strategies adjust to local environmental conditions and in return improve simulated rates of ET and
NEE (Fig. 9-11). Being able to mechanistically reproduce and explain this broad-scale stabilization of water fluxes into the
atmosphere has wide implications for DGVM modelling frameworks and simulation of ET as moisture input to the
atmosphere in Earth System Models (ESMs). Our approach can help to better quantify the role of forests for local-to-
continental scale moisture recycling and to project the fate of forests under future climate and land-use change. The approach
presented here is easily applicable for a wide range of DGVMs and ESMs which simulate fine root distribution in a similar
way as the LPJmL model family (based on Jackson *et al.*, 1996). A first and easy to implement step for other models could
be to prescribe the relative fine root distribution in a spatial explicit way in accordance to the mean rooting depth ($\overline{D_{95}}$)
presented in this study.

### 5 Conclusions

In this study we show for the first time that diverse tree rooting strategies across South-America can indeed explain the
spatial distribution of biome types, biomass, as well as the spatial and temporal pattern of the ecosystem fluxes of ET and
NEE even when the competition of tree rooting strategies, carbon investment into gradually growing deep roots, and local
soil depth are considered. Because LPJmL4.0-VR allows for a whole spectrum of tree rooting strategies, where each strategy
has the same theoretical chance to establish in every location, the simulated local distributions of tree rooting strategies are
an emergent simulation output which results from local environmental filtering and competition. This enables to estimate
mean rooting depth and below-ground biomass on a continental scale as presented here, as well as future estimates of
functional diversity of tree rooting strategies. Moreover, we conclude that tree root adaptation is a key explanatory variable
to explain forest cover and to estimate the climatic range of potential forest cover bi-stability in connection with climate-fire-
vegetation feedbacks in tropical regions. Generally, we are convinced that our approach is of high importance to all
modelling frameworks of DGVMs and Earth System Models (ESMs) aiming at quantifying continental scale moisture
recycling, forest tipping points and resilience. So far the continental scale importance of local scale tree root adaptations
shows that this potential treasure of below-ground functional diversity must be protected not only in the scope of future
global change.

### 6 Code availability

In case of manuscript acceptance all model code and post-processing scripts will be made available. The first author of this
manuscript is also willing to share all information with all reviewers upon request.



**7 Data availability**

In case of manuscript acceptance all simulation data will be made available. The first author of this manuscript is also willing to share all information with all reviewers upon request.

**8 Author contribution**

All authors helped in conceptualizing the model. BS and WvB developed the model code. BS, WvB, MD, AS, RR, FL, MB, SB, MH, RO, KT conceived the simulation experiments and BS carried them out. BS, MD, AS, RR and JH analysed model output data. BS prepared the manuscript with contributions from all co-authors.

**9 Competing interests**

The authors declare that they have no conflict of interest.

**10 Acknowledgements**

BS and KT acknowledge funding from the BMBF- and Belmont Forum-funded project "CLIMAX: Climate services through knowledge co-production: A Euro-South American initiative for strengthening societal adaptation response to extreme events", FKZ 01LP1610A. MD is funded by the DFG/FAPESP within the IRTG 1740/TRP 2015/50122-0. MH is supported by a grant from Instituto Serrapilheira/Serra 1709-18983. This work used eddy covariance data acquired and shared by the FLUXNET community, including these networks: AmeriFlux, AfriFlux, AsiaFlux, CarboAfrica, CarboEuropeIP, CarboItaly, CarboMont, ChinaFlux, Fluxnet-Canada, GreenGrass, ICOS, KoFlux, LBA, NECC, OzFlux-TERN, TCOS-Siberia, and USCCC. The ERA-Interim reanalysis data are provided by ECMWF and processed by LSCE. The FLUXNET eddy covariance data processing and harmonization was carried out by the European Fluxes Database Cluster, AmeriFlux Management Project, and Fluxdata project of FLUXNET, with the support of CDIAC and ICOS Ecosystem Thematic Center, and the OzFlux, ChinaFlux and AsiaFlux offices.

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

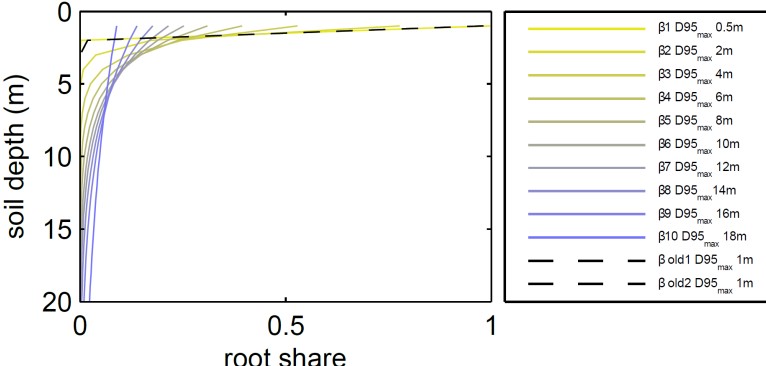

**Figure 1: Fine root distributions in LPJmL4.0 and fine root distribution at maximum rooting depth in LPJmL4.0-VR as the**
**relative amount of fine roots over soil depth. In the legend "β old1-2" correspond to the β-values of the 2 tropical tree PFTs**
**(deciduous and evergreen) employed in LPJmL4.0. The corresponding graphs lie on top of each other due to marginal differences**
**in their β-values. "β1-10" correspond to the 10 β-values used in LPJmL4.0-VR (table 2) used to create the 10 sub-PFTs of the**
**tropical evergreen and deciduous tree PFTs (see 2.2.2). Please note, the first 3 soil layer (as described in 2.2.1) in this visualization**
**are treated as 1 layer of 1 m thickness for reasons of visual clarity.**

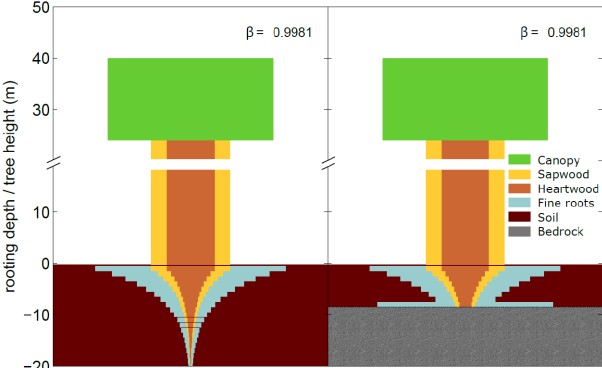

**Figure 2: Visualization of belowground carbon allocation to different carbon pools of a tree PFT in LPJmL4.0-VR with a height of**
**40m and a $D_{95\_max}$ of 14m (sub-PFT no. 8 in Table 2) growing in a grid cell with a soil depth of 20m (left panel) and a soil depth of**
**7m (right panel). As for stem sapwood, also root sapwood needs to satisfy the pipe model. In the first soil layer root sapwood cross-**
**sectional area is equal to stem sapwood cross-sectional area, as all water taken up by fine roots needs to pass this layer. In each**
**following soil layer the root sapwood cross-sectional area is reduced by the sum of the relative amount of fine roots of all soil layers**
**above, thus adjusting the amount of sapwood needed to satisfy the pipe model. Please also see Supplementary Video 1 for a**
**visualization of root growth and development of belowground carbon pools over time under http://www.pik-**
**potsdam.de/~borissa/LPJmL4_VR/Supplementary_Video_1.pptx.**

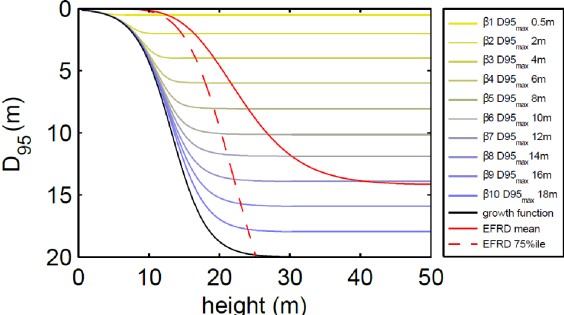

**Figure 3: Relation between tree height and rooting depth in LPJmL4.0-VR. Black line: Implemented general growth function of**
**rooting depth (Eq. 5). Lines with colour scale from yellow to blue: Growth functions of rooting depth for each of the 10 sub-PFTs**



(see 2.2.2). Here temporal rooting depth is expressed as $D_{95}$ and eventually reaches $D_{95\_max}$ (Eq. 3). Red solid line: Mean effective
functional rooting depth over tree height (EFRD) adapted from Brum *et al.* (2019) using Eq. 6. Red dashed line: Respective 75%ile
EFRD over tree height adapted from Brum *et al.* (2019). Please also see Supplementary Video 1 for a visualization of root growth
and    development    of    belowground    carbon    pools    over    time    under    http://www.pik-
potsdam.de/~borissa/LPJmL4_VR/Supplementary_Video_1.pptx.

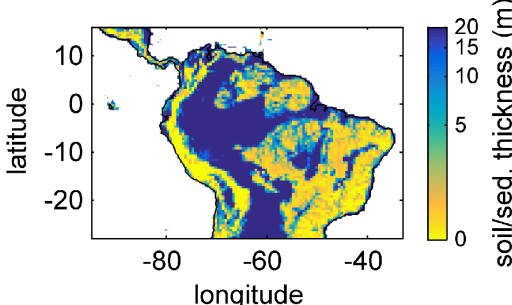


**Figure 4: Soil/sediment thickness from** (Pelletier et al., 2016) **regridded to the 0.5° x 0.5° longitude-latitude grid of LPJmL4.0-VR**
**and restricted to a maximum of 20 m. Colorbar in decadic logarithm.**

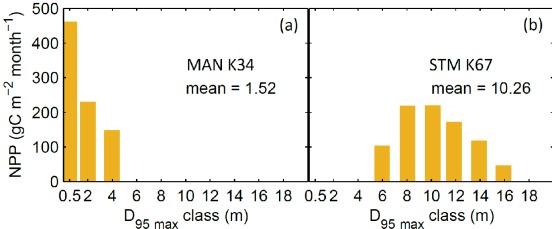


**Figure 5: Distributions of simulated mean monthly NPP for each $D_{95\_max}$-class for 2001-2010 under CRU climate input at two**
**FluxNet sites. a) Site MAN K34 near the city of Manaus. b) Site STM K67 near the city of Santarem. For more site information see**
**table 3 and figure 9a).**

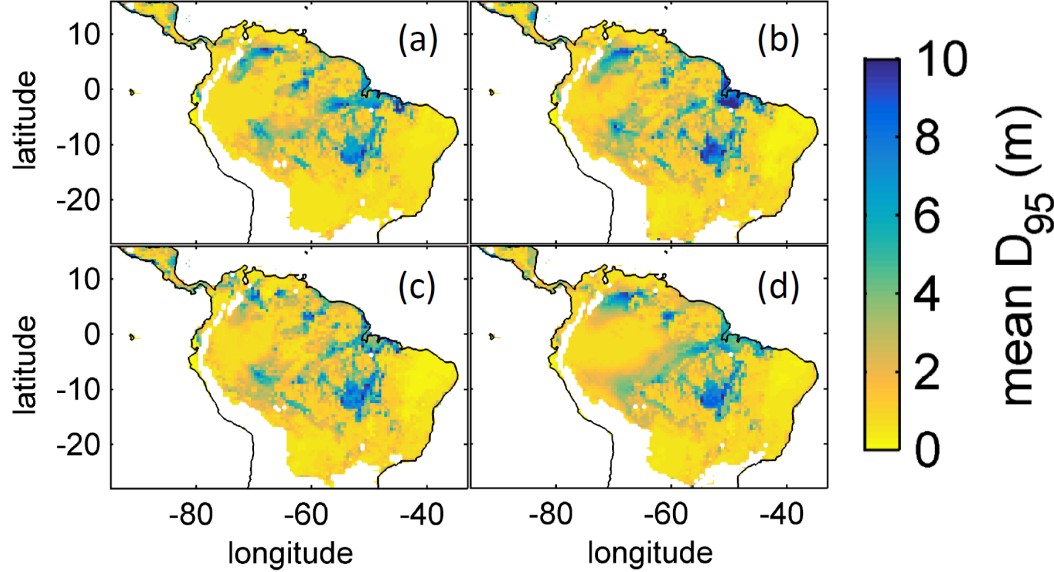


**Figure 6: Regional NPP-weighted mean rooting depth ($\overline{D_{95}}$) of all sub-PFTs (evergreen and deciduous combined) for 2001-2010**
**and different climate inputs simulated with LPJmL4.0-VR. a) CRU climate input. b) GSWP3 climate input. c) WATCH+WFDEI**
**climate input. d) GLDAS climate input. The color scale maximum is set to 10 m.**
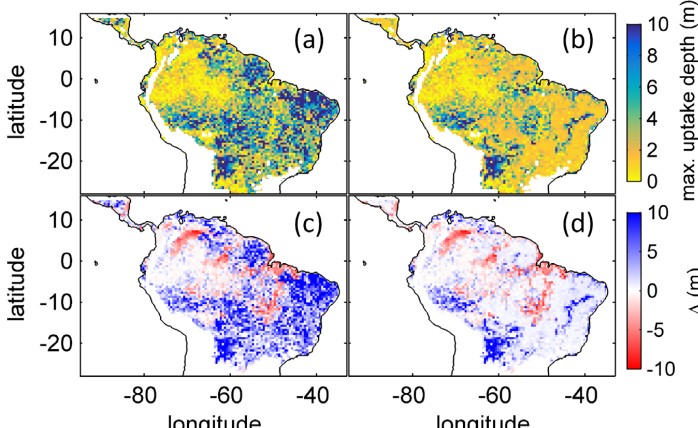

**Figure 7: Comparison of simulated $\overline{D_{95}}$ to product of maximum tree root water uptake depth (MDRU). a) Original** (Fan et al., 2017) **MDRU regridded to 0.5°x0.5° resolution of LPJmL4.0-VR. b) Same as a) but adjusted to soil depth input used in this study (see 2.3.2), in cases where values of** (Fan et al., 2017) **exceeded this soil depth. The color scale maximum for a) and b) is set to 10 m. c) Difference between a) and $\overline{D_{95}}$ simulated with LPJmL4.0-VR under CRU climate forcing (Fig. 6a). d) Difference between b) and $\overline{D_{95}}$ simulated with LPJmL4.0-VR under CRU climate forcing (Fig. 6a). Red/blue colors denote higher/lower rooting depths in LPJmL4.0-VR.**

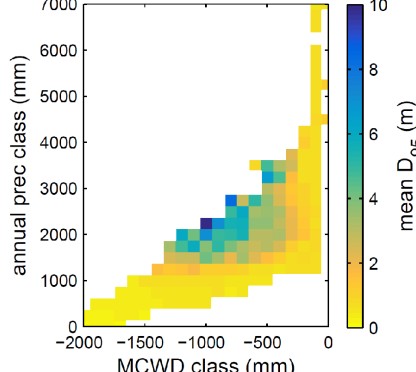

**Figure 8: Mean rooting depth depicted as mean $\overline{D_{95}}$ over classes of MCWD and annual precipitation sums. Class step size for precipitation was set to 250 mm and class size for MCWD was set to 50 mm. Regions with high amounts of annual rainfall and lower seasonality exclusively favour shallow rooted forests (low $\overline{D_{95}}$). $\overline{D_{95}}$ increases with decreasing MCWD (increasing seasonal drought stress) and decreasing sums of annual precipitation. Below 1200 mm of annual rainfall or -1100 mm of MCWD $\overline{D_{95}}$ sharply decreases again. Note this figure does not consider soil depth. The color scale maximum is set to 10 m.**





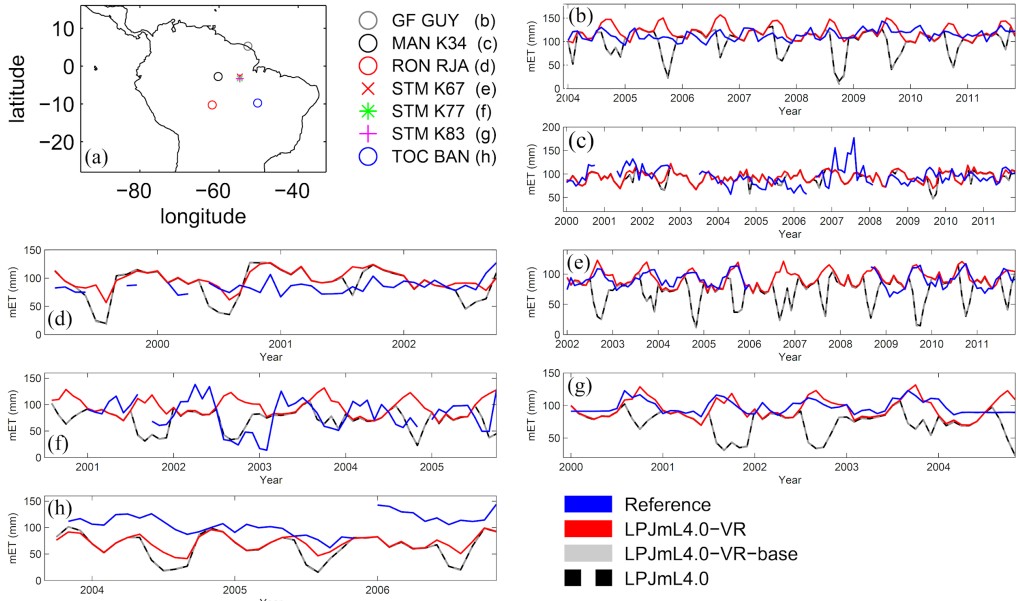

**Figure 9: Comparisons of monthly ET between different Fluxnet sites and respective simulation output of the different LPJmL model versions used in this study forced with CRU climate. a) Geographical location of different Fluxnet sites (see also table 3). For statistical measures of the individual comparison see Table 4.**

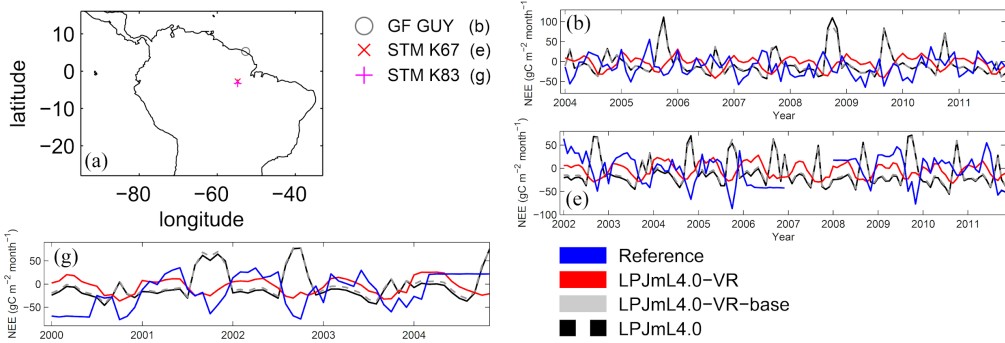

**Figure 10: Comparisons of monthly NEE between different Fluxnet sites and respective simulation output of the different LPJmL model versions used in this study forced with CRU climate. a) Geographical location of different Fluxnet sites (see also table 3). For statistical measures of the individual comparison see table 5. Note due to data scarcity only 3 Fluxnet sites are shown. Plots of all sites are shown in Fig. S3. We kept panel labelling as in Fig. 9 to ensure easy comparability.**



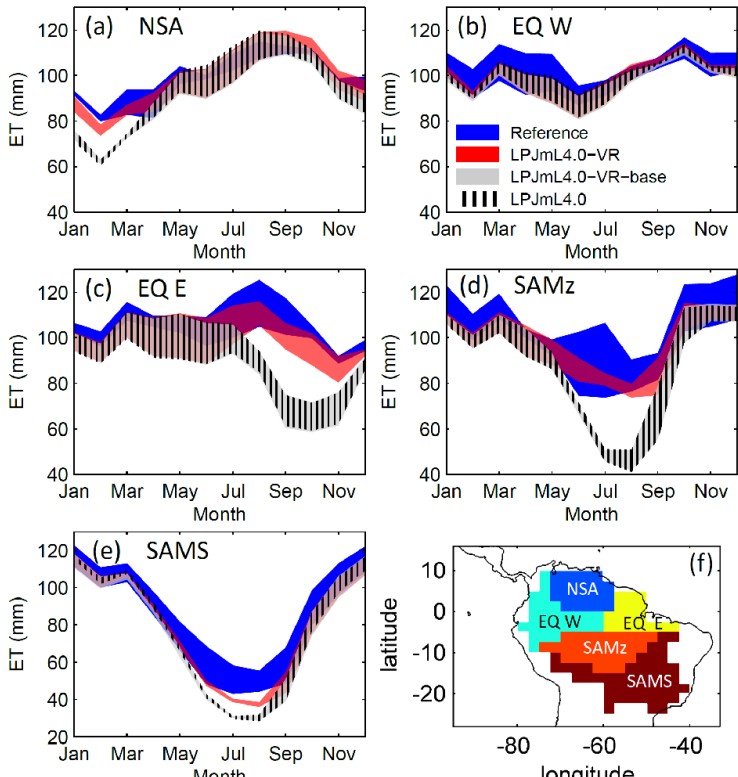

**Figure 11:** Comparisons of continental scale gridded ET products (2.4.2) against simulated ET within 5 regional climatological clusters (a-e) as defined in 2.4.2). Shown is the mean annual cycle of 1981-2010 and the mean for the whole cluster area. Corridors denote the minimum-maximum range between either the ET products or the model outputs under the different climate forcings used in this study. f) Geographical extent of climatological clusters (adapted from Sörensson and Ruscica, 2018). Statistical measures of the individual comparisons can be found in Table 6 (comparisons of corridor means).
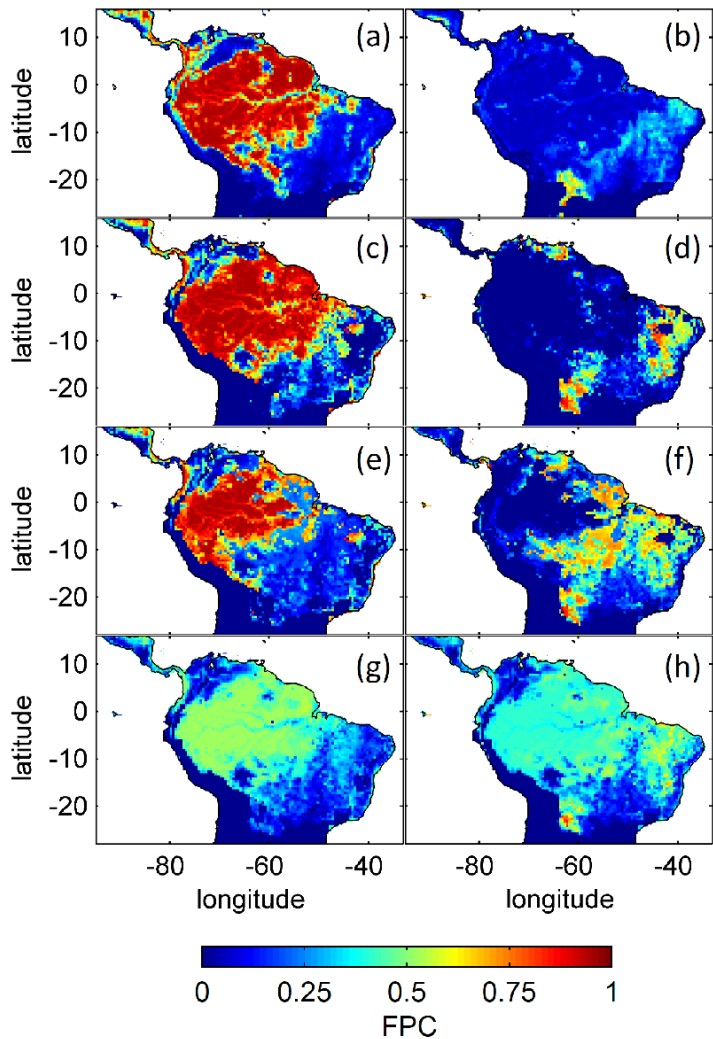

1090

**Fig. 12: Foliage projected cover (FPC) of evergreen (a, c, e ,g) and deciduous (b, d, f, h) PFTs over the study region. a)-b) Satellite-derived vegetation composition from ESA Land cover CCI V2.0.7** (Li et al., 2018) **reclassified to the PFTs of LPJmL as in** (Forkel et al., 2014)**. b)-c) LPJmL4.0-VR. d)-e) LPJmL4.0-VR-base. f)-g) LPJmL4.0. All LPJmL model versions were forced with CRU climate input. The shown FPC for all models refers to 2001-2010. For statistical measures of individual comparisons between model versions (c-h) and satellite derived vegetation composition (a-b) see Table 7.**

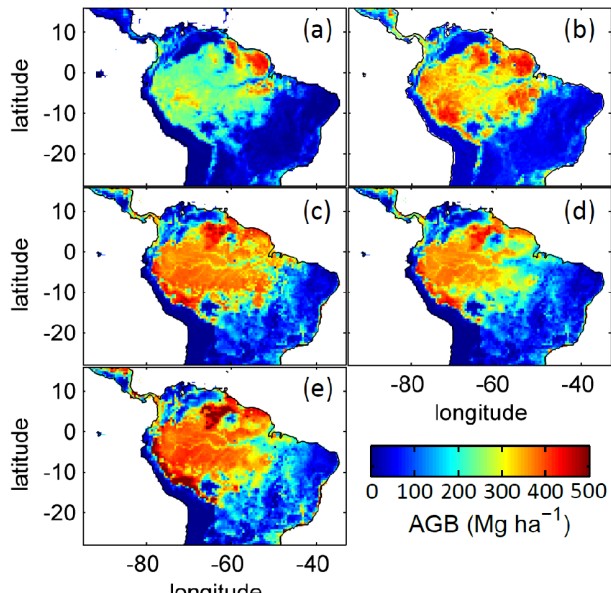

1096

**Fig. 13: Comparison of simulated AGB and satellite derived AGB validation products regridded to the spatial resolution of LPJmL models. a) Biomass validation product from Avitabile *et al.* (2016b). b) AGB validation product from Saatchi *et al.*, (2011). c)-e) Mean AGB simulated for the time span 2001-2010 with c) LPJmL4.0-VR. d) LPJmL4.0-VR-base and e) LPJmL4.0. For statistical measures of individual comparisons between model versions (c-e) and satellite derived AGB evaluation products (a-b) see Table 8.**

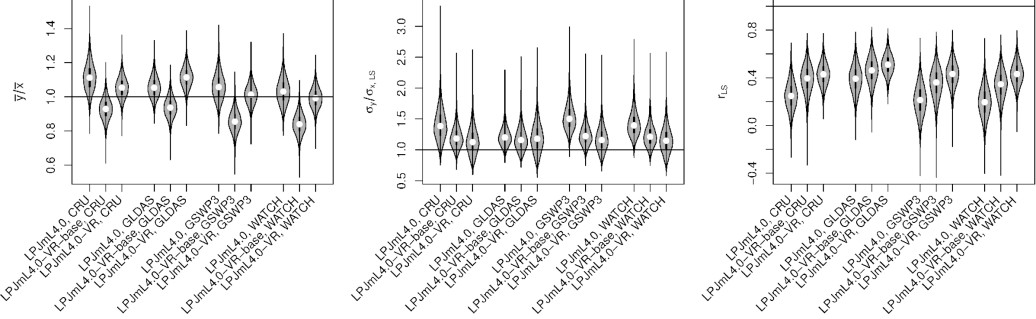

1102

**Fig. 14: Comparison of simulated large-scale average AGB ($Y$) from LPJmL4.0, LPJmL4.0-VR-base and LPJmL4.0-VR for different climate datasets to forest inventory data ($X$) from Brienen et al. (2015) using the method from Rammig et al. (2019). Three metrics are shown: the ratio of means ($\overline{y}/\overline{x}$) as a measure for the agreement of pattern average (left), the ratio of standard deviations of large scale AGB patterns ($\sigma_y/\sigma_{x,LS}$) as a measure for the agreement of pattern amplitude (middle), the corrected Pearson correlation coefficient ($r_{LS}$) as a measure for the agreement of pattern shape (right).**



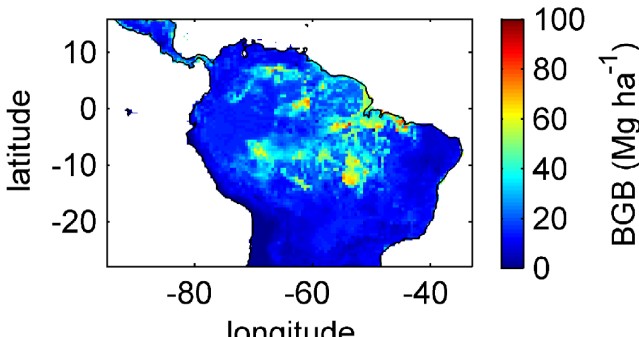


**Fig. 15: Mean sum (2001-2010) of belowground biomass (BGB; sum of tree coarse and fine roots) of evergreen and deciduous tree PFTs simulated with LPJmL4.0-VR under CRU climate forcing.**

| Soil layer number | Soil layer boundary (m) | Soil layer thickness (m) |
|---|---|---|
| 1 | 0.2 | 0.2 |
| 2 | 0.5 | 0.3 |
| 3 | 1 | 0.5 |
| 4 | 2 | 1 |
| … | … | … |
| 23 | 20 | 1 |


**Table 1: Soil layer partitioning scheme used in LPJmL4.0-VR. The first meter of the soil column is split into 3 soil layers and after 1m of soil depth each following soil layer is assigned a thickness of 1 m as in LPJmL4.0. Whereas LPJmL4.0's last soil layer reaches 3 m, LPJmL4.0-VR's last soil layer reaches 20 m.**

| sub-PFT number | $\beta$-value | $D_{95\_max}$ (m) |
|---|---|---|
| 1 | 0.9418 | 0.5 |
| 2 | 0.9851 | 2 |
| 3 | 0.9925 | 4 |
| 4 | 0.995 | 6 |
| 5 | 0.9963 | 8 |
| 6 | 0.9971 | 10 |
| 7 | 0.9976 | 12 |
| 8 | 0.9981 | 14 |
| 9 | 0.9986 | 16 |
| 10 | 0.9993 | 18 |


**Table 2: $\beta$-values assigned to the 10 sub-PFTs of each tropical PFT (evergreen and deciduous) in LPJmL4.0-VR and the corresponding maximum rooting depth reached by 95% of the roots ($D_{95\_max}$).**

| Site name | Short name | Country | LPJmL coordinate latitude | longitude |
|---|---|---|---|---|
| Ecotone Bananal Island/BR-Ban | TOC_BAN | Brazil | -9.75 | -50.25 |
| Manaus-ZF2 K34/BR-Ma2 | MAN_K34 | Brazil | -2.75 | -60.25 |
| Santarem-Km67-Primary Forest/BR-Sa1 | STM_K67 | Brazil | -2.75 | -54.75 |



| | | | | | |
|---|---|---|---|---|---|
| Santarem-Km77-Pasture/BR-Sa2 | STM_K77 | Brazil | -3.25 | -54.75 | |
| Santarem-Km83-Logged Forest/BR-Sa3 | STM_K83 | Brazil | -3.25 | -54.75 | |
| Rond.- Rebio Jaru Ji Parana-Tower B/BR-Ji3 | RON_RJA | Brazil | -10.25 | -61.75 | |
| Guyaflux | GF_GUY | French Guiana | 5.25 | -52.75 | |


**Table 3: Description of Fluxnet sites used for the evaluation of simulated ET.**

| Statistic | Model | TOC_BAN | MAN_K34 | STM_K67 | STM_K77 | STM_K83 | RON_RJA | GF_GUY |
|---|---|---|---|---|---|---|---|---|
| NME | LPJmL4.0-VR | 2.41 | 1.11 | 0.75 | 1.38 | 1.10 | 2.28 | 1.57 |
| | LPJmL4.0-VR-base | 2.92 | 1.22 | 2.29 | 0.98 | 2.74 | 2.73 | 2.38 |
| | LPJmL4.0 | 2.93 | 1.23 | 2.27 | 0.98 | 2.74 | 2.70 | 2.36 |
| $r^2$ | LPJmL4.0-VR | 0.09 | 0.03 | 0.53 | 0.17 | 0.43 | 0.01 | 0.08 |
| | LPJmL4.0-VR-base | 0.10 | 0.00 | 0.33 | 0.14 | 0.03 | 0.01 | 0.01 |
| | LPJmL4.0 | 0.09 | 0.00 | 0.33 | 0.14 | 0.03 | 0.01 | 0.01 |
| p-value | LPJmL4.0-VR | 0.075 | 0.041 | < 0.001 | 0.002 | < 0.001 | 0.575 | 0.005 |
| | LPJmL4.0-VR-base | 0.067 | 0.585 | < 0.001 | 0.005 | 0.221 | 0.517 | 0.277 |
| | LPJmL4.0 | 0.068 | 0.672 | < 0.001 | 0.005 | 0.221 | 0.514 | 0.274 |


**Table 4: Normalized mean error (NME), coefficient of determination ($r^2$) and p-value of F-statistic piecewise calculated for simulated ET of the different LPJmL model versions used in this study forced with CRU climate input and Fluxnet data of ET at 7 Fluxnet sites (in accordance with Fig. 8).**

| Statistic | Model | STM_K67 | STM_K83 | GF_GUY |
|---|---|---|---|---|
| NME | LPJmL4.0-VR | 0.90 | 0.84 | 1.30 |
| | LPJmL4.0-VR-base | 1.62 | 1.36 | 1.52 |
| | LPJmL4.0 | 1.68 | 1.39 | 1.52 |
| $r^2$ | LPJmL4.0-VR | 0.16 | 0.14 | 0.00 |
| | LPJmL4.0-VR-base | 0.32 | 0.06 | 0.03 |
| | LPJmL4.0 | 0.33 | 0.07 | 0.03 |
| p-value | LPJmL4.0-VR | < 0.001 | 0.003 | 0.515 |
| | LPJmL4.0-VR-base | < 0.001 | 0.055 | 0.046 |
| | LPJmL4.0 | < 0.001 | 0.047 | 0.059 |


**Table 5: Normalized mean error (NME), coefficient of determination ($r^2$) and p-value of F-statistic piecewise calculated for simulated NEE of the different LPJmL model versions used in this study forced with CRU climate input and Fluxnet data of NEE at 3 Fluxnet sites (in accordance with Fig. 10).**

| Statistic | Model | NSA | EQ W | EQ E | SAmz | SAMS |
|---|---|---|---|---|---|---|
| NME | LPJmL4.0-VR | 0.08 | 0.26 | 0.62 | 0.20 | 0.06 |
| | LPJmL4.0-VR-base | 0.37 | 0.42 | 1.95 | 0.58 | 0.13 |
| | LPJmL4.0 | 0.34 | 0.26 | 1.92 | 0.58 | 0.11 |
| $r^2$ | LPJmL4.0-VR | 0.98 | 0.94 | 0.91 | 0.98 | 1.00 |
| | LPJmL4.0-VR-base | 0.94 | 0.96 | 0.20 | 0.91 | 0.99 |
| | LPJmL4.0 | 0.93 | 0.96 | 0.21 | 0.90 | 0.99 |
| p-value | LPJmL4.0-VR | < 0.001 | < 0.001 | < 0.001 | < 0.001 | < 0.001 |
| | LPJmL4.0-VR-base | < 0.001 | < 0.001 | 0.143 | < 0.001 | < 0.001 |
| | LPJmL4.0 | < 0.001 | < 0.001 | 0.135 | < 0.001 | < 0.001 |


**Table 6: Normalized mean error (NME), coefficient of determination ($r^2$) and p-value of F-statistic piecewise calculated for the simulated ET of the different LPJmL model versions used in this study and continental scale gridded ET products within 5 regional climatological clusters. With respect to Fig. 11 comparisons are based on the monthly mean of corridors shown, i.e. 1) the**





monthly mean of all outputs produced by one LPJmL model version but forced with different climate inputs and 2) the monthly
mean of all continental scale gridded ET data products.

| Statistic | Model | FPC Evergreen | FPC Deciduous |
|---|---|---|---|
| NME | LPJmL4.0-VR | 0.31 | 1.01 |
| | LPJmL4.0-VR-base | 0.38 | 1.5 |
| | LPJmL4.0 | 0.47 | 1.76 |


**Table 7: Normalized mean error (NME) of FPC comparison piecewise calculated between 1) the satellite-derived vegetation**
**composition from ESA Land cover CCI V2.0.7** (Li et al., 2018) **reclassified to the PFTs of LPJmL as in Forkel et al. (2014) and 2)**
**all LPJmL model versions used in this study forced with CRU climate data (in accordance with Fig. 10).**

| Statistic | Model | Avitabile *et al.* | Saatchi *et al.* |
|---|---|---|---|
| NME | LPJmL4.0-VR | 0.78 | 0.12 |
| | LPJmL4.0-VR-base | 0.69 | 0.11 |
| | LPJmL4.0 | 1.09 | 0.14 |


**Table 8: Normalized mean error (NME) of AGB comparison piecewise calculated between 1) the satellite-derived AGB validation**
**products and 2) all LPJmL model versions used in this study forced with CRU climate data (in accordance with Fig. 12).**