# Peer review of "Variable tree rooting strategies improve tropical productivity and 1 evapotranspiration in a dynamic global vegetation model 2"

_Biogeosciences, 2020_

## Referee Comment (RC1) · Anonymous Referee #1 · 28 Apr 2020

In this manuscript, the authors extended the LPJmL4.0 dynamic vegetation model to simulate variable rooting depth. Comparisons between different model versions for tropical South America showed that the novel approach improves most of the benchmarks used in the study. Overall, the study is well-conducted and solid. I agree with the authors that including variable rooting depth is important in DGVMs and the results and the presented model approach are therefore relevant for the modeling community. Previous models typically assumed PFT specific fixed rooting depth and/or a fixed soil depth globally.

My main point is, however, that the manuscript reads like a collection of many interest-

ing results while I was missing more specific aims or questions regarding for example implications of root diversity for the ecology or biogeography of the simulated vegetation types. Currently, the aims of the study are to 'describe an approach' and to 'evaluate its effect on' various model variables.

Specific comments:

L 25: PFT-specific instead of biome-scale?

L 29: realistically simulated

L 41: delete quotation marks for evergreen and deciduous?

L 47-48: Isn't it the other way round? Traits are aggregated to define PFTs and the fractional cover of different PFTs in simulations then defines the biome type?

L 54: reword 'different attempts were carried out', maybe 'different approaches were presented'?

L 54: Schymanski et al. also developed root models (e.g. www.hydrol-earth-syst-sci.net/12/913/2008/). Although these models were not developed explicitly in the DGVM context, they might be relevant for the introduction or the discussion.

L 55-56: 'study ... searched for...' reword, I would think that a study can's search.

L 74: 'allocation strategy' instead of 'direction'?

L 97: in competition with other rooting strategies/plants/PFTs?

L 100: this suggests to me that rooting depth is related to or a function of tree height.

L 105-108: I think that such a general overview is not necessary and could be removed (also elsewhere in the methods section). Further, it only refers to some selected section and it is incomplete.

L 115: simulates instead of employs?

L 116: Should it be 'bioenergy functional type'?

L 122: 'how different tree rooting strategies (implemented in the new scheme) compete'?

L 131: Does that mean that soil texture is similar for all layers in the soil column?

L 142: PFT scale instead of biome scale?

L 144 and elsewhere: is it really the depth which is reached by 95% of the roots or does it rather mean that 95% of the fine root biomass is between the surface and D95_max? Wouldn't 'reached by 95%' mean that 95% are deeper than D95_max?

L 146: that's a question related to LPJml4.0 and not to the VR version: why were different values for evergreen and deciduous selected if the resulting root profiles are essentially identical. Would model results differ for similar values?

L 149: why 18m and not 20m or 95% of 20m?

L 154: a new carbon allocation scheme?

L 163: 'sapwood ... proportional' which constant was used to describe this relation? And shouldn't root and stem sapwood are be identical to be able to transport the same amount of water?

L 180: 'to derive a functional relation between tree height and rooting depth'?

L 203: check brackets in Huang et al

L 233: I suggest to delete 'The new features... direction.' and say 'In LPJmL4.0-VR, PFTs can...'

L 234: 'formally', I assume this should be 'formerly' or 'in previous model versions'.

L 233-240: An increase of mortality rates from 3% to 7% is quite substantial and more then twice as high. In addition, it is stated that observed rates do not exceed 6% which means that 7% is not in the real world boundaries as stated in L 237 but it overestimates

this rate by >16%.

L 239: 'We regard ...' I don't understand this sentence. And what does 'right direction' refer to? I suggest to reword.

L 243: Why were four different climate inputs used? Most simulation results are shown only for CRU anyway. Which climate variables were taken from the different datasets?

L 267: Reword to 'caused by the the presence or absence of variable rooting strategies'?

L 269: Given that model spin-up was conducted for different time periods for different climate input datasets and at 278ppm. Is there a jump in $CO_2$ in the transition between spin-up and transient phase?

L 280: I understood that replicate simulations were not conducted. I was wondering how robust or deterministic the selection of rooting strategies is? Would you expect substantial differences in the results when conducting replicate model runs?

L 285: which method was used for re-gridding?

L 289: check brackets in Brienen

L 287: I found it difficult to understand the description of the Rammig et al method and I had to go back to the original paper. I suggest to check the paragraph again for clarity.

L 304: check bar in 'average x_bar'

L 311: I am skeptical when using gridded climate products to simulate local scale EC fluxes, because these products might not capture some local rainfall events (for example) that have strong impacts on the fluxes. Hence, models will fail to simulate the fluxes. I assume that there are there local scale meteorological data available for the flux sites that could be used for running the model or at least for comparing agreement between gridded data products and observation at EC sites.

L 313: I suggest to state why NEE was only simulated for 3 sites, this information is currently hidden in the figure caption.

L 336: replace 'called' by ':'

L 376: replace 'over' by 'instead of'?

Generally the results section contains some statements or explanations that do not only describe the results but already go beyond and might be more more appropriate for the discussion.

L 387, Fig 5: when looking at this figure, I was wondering if simulated distributions are always unimodal or if there the model can also simulate bi-modal or multi-modal distributions indicating that very distinct rooting strategies can coexist? I also suggest to add to the figure caption which site is wetter and which site is drier.

L 406: Fig 11 (not 9f?)

L 434: why 4m? Is there a reference for this value?

L 445: 'behavior: Whereas..' Full stop or small w in whereas.

L 453: what exactly does 'reversely' mean?

L 377: The text in this paragraph and the figures are mainly about PFTs, not biomes.

L 519: 'uncertainty...is' or 'uncertainties...are'

L 541: I agree that it's important to look at below ground biomass but comparing Fig 15 and Fig 13 suggest that the ratio between aboveground and below ground biomass is extreme in some areas with high aboveground biomass but low below ground biomass (300-400t/ha aboveground vs ca 20t/ha belowground). Are such ratios realistic in these regions and how can this be explained?

L 557: according to figure caption in Fig 12, these are PFTs not biomes.

L 565: 'where' instead of 'were'
I was surprised that grasses and fire were only shortly mentioned in the discussion, given that the study region also includes seasonal areas with Cerrados and not only evergreen forests. How are these systems represented? Only by deciduous forest or does the model also simulate a grassy component and fire? Fire also has some impacts on biomass in these regions and it has been argued that lateritic layers constrain rooting depth and might thereby influence grass-tree coexistence in these regions.

L 606: the extent of evergreen forest has not been presented, but rather the extent of the evergreen PFT and the deciduous PFT. Further, Fig 12 shows that the extent of the evergreen PFT is very similar in the original and the VR version (although the FPC is much lower in the original version). I suggest to clarify or to classify vegetation into biomes based on the FPC of different PFTs. This would allow comparisons of biome cover in different scenarios.

L 627: 'Expansion' instead of 'Extent'?

L 638: PFT instead of biome types?

In the discussion I was missing some discussion of the results in the context of previous modeling studies, such as the studies cited in the introduction.

As the study region als includes the Cerrados, the rooting niche separation ideas that explain grass-tree coexistence in savannas might be relevant for the discussion, e.g. Van Langevelde et al 2013 Ecology.

L 1083: Figure S3 not provided.

Just out of interest, can the model easily be adapted to global scale, and will these model developments be included in the global 'default' version of LPJmL4.0? Or would this lead to computational constraints?

---

## Referee Comment (RC2) · Daniel Falster (Referee) · 23 May 2020

The article by Sakschewski et al introduces a scheme for modelling variable rooting depths into the LPJmL4.0 DGVM. This is the version of LPJ dealing with managed lands (ML). Otherwise, it shares much with the original LPJ (as opposed to the individual based and gap variants). I agree the problem identified is potentially very significant: previously this and other DGVMS have used a fixed rooting depth, which may limit the ability of modelled vegetation to adjust to different water environments. The new scheme is assessed by comparing model outputs (including NPP, max rooting depth, ET, NEE, and cover) to various datasets and products for South America, mostly

at large (regional or larger) spatial scales. The new model is compared to two other versions of LPJ - the base version and the new model but with variable rooting effectively disabled. The authors claim that overall the variable rooting scheme improves the models ability to capture observed dynamics and distributions in the different datasets.

I like many aspects of this paper. In particular I appreciate the approach of carefully examining the behaviour of the model across a range of climate scenarios,

Overall I think this paper and model variant have great potential. The new scheme is clever and I have little to critique in its design. The various analyses do suggest the new model is providing a better match to available data. But I am yet to be convinced this improved match is for the reasons the authors claim (better handling of soil water & rooting).

The biggest concern I have is that despite all the results presented, these mostly for aggregate outputs at largish spatial scales. Almost no evidence is presented to show the effect of the new scheme on the actual water balance in the soil. Moreover, we don't even know how water is modelled in the different versions. Sure, the soil and root depth is changed, but what does this mean for water balance at different soil depths? Surely this is key to assessing why the model behaves differently. This is important, as the changes in rooting will also change the way carbon is allocated within the model (e.g. deeper roots divert carbon away from leaves). Are the model improvements due to changes in hydraulics, or changes in carbon allocated between leaves and roots?

Second, I feel the authors need to come up with a stronger story and reduced set of results. The paper is currently very long and dense. The authors have made many comparisons using a variety of datasets. Consequently, there is a large number of figures (15) and tables (8). This makes it hard for us to know where to put our attention.

Finally, more work is needed to make the different results accessible and easy to interpret. I found that each figure required a fair bit of work to interpret what is going on. Some simple changes could make it much easier for the reader, then we could

spend less time deciphering your results and more time thinking about the science! As examples,

- In Fig 1: confusing caption. Simplify labels in legend.

- In Fig 6: label panels with dataset name, so that we don't need to refer to legend as much

- In Fig 9, uses different colours in the map and traces, otherwise these are easily confused. Label each subplot with site name.

- In Fig 12, put labels on the columns (evergreen, deciduous) and rows (models), so that we can easily see what the different panels are without constantly referring to the caption.

Some minor issues

- Some line breaks between paragraphs would make the text much easier to read

- Eq 8: I've looked over this a few times and wonder if the $n\_est\_tree$ at the end should be removed?

- ln 413 – It didn't make much sense to me to compare your results to a modelled product of rooting depth.

- I found the talk of offspring, saplings, and "growth" throughout the paper a bit misleading. My understanding of this version of LPJ is that each patch has a single functional type which has a density 'n' of average sized individuals. When new offspring are recruited, they don't grow from seed to adult, but rather enter fully formed at the average size (this occurs by increasing n, the number of individuals). The only time the individuals seemingly grow from small to large plants, is when starting from bare soil, i.e. during spin up. Yet, often the paper gave the impression that individuals could be born and grow.

---

## Author Comment (AC1) · 3 Jul 2020

**Response to the comments of referee 1 on Sakschewski et al. 2020: Variable tree rooting strategies improve tropical productivity and evapotranspiration in a dynamic global vegetation model**

In this manuscript, the authors extended the LPJmL4.0 dynamic vegetation model to simulate variable rooting depth. Comparisons between different model versions for tropical South America showed that the novel approach improves most of the bench-marks used in the study. Overall, the study is well-conducted and solid. I agree with the authors that including variable rooting depth is important in DGVMs and the results and the presented model approach are therefore relevant for the modeling community. Previous models typically assumed PFT specific fixed rooting depth and/or a fixed soil depth globally. My main point is, however, that the manuscript reads like a collection of many interesting results while I was missing more specific aims or questions regarding for example implications of root diversity for the ecology or biogeography of the simulated vegetation types. Currently, the aims of the study are to 'describe an approach' and to' evaluate its effect on' various model variables.

Dear referee,

Thank you for your positive evaluation and many constructive comments on our manuscript. We agree that the paper benefits from focusing on a specific research question which creates a stronger storyline. As we will now highlighted in the introduction in line XX we now focus on the research question "What is the role of diverse tree rooting strategies for productivity and evapotranspiration in tropical South America?" and structure the manuscript accordingly. In connection to comments of referee 2 we will shorten the manuscript and move parts of the results to the supplement. This also helps to create a stronger and clearer storyline.

L 25: PFT-specific instead of biome-scale?
Thank you for this comment. In the abstract we wanted to avoid specific terms like "PFT" to keep it easy to understand for readers which are not familiar with those terms, but still convey the main message. Therefore we now changed the phrase into "often condense this variety of tree rooting strategies into averages of tree growth startegies from the ecosystem- to biome-scale" in line XX.

L 29: realistically simulated
Thank you for this suggestion. We changed it accordingly, now in line XX.

L 41: delete quotation marks for evergreen and deciduous?
Thank you for this suggestion. We changed it accordingly, now in line XX.

L 47-48: Isn't it the other way round? Traits are aggregated to define PFTs and the fractional cover of different PFTs in simulations then defines the biome type?
Thank you for raising this point. We apologize for the sloppy usage of "biome" and "PFT". To clarify the text we now state "In general these models condense the diversity of such functional tree traits into so called plant functional types (PFTs) which represent average tree individual on scales as large as biomes" in line XX.

L 54: reword 'different attempts were carried out', maybe 'different approaches were presented'?
Thank you for this suggestion. We changed it accordingly, now in line XX.

L 54: Schymanski et al. also developed root models (e.g. www.hydrol-earth-syst-sci.net/12/913/2008/). Although these models were not developed explicitly in theDGVM context, they might be relevant for the introduction or the discussion.
Thank you for this suggestion. We will incorporate this important study in the suggested sections.

L 55-56: 'study ... searched for...' reword, I would think that a study can's search.
Thank you for this correction. We changed the sentence to "In a pioneering study more than 20 years ago, Kleidon and Heimann (1998) systematically searched for rooting strategies which yield highest net primary productivity ..." in line XX.

L 74: 'allocation strategy' instead of 'direction'?
Thank you for this suggestion. We changed it accordingly, now in line XX.

L 97: in competition with other rooting strategies/plants/PFTs?
Thank you for this question. We changed the respective sentence to "all tree rooting strategies of this spectrum grow in competition..." in line XX as it is meant to contrast this study with some of the former approaches. A more detailed sentence would lead to a full description of the competition scheme which can be found in the methods now in line XX formerly in line 208-221.

L 100: this suggests to me that rooting depth is related to or a function of tree height.
Thank you for this comment. Yes as we clarified in the methods we used this approach. Formerly in line 168-206. Now line XX.

L 105-108: I think that such a general overview is not necessary and could be removed (also elsewhere in the methods section). Further, it only refers to some selected section and it is incomplete.
We agree with the referee and deleted this part.

L 115: simulates instead of employs?
Thank you for this remark. We changed it accordingly, now in line XX.

L 116: Should it be 'bioenergy functional type'?
Thank you for pointing out this mistake. We inserted "functional" now in line XX.

L 122: 'how different tree rooting strategies (implemented in the new scheme) compete'?
Thank you for this suggestion. We implemented it, now in line XX.

L 131: Does that mean that soil texture is similar for all layers in the soil column?
Thank you for this question. Yes. In LPJmL4.0 each grid cell has one soil texture information for its 3 m soil column only. We followed this approach for our larger soil columns as well. In fact a high resolution soil texture information in 3 dimensions for the study region is so far not available, or only partly available. We wanted to keep things simple and comprehensible also with regard to comparing results of LPJmL4.0-VR to LPJmL4.0. We now better clarify the approach in line XX stating "Equal to LPJmL4.0, we apply a single grid cell specific soil texture information to the whole soil column as in (Schaphoff et al., 2018)".

L 142: PFT scale instead of biome scale?
Thank you for this suggestion. We changed it as suggested, now line XX.

L 144 and elsewhere: is it really the depth which is reached by 95% of the roots or does it rather mean that 95% of the fine root biomass is between the surface and D95_max? Wouldn't 'reached by 95%' mean that 95% are deeper than D95_max?

Thank you very much for pointing out that out. We changed the sentence to "…we here calculate the depth at which the cumulated fine root biomass from the soil surface downwards makes up 95% of all fine root biomass", now in line XX.

L 146: that's a question related to LPJml4.0 and not to the VR version: why were different values for evergreen and deciduous selected if the resulting root profiles are essentially identical. Would model results differ for similar values?
Thank you for this question. The parameter values are average values for those 2 PFTs found in Jackson et al. (1996). At model development of LPJmL those values were the best available. We now specifically mention where those parameter values came from in line XX.

For now it is clear that increasing the difference between the beta values of the 2 PFTs would most probably enhance model performance. Even with only 2 PFTs present. Here the best way to go would probably be to increase the beta value of the evergreen PFT to reach a D95_max of maybe 2-4 m, which would buffer against some dry season signal. Regarding the second question: If regarding beta values only, the answer is no, there would be no difference between the evergreen and deciduous PFT. Fortunately, they also differ in other functional traits such as specific leaf are (SLA) or leaf longevity (LL), determining their phenological strategy and therefore their performance under different climate regimes.

L 149: why 18m and not 20m or 95% of 20m?
Thank you very much for this question. As we have chosen a maximum soil depth of 20m we wanted to avoid a significant accumulation of fine roots in the last soil layers. A D95_max of 20m in a soil column of 20 m would mean that the additional 5% of fine roots are also distributed between 19-20m. In principle it was a choice between a round number of soil depth or a round number at the largest value of D95_max. In the current model version of LPJmL4.0-VR a D95_max of 20m is not missing in the study area as distributions of D95_max always flatten towards the 18m bin in grid cells where deep roots dominate. Nevertheless, it is possible to substantially increase soil depth and create more sub-PFTs with larger D95_max values. This step might even be necessary in future versions of the model or other study areas. We now explain why we chose the largest D95_max at 18m in line XX stating "We chose 18m as the largest D95_max value in order to avoid that roots of the respective sub-PFT significantly exceed the maximum soil depth of 20m (see also 2.2.4 and Fig. 2 right panel)."

L 154: a new carbon allocation scheme?
Thank you for this suggestion. We changed it accordingly, now in line XX.

L 163: 'sapwood ... proportional' which constant was used to describe this relation? And shouldn't root and stem sapwood are be identical to be able to transport the same amount of water?
Thank you for pointing this out. Our explanation was misleading. We do not use a constant here. The term proportional is falsely used. It is as your second question suggests and as it is written in the description of Fig. 2. We now clarify the approach in the methods section in line XX with "Root sapwood cross-sectional area in the first soil layer is equal to stem sapwood cross-sectional area, as all water must be transported through the root sapwood within this soil layer. In the following soil layers downwards, root cross-sectional area decreases by the relative amount of fine roots in all soil layers above (Fig. 2)."

L 180: 'to derive a functional relation between tree height and rooting depth'?
Thank you for this suggestion. We changed it accordingly, now in line XX.

L 203: check brackets in Huang et al
Thank you for pointing this out. We changed the brackets, now in line XX.

L 233: I suggest to delete 'The new features... direction.' and say 'In LPJmL4.0-VR, PFTs can...'
Thank you for this suggestion. We changed it accordingly, now in line XX.

L 234: 'formally', I assume this should be 'formerly' or 'in previous model versions'.
Thank you very much. We changed it accordingly, now in line XX.

L 233-240: An increase of mortality rates from 3% to 7% is quite substantial and more then twice as high. In addition, it is stated that observed rates do not exceed 6% which means that 7% is not in the real world boundaries as stated in L 237 but it overestimates this rate by >16%.
Thank you very much for raising this point. We agree that claiming 7% is in the boundary of 6% is overstated. The point we wanted to make was mixed up and lost by citing the study mentioning a observed maximum of 6% background mortality rate in 167 Amazon forest plots (Johnson et al., 2016) and comparing it to the maximum (not actual background) mortality of LPJmL4.0-VR. In fact observed background mortality rate and conceivable maximum mortality rate are not the same thing. The growth efficiency related mortality which is used in many DGVMs needs a maximum mortality rate, because otherwise the simulated mortality rate would rise to 100% when total respiration exceeds productivity. In the real world though plants would optimize and reallocate carbon pools under those circumstances as far as they can. Nevertheless, a mortality rate of 100% in the real world is conceivable as well. The maximum mortality rate of a forest under a hypothetical super extreme scenario (no rain for 12 month) could of course be 100%.
Accordingly, in a hypothetical model which truly captures all important mechanisms of tree survival and mortality, the maximum mortality rate could be set close to 100% as we tried to explain formerly in line 239-240. In other words this constant could eventually be obsolete. The road to such a model will be long and might never be ending, but until then, a maximum mortality rate at any value will always be necessary to achieve biomass values in acceptable ranges. We now deleted the reference of observed mortality rates of 6% in the manuscript as this number is not comparable to the maximum mortality rate set in the model. Moreover, we will describe the true purpose of a maximum mortality rate and why we changed it (as described in this answer) now in line XX.

L 239: 'We regard ...' I don't understand this sentence. And what does 'right direction' refer to? I suggest to reword.
We deleted this sentence in accordance with our response above.

L 243: Why were four different climate inputs used? Most simulation results are shown only for CRU anyway. Which climate variables were taken from the different datasets?
Thank you very much for those questions. We wanted to show that our results remain robust using various climate inputs and decided (also with regard to the amount of figures and respective manuscript text) to show this with regional rooting depth (Fig. 6) and regional evapotranspiration (corridors in Fig. 11). In connection to referee 2 requesting a substantial reduction of the manuscript size and amount of figures, it seems impracticable to produce all figures for all climate inputs. We also argue that, Fig. 6 & 11 show some proof that results for different climate inputs are very similar. There are local differences, but the big picture remains the same. As we do not want to go into detail regarding local scale differences we hope the current amount of results is enough.
We will add a sentence of clarification in the discussion in line XX.
We now clarify what climate variables were taken from the different data sets in the respective methods section in line XX-YY.

L 267: Reword to 'caused by the the presence or absence of variable rooting strategies'?
Thank you very much for this suggestion. We changed it accordingly, now in line XX.

L 269: Given that model spin-up was conducted for different time periods for different climate input datasets and at 278ppm. Is there a jump in CO2 in the transition between spin-up and transient phase?

Thank you very much for this question. No there is a smooth transition between spin-ups and transient simulations regarding atmospheric CO2 content. Before the year 1840 a constant value of 278ppm is used, while after this year values are rising according to a LPJmL protocol first introduced in Sitch et al. (2003) and eventually following the Mona Loa record of Tans and Keeling (2015). We now clarify this approach in the methods section in line XX.

L 280: I understood that replicate simulations were not conducted. I was wondering how robust or deterministic the selection of rooting strategies is? Would you expect substantial differences in the results when conducting replicate model runs?

Thank you very much for this question. Given the vast amount of different model test runs during model development, we can assure that results are very robust. We admit we do not fully proof this in this study. However, the 4 very similar rooting depths maps (Fig. 6) provide some proof.
Since there are no real stochastic processes in LPJmL which could lead to a path dependency of vegetation dynamics, this model behavior was expected. Therefore, it is also a standard procedure to not conduct any replicate simulations when using LPJmL. We now clarify this point in the methods section in line XX.

L 285: which method was used for re-gridding?

Thank you for pointing out this lack of information. We now name the methods, software packages and underlying studies in line XX. Regridding was conducted in R (R Core Team, 2019) using the "aggregate" function of the R-package "raster" (Hijmans, 2019), which aggregates to lower resolutions by taking the arithmetic mean excluding NAs.

L 289: check brackets in Brienen

Thank you for pointing that out. We corrected the brackets, now in line XX.

L 287: I found it difficult to understand the description of the Rammig et al method and I had to go back to the original paper. I suggest to check the paragraph again for clarity.

We will provide a more sophisticated explanation of the respective methods section now in line XX-YY.

L 304: check bar in 'average x_bar'

Thank you for pointing out this mistake. We corrected the bar, now in line XX.

L 311: I am skeptical when using gridded climate products to simulate local scale EC fluxes, because these products might not capture some local rainfall events (for example) that have strong impacts on the fluxes. Hence, models will fail to simulate the fluxes. I assume that there are there local scale meteorological data available for the flux sites that could be used for running the model or at least for comparing agreement between gridded data products and observation at EC sites.

We fully agree that using gridded climate products is not optimal to reproduce locally measured ET fluxes, because they can lack information of local weather events. We solely use gridded data in this study due to several reasons: 1) We want to stay consistent with the regional results of this study. As we e.g. show a simulated regional rooting depth maps (Fig. 6) and plots of underlying local tree rooting strategies (Fig. 5) as well as regional ET (Fig. 11) all based on gridded climate data, changing the climate input for simulations at local scale seemed inconsistent. 2) Even though we apply statistical metrics to compare model vs. flux agreement, our focus was not on the effects of local

weather events on ET, rather on the general climate signal, most importantly the presence or absence of a dry season and the effects on simulated rooting depth and ET and the differences between the different model versions. We fully agree that forcing the model with local climate data could in principle enhance model performance, but we never aimed for a perfect match of ET and NEE at all sites. 3) There is meteorological data available for different flux sites, but these data sets are often cluttered with gaps and are only available for a few up to 10 years only. Moreover, each site has its own limitations when it comes to model implementation. Taken together, this creates quite some problems for DGVM simulations. The LPJmL model needs continues climate data and long time series. Jumping from a spin-up simulation into repeating 5-10 very similar years can cause artifacts which should be avoided. We would be happy simulate rainforest sites with real site meteorological data, but for this we would need longer time spans than currently available. There are approaches trying to solve those problems, but they are currently beyond the scope of this study. 4) Given that we compare monthly means of simulated and measured ET at the local scale and that simulation results appear to capture seasonal signals, we are convinced that our approach is sufficient to deliver the message of this manuscript.
We will insert a clarification of why we used gridded climate input data only in the discussion in line XX.

L 313: I suggest to state why NEE was only simulated for 3 sites, this information is currently hidden in the figure caption.

Thank you very much for this remark. Unfortunately, in the data sets accessible to us, continues NEE data covering at least 2 years was only available for 3 sites. We now clarify why we compare NEE only at 3 sites in the methods sections in line XX and in the respective figure caption.

L 336: replace 'called' by ':'

Thank you very much. We changed it accordingly, now line XX.

L 376: replace 'over' by 'instead of'?

Thank you for pointing that out. We changed it accordingly, now line XX.

Generally the results section contains some statements or explanations that do not only describe the results but already go beyond and might be more more appropriate for the discussion.

We will check the results section and will transfer all potential interpretations into the discussion.

L 387, Fig 5: when looking at this figure, I was wondering if simulated distributions are always unimodal or if there the model can also simulate bi-modal or multi-modal distributions indicating that very distinct rooting strategies can coexist? I also suggest to add to the figure caption which site is wetter and which site is drier.

Thank you for this comment. Indeed these distributions can be bimodal, indicating that very distinct rooting strategies can co-exist. We have not systematically checked all grid cells, but a clear tri-modal distribution was not observed so far. This might also need different ways of detecting them as multiple modes might be hidden in a continuous distribution. So far we observed 2 cases where distributions can clearly be bi-modal. 1) In areas with dominant evergreen tree cover and a "medium" dry season, where shallow and deep rooting evergreen sub-PFTs can co-exist. 2) Areas with a substantial dry season where the evergreen and deciduous PFT co-exist. Here the deciduous PFT shows shallower roots and the evergreen PFT deeper roots. Multi-modal distributions are highly

connected to the topic of niche segregation and has many ecological implications, which we want to avoid in this study. With regard to the comment of referee 2 to significantly shorten the manuscript, we regard this topic as beyond the scope of this study. It will definitively be in the focus of future studies.

We will indicate the drier and wetter site directly in Fig. 5 as suggested.

L 406: Fig 11 (not 9f?)

Thank you for pointing out this mistake. We changed the text accordingly, now in line XX.

L 434: why 4m? Is there a reference for this value?

This short paragraph is a rough qualitative description of our simulation results of mean rooting depth in relation to climate variables. It is not a description of results from other studies. We now clarify this by inserting the word "simulation" in line XX.

L 445: 'behavior: Whereas..' Full stop or small w in whereas.

Thank you very much for this suggestion. We exchanged ':' with '.' As suggested, now in line XX.

L 453: what exactly does 'reversely' mean?

Thank you very much for pointing out this unclear formulation. We reformulated this paragraph into "At STM K77 (Fig. 9f) local circumstances show the influence of variable rooting strategies on ET in a different way. This former rainforest site was converted to pasture before Eddy covariance measurements began. This local land-use at STM K77 is not representative for the respective 0.5° grid cell, and all 3 LPJmL model versions simulate dominant natural forest instead of pasture." now in line XX.

L 377: The text in this paragraph and the figures are mainly about PFTs, not biomes.

Thank you for pointing that out. We changed the heading of this section into "Distribution of plant functional types" now in line XX.

L 519: 'uncertainty...is' or 'uncertainties...are'

Thank you for pointing that out. We decided to change it into "uncertainties are" now in line XX.

L 541: I agree that it's important to look at below ground biomass but comparing Fig 15 and Fig 13 suggest that the ratio between aboveground and below ground biomass is extreme in some areas with high aboveground biomass but low below ground biomass(300-400t/ha aboveground vs ca 20t/ha belowground). Are such ratios realistic in these regions and how can this be explained?

Thank you for this question. In a recent review Fearnside (2018) found that information on belowground biomass of trees in the Amazon region is still very sparse. Available empirical data for 3 sites showed a range of about 15.2 – 33.4 % and a mean of 23.7 % of tree biomass below-ground. No data seems to be available for the western Amazon. Especially in this area we simulate high AGB and low BGB. Here a ratio of 20/(350+20) yields 5.4% of total biomass below-ground. While we are not in the position to validate these values we agree that they might be too low. According to our approach, more root biomass might not be necessary for water uptake and conduction in those

regions, but more root biomass might very well be necessary for the statics of trees. Structures ensuring a tree's stability are neglected by the LPJmL model. The implementation of tree statics would most likely increase the belowground biomass in regions which currently show a very low percentage of total biomass allocated to roots. Nevertheless those structures would be necessary for all sub-PFTs and the overall results presented in our manuscript would most likely not change significantly. We now critically discuss our findings of belowground biomass in this context in line XX-YY.

L 557: according to figure caption in Fig 12, these are PFTs not biomes.

Thank you for pointing that out. We changed the wording to "PFT distribution", now in line XX.

L 565: 'where' instead of 'were'

Thank you for pointing out this mistake. We changed the word accordingly, now in line XX.

I was surprised that grasses and fire were only shortly mentioned in the discussion, given that the study region also includes seasonal areas with Cerrados and not only evergreen forests. How are these systems represented? Only by deciduous forest or does the model also simulate a grassy component and fire? Fire also has some impacts on biomass in these regions and it has been argued that lateritic layers constrain rooting depth and might thereby influence grass-tree coexistence in these regions.

Thank you for this comment. We agree that fire is an important driver forming the vegetation distribution especially outside tropical evergreen forests. For the current version of LPJmL4.0-VR we used the most simplistic fire module available in LPJmL (GlobFirm, Thonicke et al., 2001), which calculates a fire return interval and burned area based on litter moisture only. A PFT dependent parameter for the fraction of killed individuals then determines the burned biomass. Fire-vegetation-feedbacks are therefore existent, but very simplistic. Future studies will incorporate LPJmL's recently updated and much more complex SPITFIRE module (Drüke et al., 2019) and enable to investigate those fire-vegetation feedbacks in a more comprehensive way.

Even though C3 and C4 grasses are explicitly simulated as PFTs in LPJmL4.0, LPJmL4.0 and therefore also LPJmL4.0-VR currently underrepresents the occurrence of grass. This is mainly due to the fact that grass PFTs compete with tree PFTs for area. In that way grass abundance is often highly underestimated when tree PFTs are present. This in turn has the effect that grass-fire feedbacks which naturally stabilize grasslands by reducing tree cover, are not simulated as desired even with a better fire module. Current ongoing developments of LPJmL aim at allowing grass PFTs to grow under any tree PFT canopy. Here, grass PFTs would mainly be affected by the light reduction of trees. In that way grass fire feedbacks could transform areas currently dominated by deciduous forest to become savanna-like vegetation types. We now show grass PFT FPC in a new supplementary figure (Fig. SX). With regard to shortening the paper as suggested by referee 2 and our new research question we briefly discuss the implications of low grass PFT cover for our results in line XX-YY.

With regard to 1) the simplistic fire module used, 2) the underrepresentation of the grass PFTs, 3) our new research question, 4) the comment of referee 2 to shorten the manuscript, and 5) that we do not assess the distribution or stability of potential natural vegetation (i.e. without land-use), we regard the topic of grass-tree coexistence as beyond the scope of this study. Nevertheless, we will

discuss the aforementioned shortcomings of LPJmL4.0 and their implications in the discussion now in line XX.

L 606: the extent of evergreen forest has not been presented, but rather the extent of the evergreen PFT and the deciduous PFT.

Thank you for pointing that out. We now changed the wording "biome" into "PFT" throughout the text.

Further, Fig 12 shows that the extent of the evergreen PFT is very similar in the original and the VR version (although the FPC is much lower in the original version).

Thank you for this remark. We may have missed explaining this result in the manuscript. Fig. 12 g) and h) show that the standard LPJmL4.0 model simulates a rather similar dominance of the evergreen and the deciduous tree PFT in the Amazon region (an almost 50/50 dominance of both PFTs in this region). This model behavior can be explained by the fact that LPJmL4.0 is not capable of simulating a true competitive exclusion over time. PFT establishment rates are not coupled to PFT performance and are in fact equal for all PFTs for every time step (even though the overall establishment rate can vary, but for all tree PFTs in the same way). In Fig. 12 e) and f) we see clear dominance patterns of the evergreen and deciduous tree PFT even though they vastly deviate from the evaluation data (a-b) as well. The underlying model LPJmL4.0-VR-base (just as LPJmL4.0-VR) does simulate a performance dependent PFT establishment as described in the methods section formerly in line 214-228. Therefore, the dominance pattern of evergreen and deciduous tree PFTs in Fig. 12e) and f) can be explained by competitive exclusion. A similar pattern would also be expected in standard LPJmL4.0 when a performance dependent PFT establishment would be implemented. The reason why standard LPJmL4.0's evergreen PFT (Fig. 12g) (and also the deciduous PFT(Fig. 12h)) show a similar extent towards the Southern and Eastern border of the Amazon region compared to the extent of the evergreen PFT in LPJmL4.0-VR (Fig. 12c) is simply because of the human land-use of 2001-2010 applied to all model versions used in our study. The similar extent marks the border of the arc of deforestation in this region. We will clarify these coherences in the main manuscript or the supplement depending on our decision of whether Fig. 12 remains in the main manuscript.

I suggest to clarify or to classify vegetation into biomes based on the FPC of different PFTs. This would allow comparisons of biome cover in different scenarios.

Thank you very much for this suggestion. While we regard some of the PFTs used in LPJmL as representatives of biomes, (e.g. the "tropical broadleaved evergreen tree", as the representative of the biome "tropical rainforest") we agree not to use the word "biome" in the manuscript. When it comes to accounting for what factors actually determine a biome the scientific community follows different definitions. Therefore, we also want to avoid classifying simulated vegetation in yet a new way as suggested. Moreover any classification would lead to a loss of information regarding the results of simulated geographical PFT distribution, especially for those familiar with PFTs and DGVMs. We rather follow the other suggestion and now better clarify our results now in line XX-YY and avoid the word "biome" throughout the text.

L 627: 'Expansion' instead of 'Extent'?

Thank you very much. We changed it accordingly, now in line XX.

L 638: PFT instead of biome types? In the discussion I was missing some discussion of the results in the context of previous modeling studies, such as the studies cited in the introduction. As the study region also includes the Cerrados, the rooting niche separation ideas that explain grass-tree coexistence in savannas might be relevant for the discussion, e.g. Van Langevelde et al 2013 Ecology.

Thank you very much for pointing out that missing connection. We will discuss our results and studies mentioned in the introduction in the context of our new research question. As mentioned above in our answer to the comment regarding L565, grass-tree coexistence is hardly possible to realistically simulate with the current status of the LPJmL model. Yet we will discuss the rooting niche separation hypothesis in the discussion as suggested. With respect to referee 2 requesting a significant shortening of the manuscript we see ourselves forced to keep this discussion rather short.

L 1083: Figure S3 not provided.

We are very sorry for this mistake. We now provide Fig. S3.

Just out of interest, can the model easily be adapted to global scale, and will these model developments be included in the global 'default' version of LPJmL4.0? Or would this lead to computational constraints?

Thank you for your interest. First tests of global scale simulations look very promising, but of course have to be evaluated in detail. The principles found in this study seem to apply to other regions as well. The model runs stable on a global scale and currently needs about 2-4 times longer when including 10 sub-PFTs for all 8 natural PFTs in LPJmL.

References:

Fan, Y., Miguez-Macho, G., Jobbágy, E. G., Jackson, R. B. and Otero-Casal, C.: Hydrologic regulation of plant rooting depth., Proc. Natl. Acad. Sci. U. S. A., 114(40), 10572–10577, doi:10.1073/pnas.1712381114, 2017.

Fearnside, P. M.: Brazil's Amazonian forest carbon: the key to Southern Amazonia's significance for global climate, Reg. Environ. Chang., 18(1), 47–61, doi:10.1007/s10113-016-1007-2, 2018.

Jackson, R. B., Canadell, J., Ehleringer, J., Mooney, H., Sala, O. and Schulze, E.: A global analysis of root distributions for terrestrial biomes, Oecologica, 108, 389–411, 1996.

Johnson, M. O., Galbraith, D., Gloor, M., De Deurwaerder, H., Guimberteau, M., Rammig, A., Thonicke, K., Verbeeck, H., von Randow, C., Monteagudo, A., Phillips, O. L., Brienen, R. J. W., Feldpausch, T. R., Lopez Gonzalez, G., Fauset, S., Quesada, C. A., Christoffersen, B., Ciais, P., Sampaio, G., Kruijt, B., Meir, P., Moorcroft, P., Zhang, K., Alvarez-Davila, E., Alves de Oliveira, A., Amaral, I., Andrade, A., Aragao, L. E. O. C., Araujo-Murakami, A., Arets, E. J. M. M., Arroyo, L., Aymard, G. A., Baraloto, C., Barroso, J., Bonal, D., Boot, R., Camargo, J., Chave, J., Cogollo, A., Cornejo Valverde, F., Lola da Costa, A. C., Di Fiore, A., Ferreira, L., Higuchi, N., Honorio, E. N., Killeen, T. J., Laurance, S. G., Laurance, W. F., Licona, J., Lovejoy, T., Malhi, Y., Marimon, B., Marimon, B. H., Matos, D. C. L., Mendoza, C., Neill, D. A., Pardo, G., Peña-Claros, M., Pitman, N. C. A., Poorter, L., Prieto, A., Ramirez-Angulo, H., Roopsind, A., Rudas, A., Salomao, R. P., Silveira, M., Stropp, J., ter Steege, H., Terborgh, J., Thomas, R., Toledo, M., Torres-Lezama, A., van der Heijden, G. M. F., Vasquez, R., Guimarães Vieira, I. C., Vilanova, E., Vos, V. A. and Baker, T. R.: Variation in stem mortality rates determines patterns of above-ground biomass in Amazonian forests: implications for dynamic global vegetation models,

Glob. Chang. Biol., 22(12), 3996–4013, doi:10.1111/gcb.13315, 2016.

Kleidon, A. and Heimann, M.: A method of determining rooting depth from a terrestrial biosphere model and its impacts on the global water and carbon cycle, Glob. Chang. Biol., 4(3), 275–286, doi:10.1046/j.1365-2486.1998.00152.x, 1998.

R Core Team (2019). R: A language and environment for statistical computing. R Foundation for Statistical Computing, Vienna, Austria. URL https://www.R-project.org/.

Robert J. Hijmans (2019). raster: Geographic Data Analysis and Modeling. R package version 2.9-5. https://CRAN.R-project.org/package=raster

Sitch, S., Smith, B., Prentice, I. C., Arneth, A., Bondeau, A., Cramer, W., Kaplan, J. O., Levis, S., Lucht, W., Sykes, M. T., Thonicke, K. and Venevsky, S.: Evaluation of ecosystem dynamics, plant geography and terrestrial carbon cycling in the LPJ dynamic global vegetation model, Glob. Chang. Biol., 9(2), 161–185, doi:10.1046/j.1365-2486.2003.00569.x, 2003.

Tans, P. and Keeling, R.: Trends in Atmospheric Carbon Dioxide, National Oceanic & Atmospheric Administration, Earth System Research Laboratory (NOAA/ESRL), available at: http://www.esrl.noaa.gov/gmd/ccgg/trends, 2015.

Thonicke, K., Venevsky, S., Sitch, S. and Cramer, W.: The role of fire disturbance for global vegetation dynamics: Coupling fire into a dynamic global vegetation model, Glob. Ecol. Biogeogr., 10(6), 661–677, doi:10.1046/j.1466-822X.2001.00175.x, 2001.

---

## Author Response (AR1)

Dear Editor Martin De Kauwe and referees,

We have thoroughly addressed all points raised by the 2 referees and prepared a new manuscript version with increased readability and a clearer storyline.

Please find below our new specific responses to the comments of referee 1 & 2. Original referee comments are marked as black text. Our answers to each comment are marked as blue text.

Please note in addition to the requested changes we also propose a new slightly different manuscript title which fits better to the storyline.

With best regards and on behalf of all co-authors,

Boris Sakschewski

Response to the comments of referee 1 on Sakschewski et al. 2020: Variable tree rooting strategies improve tropical productivity and evapotranspiration in a dynamic global vegetation model

In this manuscript, the authors extended the LPJmL4.0 dynamic vegetation model to simulate variable rooting depth. Comparisons between different model versions for tropical South America showed that the novel approach improves most of the bench-marks used in the study. Overall, the study is well-conducted and solid. I agree with the authors that including variable rooting depth is important in DGVMs and the results and the presented model approach are therefore relevant for the modeling community. Previous models typically assumed PFT specific fixed rooting depth and/or a fixed soil depth globally. My main point is, however, that the manuscript reads like a collection of many interesting results while I was missing more specific aims or questions regarding for example implications of root diversity for the ecology or biogeography of the simulated vegetation types. Currently, the aims of the study are to 'describe an approach' and to' evaluate its effect on' various model variables.

Dear referee,

Thank you for your positive evaluation and many constructive comments on our manuscript. We agree that the paper benefits from focusing on a specific research question which creates a stronger storyline. As we will now highlighted in the introduction in lines 101-103 we now focus on "the hypothesis that varying tree rooting depth is key to explain major patterns of evapotranspiration, productivity and the geographical distribution of tropical evergreen forests in South America" and structure the manuscript accordingly. In connection to comments of referee 2 we will shorten the manuscript and move parts of the results to the supplement. This also helps to create a stronger and clearer storyline.

L 25: PFT-specific instead of biome-scale?
Thank you for this comment. In the abstract we wanted to avoid specific terms like "PFT" to keep it easy to understand for readers which are not familiar with those terms, but still convey the main message. In the new version of the abstract there is no introduction of PFTs anymore.

L 29: realistically simulated
Thank you for this suggestion. We changed it accordingly, now in line 27.

L 41: delete quotation marks for evergreen and deciduous?
Thank you for this suggestion. We changed it accordingly, now in line 40.

L 47-48: Isn't it the other way round? Traits are aggregated to define PFTs and the fractional cover of different PFTs in simulations then defines the biome type?
Thank you for raising this point. We apologize for the sloppy usage of "biome" and "PFT". To clarify the text we now state "In general these models condense the diversity of such functional tree traits into so called plant functional types (PFTs), which represent average tree growing strategies on scales as large as biomes." in line 45-47.

L 54: reword 'different attempts were carried out', maybe 'different approaches were presented'?
Thank you for this suggestion. We changed it accordingly, now in line 52.

L 54: Schymanski et al. also developed root models (e.g. www.hydrol-earth-syst-sci.net/12/913/2008/). Although these models were not developed explicitly in the DGVM context, they might be relevant for the introduction or the discussion.

Thank you for this suggestion. We now refer to Schymanski et al. (2008) in the introduction in line 84 to acknowledge that root optimization models do exist, but the general knowledge base on phenotypic plasticity of plant traits remains too little to be applied in a DGVM. E.g. it remains unclear how fast plasticity can occur and what the true drivers of plasticity are.

L 55-56: 'study ... searched for...' reword, I would think that a study can's search.

Thank you for this correction. We changed the sentence to "In a pioneering study more than 20 years ago, Kleidon and Heimann (1998) systematically searched for rooting strategies which yield highest net primary productivity …" in line 52.

L 74: 'allocation strategy' instead of 'direction'?

Thank you for this suggestion. We changed it accordingly, now in line 73.

L 97: in competition with other rooting strategies/plants/PFTs?

Thank you for this question. We changed the respective sentence to "all tree rooting strategies of this spectrum grow in competition…" in line 96-97 as it is meant to contrast this study with some of the former approaches. A more detailed sentence would lead to a full description of the competition scheme which can be found in the methods in Sect. 2.2.6.

L 100: this suggests to me that rooting depth is related to or a function of tree height.

Thank you for this comment. Yes as we clarified in the methods we used this approach in Sect. 2.2.5.

L 105-108: I think that such a general overview is not necessary and could be removed (also elsewhere in the methods section). Further, it only refers to some selected section and it is incomplete.

We agree with the referee and deleted this part.

L 115: simulates instead of employs?

Thank you for this remark. We changed it accordingly, now in line 114.

L 116: Should it be 'bioenergy functional type'?

Thank you for pointing out this mistake. We inserted "functional" now in line 114.

L 122: 'how different tree rooting strategies (implemented in the new scheme) compete'?

Thank you for this suggestion. We implemented it, now in line 124-125.

L 131: Does that mean that soil texture is similar for all layers in the soil column?

Thank you for this question. Yes. In LPJmL4.0 each grid cell has one soil texture information for its 3 m soil column only. We followed this approach for our larger soil columns as well. In fact a high resolution soil texture information in 3 dimensions for the study region is so far not available, or only partly available. We wanted to keep things simple and comprehensible also with regard to comparing results of LPJmL4.0-VR to LPJmL4.0. We now better clarify the approach in line 134-135 stating "Equal to LPJmL4.0 (Schaphoff et al., 2018), we use a grid cell specific soil texture information which is applied to the whole soil column".

L 142: PFT scale instead of biome scale?

Thank you for this suggestion. We changed it as suggested, now line 161.

L 144 and elsewhere: is it really the depth which is reached by 95% of the roots or does it rather mean that 95% of the fine root biomass is between the surface and D95_max? Wouldn't 'reached by 95%' mean that 95% are deeper than D95_max?

Thank you very much for pointing out that out. We changed the sentence to "...we here calculate the depth at which the cumulated fine root biomass from the soil surface downwards makes up 95% of all fine root biomass", now in line 162-163.

L 146: that's a question related to LPJml4.0 and not to the VR version: why were different values for evergreen and deciduous selected if the resulting root profiles are essentially identical. Would model results differ for similar values?

Thank you for this question. The parameter values are average values for those 2 PFTs found in Jackson et al. (1996). At model development of LPJmL those values were the best available. We now specifically mention where those parameter values came from in line 165-166.

For now it is clear that increasing the difference between the beta values of the 2 PFTs would most probably enhance model performance. Even with only 2 PFTs present. Here the best way to go would probably be to increase the beta value of the evergreen PFT to reach a D95_max of maybe 2-4 m, which would buffer against some dry season signal. Regarding the second question: If regarding beta values only, the answer is no, there would be no difference between the evergreen and deciduous PFT. Fortunately, they also differ in other functional traits such as specific leaf are (SLA) or leaf longevity (LL), determining their phenology strategy and therefore their performance under different climate regimes.

L 149: why 18m and not 20m or 95% of 20m?

Thank you very much for this question. As we have chosen a maximum soil depth of 20m we wanted to avoid a significant accumulation of fine roots in the last soil layers. A D95_max of 20m in a soil column of 20 m would mean that the additional 5% of fine roots are also distributed between 19-20m. In principle it was a choice between a round number of soil depth or a round number at the largest value of D95_max. In the current model version of LPJmL4.0-VR a D95_max of 20m is not missing in the study area as distributions of D95_max always flatten towards the 18m bin in grid cells where deep roots dominate. Nevertheless, it is possible to substantially increase soil depth and create more sub-PFTs with larger D95_max values. This step might even be necessary in future versions of the model or other study areas. We now explain why we chose the largest D95_max at 18m in line 169-170 stating "We chose 18m as the largest D95_max value in order to avoid that roots of the respective sub-PFT significantly exceed the maximum soil depth of 20m (see also 2.2.4 and Fig. 2 right panel)."

L 154: a new carbon allocation scheme?

Thank you for this suggestion. We changed it accordingly, now in line 175.

L 163: 'sapwood ... proportional' which constant was used to describe this relation? And shouldn't root and stem sapwood are be identical to be able to transport the same amount of water?

Thank you for pointing this out. Our explanation was misleading. We do not use a constant here. The term proportional is falsely used. It is as your second question suggests and as it is written in the description of Fig. 2. We now clarify the approach in the methods section in line 184-187 with "Root sapwood cross-sectional area in the first soil layer is equal to stem sapwood cross-sectional area, as all water must be transported through the root sapwood within this soil layer. In the following soil layers downwards, root cross-sectional area decreases by the relative amount of fine roots in all soil layers above (Fig. 2)."

L 180: 'to derive a functional relation between tree height and rooting depth'?

Thank you for this suggestion. We changed it accordingly, now in line 200-203.

L 203: check brackets in Huang et al
Thank you for pointing this out. We changed the brackets, now in line 226.

L 233: I suggest to delete 'The new features... direction.' and say 'In LPJmL4.0-VR, PFTs can...'
Thank you for this suggestion. We changed it accordingly, now in line 258-259.

L 234: 'formally', I assume this should be 'formerly' or 'in previous model versions'.
Thank you very much. We changed it accordingly, now in line 258.

L 233-240: An increase of mortality rates from 3% to 7% is quite substantial and more then twice as high. In addition, it is stated that observed rates do not exceed 6% which means that 7% is not in the real world boundaries as stated in L 237 but it overestimates this rate by >16%.
Thank you very much for raising this point. We agree that claiming 7% is in the boundary of 6% is overstated. The point we wanted to make was mixed up and lost by citing the study mentioning a observed maximum of 6% background mortality rate in 167 Amazon forest plots (Johnson et al., 2016) and comparing it to the maximum (not actual background) mortality of LPJmL4.0-VR. In fact observed background mortality rate and conceivable maximum mortality rate are not the same thing. The growth efficiency related mortality which is used in many DGVMs needs a maximum mortality rate, because otherwise the simulated mortality rate would rise to 100% when total respiration exceeds productivity. In the real world though plants would optimize and reallocate carbon pools under those circumstances as far as they can. Nevertheless, a mortality rate of 100% in the real world is conceivable as well. The maximum mortality rate of a forest under a hypothetical super extreme scenario (no rain for 12 month) could of course be 100%.
Accordingly, in a hypothetical model which truly captures all important mechanisms of tree survival and mortality, the maximum mortality rate could be set close to 100% as we tried to explain formerly in line 239-240. In other words this constant could eventually be obsolete. The road to such a model will be long and might never be ending, but until then, a maximum mortality rate at any value will always be necessary to achieve biomass values in acceptable ranges. We now deleted the reference of observed mortality rates of 6% in the manuscript as this number is not comparable to the maximum mortality rate set in the model. Moreover, we will describe the true purpose of a maximum mortality rate and why we changed it (as described in this answer) now in Sect. 2.2.7.

L 239: 'We regard ...' I don't understand this sentence. And what does 'right direction' refer to? I suggest to reword.
We deleted this sentence in accordance with our response above regarding "L 233-240".

L 243: Why were four different climate inputs used? Most simulation results are shown only for CRU anyway. Which climate variables were taken from the different datasets?
Thank you very much for those questions. We wanted to show that our results remain robust using various climate inputs and decided (also with regard to the amount of figures and respective manuscript text) to show this with regional rooting depth (Fig. 6) and regional evapotranspiration (corridors in Fig. 11). In connection to referee 2 requesting a substantial reduction of the manuscript size and amount of figures, it seems impracticable to produce all figures for all climate inputs. We also argue that, Fig. 6 & 11 show some proof that results for different climate inputs are very similar. There are local differences, but the big picture remains the same. As we do not want to go into detail regarding local scale differences we hope the current amount of results is enough.
We added 2 sentences to clarify our point in the discussion in line 495-498.
We now better clarify what climate variables were taken from the different data sets in Sect. 2.3.1 in line 265-266.

L 267: Reword to 'caused by the the presence or absence of variable rooting strategies'?
Thank you very much for this suggestion. We changed it accordingly, now in line 289-291.

L 269: Given that model spin-up was conducted for different time periods for different climate input datasets and at 278ppm. Is there a jump in CO2 in the transition between spin-up and transient phase?
Thank you very much for this question. No there is a smooth transition between spin-ups and transient simulations regarding atmospheric CO2 content. Before the year 1840 a constant value of 278ppm is used, while after this year values are rising according to a LPJmL protocol first introduced in Sitch et al. (2003) and eventually following the Mona Loa record of Tans and Keeling (2015). We now clarify this approach in the methods section in line 292-302.

L 280: I understood that replicate simulations were not conducted. I was wondering how robust or deterministic the selection of rooting strategies is? Would you expect substantial differences in the results when conducting replicate model runs?
Thank you very much for this question. Given the vast amount of different model test runs during model development, we can assure that results are very robust. We admit we do not fully proof this in this study. However, the 4 very similar rooting depths maps (formerly Fig. 6 now Fig. 5) provide some proof.
Since there are no real stochastic processes in LPJmL which could lead to a path dependency of vegetation dynamics, this model behavior was expected. Therefore, it is also a standard procedure to not conduct any replicate simulations when using LPJmL. We now clarify this point in the discussion in line 495-498.

L 285: which method was used for re-gridding?
Thank you for pointing out this lack of information. We now name the methods, software packages and underlying studies. Regridding of Avitabile et al. (2016) was conducted in R (R Core Team, 2019) using the "aggregate" function of the R-package "raster" (Hijmans, 2019), which aggregates to lower resolutions by taking the arithmetic mean excluding NAs. Regridded data of Saatchi et al. (2011) was taken from Carvalhais et al. (2014). This information is now provided in line 313-316.

L 289: check brackets in Brienen
Thank you for pointing that out. In order to shorten the manuscript according to referee 2 we decided to remove the biomass comparisons based on inventory biomass. Therefore, the citation is now deleted.

L 287: I found it difficult to understand the description of the Rammig et al method and I had to go back to the original paper. I suggest to check the paragraph again for clarity.
Thank you for raising this point. In order to shorten the manuscript according to referee 2 we decided to remove the biomass comparisons based on inventory biomass from the manuscript.

L 304: check bar in 'average x_bar'
Thank you for pointing out this mistake.
In order to shorten the manuscript according to referee 2 we decided to remove the biomass comparisons based on inventory biomass from the manuscript.

L 311: I am skeptical when using gridded climate products to simulate local scale EC fluxes, because these products might not capture some local rainfall events (for example) that have strong impacts on the fluxes. Hence, models will fail to simulate the fluxes. I assume that there are there local scale meteorological data available for the flux sites that could be used for running the model or at least for comparing agreement between gridded data products and observation at EC sites.

We fully agree that using gridded climate products is not optimal to reproduce locally measured ET fluxes, because they can lack information of local weather events. We solely use gridded data in this study due to several reasons: 1) We want to stay consistent with the regional results of this study. As we e.g. show a simulated regional rooting depth maps (formerly Fig. 6 now Fig. 5) and plots of underlying local tree rooting strategies (formerly Fig. 5 now Fig. 4) as well as regional ET (formerly Fig. 11 now Fig. 9) which are all based on gridded climate data, changing the climate input for simulations at local scale seemed inconsistent. 2) Even though we apply statistical metrics to compare model vs. flux agreement, our focus was not on the effects of local weather events on ET, rather on the general climate signal, most importantly the presence or absence of a dry season and the effects on simulated rooting depth and ET and the differences between the different model versions. We fully agree that forcing the model with local climate data could in principle enhance model performance, but we never aimed for a perfect match of ET and NEE at all sites. 3) There is meteorological data available for different flux sites, but these data sets are often cluttered with gaps and are only available for a few up to 10 years only. Moreover, each site has its own limitations when it comes to model implementation. Taken together, this creates quite some problems for DGVM simulations. The LPJmL model needs continues climate data and long time series. Jumping from a spin-up simulation into repeating 5-10 very similar years can cause artifacts which should be avoided. We would be happy simulate rainforest sites with real site meteorological data, but for this we would need longer time spans than currently available. There are approaches trying to solve those problems, but they are currently beyond the scope of this study. 4) Given that we compare monthly means of simulated and measured ET at the local scale and that simulation results appear to capture seasonal signals (formerly Fig. 9-10 now Fig. 7-8), we are convinced that our approach is sufficient to deliver the message of this manuscript.
We will insert a clarification of why we used gridded climate input data only in the discussion in line 584-588.

L 313: I suggest to state why NEE was only simulated for 3 sites, this information is currently hidden in the figure caption.

Thank you very much for this remark. Unfortunately, in the data sets accessible to us, continues NEE data covering at least 2 years was only available for 3 sites. We now clarify why we compare NEE only at 3 sites in the methods sections in line 320-321 and in the caption of Fig. 8.

L 336: replace 'called' by ':'

Thank you very much. We changed it accordingly, now line 345.

L 376: replace 'over' by 'instead of'?

Thank you for pointing that out. We changed it accordingly, now line 382.

Generally the results section contains some statements or explanations that do not only describe the results but already go beyond and might be more more appropriate for the discussion.

We will check the results section and will transfer all potential interpretations into the discussion. We have moved and rewritten large parts of the results and discussion section in order to avoid interpretations in the results.

L 387, Fig 5: when looking at this figure, I was wondering if simulated distributions are always unimodal or if there the model can also simulate bi-modal or multi-modal distributions indicating that very distinct rooting strategies can coexist? I also suggest to add to the figure caption which site is wetter and which site is drier.

Thank you for this comment. Indeed these distributions can be bimodal, indicating that very distinct rooting strategies can co-exist. We have not systematically checked all grid cells, but a clear tri-modal distribution was not observed so far. This might also need different ways of detecting them as multiple modes might be hidden in a continuous distribution. So far we observed 2 cases where distributions can clearly be bi-modal. 1) In areas with dominant evergreen tree cover and a "medium" dry season, where shallow and deep rooting evergreen sub-PFTs can co-exist. 2) Areas with a substantial dry season where the evergreen and deciduous PFT co-exist. Here the deciduous PFT shows shallower roots and the evergreen PFT deeper roots. Multi-modal distributions are highly connected to the topic of niche segregation and have many ecological implications, which we want to avoid in this study. With regard to the comment of referee 2 to significantly shorten the manuscript, we regard this topic as beyond the scope of this study. It will definitely be in the focus of future studies.

We indicate the drier and wetter site directly in the respective figure (formerly Fig. 5 now Fig. 4) as suggested.

L 406: Fig 11 (not 9f?)

Thank you for pointing out this mistake. By coincidence the former Fig. 11 is now Fig. 9. The reference is now in line 414.

L 434: why 4m? Is there a reference for this value?

This short paragraph is a rough qualitative description of our simulation results of mean rooting depth in relation to climate variables. It is not a description of results from other studies. We now clarify this by inserting the word "simulation" in line 423.

L 445: 'behavior: Whereas..' Full stop or small w in whereas.

Thank you very much for this suggestion. The sentenced is deleted in the new version of the manuscript.

L 453: what exactly does 'reversely' mean?

Thank you very much for pointing out this unclear formulation. We reformulated this paragraph into "At STM K77 (Fig. 7f) local circumstances show the influence of variable rooting strategies on ET in a different way. This former rainforest site was converted to pasture before Eddy covariance measurements began. This local land-use at STM K77 is not representative for the respective 0.5° grid cell, and thus all 3 LPJmL model versions simulate mainly natural vegetation instead of pasture." now in line 442-445.

L 377: The text in this paragraph and the figures are mainly about PFTs, not biomes.

Thank you for pointing that out. We changed the heading of this section into "Distribution of plant functional types" now in line 463.

L 519: 'uncertainty...is' or 'uncertainties...are'

Thank you for pointing that out. We decided to remove the whole paragraph from the manuscript in order to shorten it.

L 541: I agree that it's important to look at below ground biomass but comparing Fig 15 and Fig 13 suggest that the ratio between aboveground and below ground biomass is extreme in some areas with high aboveground biomass but low below ground biomass(300-400t/ha aboveground vs ca 20t/ha belowground). Are such ratios realistic in these regions and how can this be explained?

Thank you for this question. In a recent review Fearnside (2018) found that information on belowground biomass of trees in the Amazon region is still very sparse. Available empirical data for 3 sites showed a range of about 15.2 – 33.4 % and a mean of 23.7 % of tree biomass below-ground. No data seems to be available for the western Amazon. Especially in this area we simulate high AGB and low BGB. Here a ratio of 20/(350+20) yields 5.4% of total biomass below-ground. While we are not in the position to validate these values we agree that they might be too low. According to our approach, more root biomass might not be necessary for water uptake and conduction in those regions, but more root biomass might very well be necessary for the statics of trees. Structures ensuring a tree's stability are neglected by the LPJmL model. The implementation of tree statics would most likely increase the belowground biomass in regions which currently show a very low percentage of total biomass allocated to roots. Nevertheless those structures would be necessary for all sub-PFTs and the overall results presented in our manuscript would most likely not change significantly. We now critically discuss our findings of belowground biomass in this context in line 567-571.

L 557: according to figure caption in Fig 12, these are PFTs not biomes.

Thank you for pointing that out. We deleted this paragraph in order to shorten the manuscript.

L 565: 'where' instead of 'were'

Thank you for pointing out this mistake. We changed the word accordingly, now in line 504.

I was surprised that grasses and fire were only shortly mentioned in the discussion, given that the study region also includes seasonal areas with Cerrados and not only evergreen forests. How are these systems represented? Only by deciduous forest or does the model also simulate a grassy component and fire? Fire also has some impacts on biomass in these regions and it has been argued that lateritic layers constrain rooting depth and might thereby influence grass-tree coexistence in these regions.

Thank you for this comment. We agree that fire is an important driver forming the vegetation distribution especially outside tropical evergreen forests. For the current version of LPJmL4.0-VR we used the most simplistic fire module available in LPJmL (GlobFirm, Thonicke et al., 2001), which calculates a fire return interval and burned area based on litter moisture only. A PFT dependent parameter for the fraction of killed individuals then determines the burned biomass. Fire-vegetation-feedbacks are therefore existent, but very simplistic. Future studies will incorporate LPJmL's recently updated and much more complex SPITFIRE module (Drüke et al., 2019)and enable to investigate those fire-vegetation feedbacks in a more comprehensive way.

Even though C3 and C4 grasses are explicitly simulated as PFTs in LPJmL4.0, LPJmL4.0 and therefore also LPJmL4.0-VR currently underrepresents the occurrence of grass. This is mainly due to the fact that grass PFTs compete with tree PFTs for area. In that way grass abundance is often highly underestimated when tree PFTs are present. This in turn has the effect that grass-fire feedbacks which naturally stabilize grasslands by reducing tree cover, are not simulated as desired even with a

better fire module. Current ongoing developments of LPJmL aim at allowing grass PFTs to grow under any tree PFT canopy.  Here, grass PFTs would mainly be affected by the light reduction of trees. In that way grass fire feedbacks could transform areas currently dominated by the deciduous PFT to become more dominated by the grass PFT, i.e. comparable to savanna-like vegetation types. We now show grass PFT FPC in a new appendix figure (Fig. A6).

With regard to 1) the simplistic fire module used, 2) the underrepresentation of the grass PFTs, 3) our new research question, 4) the comment of referee 2 to shorten the manuscript, and 5) that we do not assess the mechanisms that influence the distribution or stability of potential natural vegetation (i.e. without land-use), we regard the topic of grass-tree coexistence as beyond the scope of this study. Nevertheless, we will discuss the aforementioned shortcomings of LPJmL4.0 and their implications in the discussion now in line 545-551.

L 606: the extent of evergreen forest has not been presented, but rather the extent of the evergreen PFT and the deciduous PFT.

Thank you for pointing that out. We now changed the wording "biome" into "PFT" throughout the text.

Further, Fig 12 shows that the extent of the evergreen PFT is very similar in the original and the VR version (although the FPC is much lower in the original version).

Thank you for this remark. We may have missed explaining this result in the manuscript. The former Fig. 12g-h which is now Fig. 10g-h show that the standard LPJmL4.0 model simulates a rather similar dominance of the evergreen and the deciduous tree PFT in the Amazon region (an almost 50/50 dominance of both PFTs in this region). This model behavior can be explained by the fact that LPJmL4.0 is not capable of simulating a true competitive exclusion over time. PFT establishment rates are not coupled to PFT performance and are in fact equal for all PFTs for every time step (even though the overall establishment rate can vary, but for all tree PFTs in the same way). In the former Fig. 12e-f now Fig. 10e-f we see clear dominance patterns of the evergreen and deciduous tree PFT even though they vastly deviate from the evaluation data (formerly Fig12a-b now Fig. 10a-b) as well. The underlying model LPJmL4.0-VR-base (just as LPJmL4.0-VR) does simulate a performance dependent PFT establishment as described in the methods in Sect. 2.2.6 (formerly 2.2.5). Therefore, the dominance pattern of evergreen and deciduous tree PFTs in Fig. 10e-f can be explained by competitive exclusion. A similar pattern would also be expected in standard LPJmL4.0 when a performance dependent PFT establishment would be implemented. The reason why standard LPJmL4.0's evergreen PFT (Fig. 10g) shows a similar geographical distribution towards the Southern and Eastern border of the Amazon region compared to the extent of the evergreen PFT in LPJmL4.0-VR (formerly Fig. 12c now Fig.10c) is simply because of the human land-use of 2001-2010 applied to all model versions used in our study. The similar extent marks the border of the arc of deforestation in this region. We now clarify this coherence in line 518-528.

I suggest to clarify or to classify vegetation into biomes based on the FPC of different PFTs. This would allow comparisons of biome cover in different scenarios.

Thank you very much for this suggestion. While we regard some of the PFTs used in LPJmL as representatives of biomes, (e.g. the "tropical broadleaved evergreen tree", as the representative of the biome "tropical rainforest") we agree not to use the word "biome" in the manuscript. When it

comes to accounting for what factors actually determine a biome the scientific community follows different definitions. Therefore, we also want to avoid classifying simulated vegetation in yet a new way as suggested. Moreover any classification would lead to a loss of information regarding the results of simulated geographical PFT distribution, especially for those familiar with PFTs and DGVMs. We rather follow the other suggestion and now better clarify what the tropical PFTs stand for in line 116-119 and avoid the word "biome" throughout the text.

L 627: 'Expansion' instead of 'Extent'?

Thank you very much. The sentence containing this mistake is now deleted.

L 638: PFT instead of biome types? In the discussion I was missing some discussion of the results in the context of previous modeling studies, such as the studies cited in the introduction. As the study region also includes the Cerrados, the rooting niche separation ideas that explain grass-tree coexistence in savannas might be relevant for the discussion, e.g. Van Langevelde et al 2013 Ecology.

Thank you very much for pointing out that missing connection. We will discuss our results and studies mentioned in the introduction in the context of our new research question. As mentioned above in our answer to the comment regarding L565, grass-tree coexistence is hardly possible to realistically simulate with the current status of the LPJmL model. As stated above we now point out the model shortcomings in line 545-551. In this study we want to avoid the analyses on evergreen to deciduous PFT transition as this would require a whole new set of experiments, figures and statistics which is currently beyond the scope of the manuscript (also with respect to referee 2 requesting a significant shortening of the manuscript) and beyond the capabilities of the model settings. We here want to focus on the role of root adaptations for the broad scale distribution of tropical evergreen forest, ET and productivity in South America. We discuss the potential causes for PFT transitions in line 545-557, but don't go into a in depth analyses. In fact we argue that analyses of e.g. natural forest-savannah transition and bi-stability should rather be based on simulations excluding human land use and human fire ignitions (which make up the majority of ignitions in South America), i.e. such analyses should be based on simulations of potential natural vegetation, which would bias simulated regional ET rates and their comparison to current regional ET rates based on remotely sensed data sets.

L 1083: Figure S3 not provided.

We are very sorry for this mistake. We now provide the figure as Fig. A5.

Just out of interest, can the model easily be adapted to global scale, and will these model developments be included in the global 'default' version of LPJmL4.0? Or would this lead to computational constraints?

Thank you for your interest. First tests of global scale simulations look very promising, but of course have to be evaluated in detail. The principles found in this study seem to apply to other regions as well. The model runs stable on a global scale and currently needs about 2-4 times longer when including 10 sub-PFTs for all 8 natural PFTs in LPJmL. It is therefore conceivable to make this model development (or a version with a somewhat reduced amount of sub-PFTs per PFT) a constant model feature or at least an optional feature in the future. This still has to be decided by the LPJmL model development community.

References:

Avitabile, V., Herold, M., Heuvelink, G. B. M., Lewis, S. L., Phillips, O. L., Asner, G. P., Armston, J., Ashton, P. S., Banin, L., Bayol, N., Berry, N. J., Boeckx, P., de Jong, B. H. J., Devries, B., Girardin, C. A. J., Kearsley, E., Lindsell, J. A., Lopez-Gonzalez, G., Lucas, R., Malhi, Y., Morel, A., Mitchard, E. T. A., Nagy, L., Qie, L., Quinones, M. J., Ryan, C. M., Ferry, S. J. W., Sunderland, T., Laurin, G. V., Gatti, R. C., Valentini, R., Verbeeck, H., Wijaya, A. and Willcock, S.: An integrated pan-tropical biomass map using multiple reference datasets, Glob. Chang. Biol., 22(4), 1406–1420, doi:10.1111/gcb.13139, 2016.

Carvalhais, N., Forkel, M., Khomik, M., Bellarby, J., Jung, M., Migliavacca, M., Saatchi, S., Santoro, M., Thurner, M. and Weber, U.: Global covariation of carbon turnover times with climate in terrestrial ecosystems, Nature, 514(7521), 213–217, 2014.

Drüke, M., Forke, M., Bloh, W. Von, Sakschewski, B., Cardoso, M., Bustamante, M., Kurths, J. and Thonicke, K.: Improving the LPJmL4-SPITFIRE vegetation-fire model for South America using satellite data, Geosci. Model Dev., 12(12), 5029–5054, doi:10.5194/gmd-12-5029-2019, 2019.

Fearnside, P. M.: Brazil's Amazonian forest carbon: the key to Southern Amazonia's significance for global climate, Reg. Environ. Chang., 18(1), 47–61, doi:10.1007/s10113-016-1007-2, 2016.

Jackson, R. B., Canadell, J., Ehleringer, J., Mooney, H., Sala, O. and Schulze, E.: A global analysis of root distributions for terrestrial biomes, Oecologica, 108, 389–411, 1996.

Johnson, M. O., Galbraith, D., Gloor, M., De Deurwaerder, H., Guimberteau, M., Rammig, A., Thonicke, K., Verbeeck, H., von Randow, C., Monteagudo, A., Phillips, O. L., Brienen, R. J. W., Feldpausch, T. R., Lopez Gonzalez, G., Fauset, S., Quesada, C. A., Christoffersen, B., Ciais, P., Sampaio, G., Kruijt, B., Meir, P., Moorcroft, P., Zhang, K., Alvarez-Davila, E., Alves de Oliveira, A., Amaral, I., Andrade, A., Aragao, L. E. O. C., Araujo-Murakami, A., Arets, E. J. M. M., Arroyo, L., Aymard, G. A., Baraloto, C., Barroso, J., Bonal, D., Boot, R., Camargo, J., Chave, J., Cogollo, A., Cornejo Valverde, F., Lola da Costa, A. C., Di Fiore, A., Ferreira, L., Higuchi, N., Honorio, E. N., Killeen, T. J., Laurance, S. G., Laurance, W. F., Licona, J., Lovejoy, T., Malhi, Y., Marimon, B., Marimon, B. H., Matos, D. C. L., Mendoza, C., Neill, D. A., Pardo, G., Peña-Claros, M., Pitman, N. C. A., Poorter, L., Prieto, A., Ramirez-Angulo, H., Roopsind, A., Rudas, A., Salomao, R. P., Silveira, M., Stropp, J., ter Steege, H., Terborgh, J., Thomas, R., Toledo, M., Torres-Lezama, A., van der Heijden, G. M. F., Vasquez, R., Guimarães Vieira, I. C., Vilanova, E., Vos, V. A. and Baker, T. R.: Variation in stem mortality rates determines patterns of above-ground biomass in Amazonian forests: implications for dynamic global vegetation models, Glob. Chang. Biol., 22(12), 3996–4013, doi:10.1111/gcb.13315, 2016.

Kleidon, A. and Heimann, M.: A method of determining rooting depth from a terrestrial biosphere model and its impacts on the global water and carbon cycle, Glob. Chang. Biol., 4(3), 275–286, doi:10.1046/j.1365-2486.1998.00152.x, 1998.

Schaphoff, S., von Bloh, W., Rammig, A., Thonicke, K., Biemans, H., Forkel, M., Gerten, D., Heinke, J., Jägermeyr, J., Knauer, J., Langerwisch, F., Lucht, W., Müller, C., Rolinski, S. and Waha, K.: LPJmL4 – a dynamic global vegetation model with managed land – Part 1: Model description, Geosci. Model Dev., 11(4), 1343–1375, doi:10.5194/gmd-11-1343-2018, 2018.

Schymanski, S. J., Sivapalan, M., Roderick, M. L., Beringer, J. and Hutley, L. B.: An optimality-based model of the coupled soil moisture and root dynamics, Hydrol. Earth Syst. Sci., 12(3), 913–932, doi:10.5194/hess-12-913-2008, 2008.

Sitch, S., Smith, B., Prentice, I. C., Arneth, A., Bondeau, A., Cramer, W., Kaplan, J. O., Levis, S., Lucht, W., Sykes, M. T., Thonicke, K. and Venevsky, S.: Evaluation of ecosystem dynamics, plant geography

and terrestrial carbon cycling in the LPJ dynamic global vegetation model, Glob. Chang. Biol., 9(2), 161–185, doi:10.1046/j.1365-2486.2003.00569.x, 2003.

Thonicke, K., Venevsky, S., Sitch, S. and Cramer, W.: The role of fire disturbance for global vegetation dynamics: Coupling fire into a dynamic global vegetation model, Glob. Ecol. Biogeogr., 10(6), 661–677, doi:10.1046/j.1466-822X.2001.00175.x, 2001.

**Response to the comments of referee 2 on Sakschewski et al. 2020: Variable tree rooting strategies improve tropical productivity and evapotranspiration in a dynamic global vegetation model**

I like many aspects of this paper. In particular I appreciate the approach of carefully examining the behaviour of the model across a range of climate scenarios,
Overall I think this paper and model variant have great potential. The new scheme is clever and I have little to critique in its design. The various analyses do suggest the new model is providing a better match to available data. But I am yet to be convinced this improved match is for the reasons the authors claim (better handling of soil water & rooting).

Dear Daniel Falster,

Thank you very much for your positive evaluation of our approach.

The biggest concern I have is that despite all the results presented, these mostly for aggregate outputs at largish spatial scales. Almost no evidence is presented to show the effect of the new scheme on the actual water balance in the soil. Moreover, we don't even know how water is modelled in the different versions. Sure, the soil and root depth is changed, but what does this mean for water balance at different soil depths? Surely this is key to assessing why the model behaves differently. This is important, as the changes in rooting will also change the way carbon is allocated within the model (e.g. deeper roots divert carbon away from leaves). Are the model improvements due to changes in hydraulics, or changes in carbon allocated between leaves and roots?

Thank you very much for raising this point. Indeed we did not explicitly explain how the water percolation scheme and water balance of all 3 LPJmL model versions works. We inserted a new description of this part of the model in Sect. 2.2.2 line 136-151. We agree that we do not present results of how the new scheme affects the water balance in the soil. However, we compare the results of our new scheme (LPJmL4.0-VR) to a baseline model (LPJmL4.0-VR-base). LPJmL4.0-VR-base differs from LPJmL4.0-VR only in the amount of tree rooting strategies present. Both models have the same soil depth information. Hence, soil hydraulics are the same as well and differences in evapotranspiration, net primary productivity, biomass and PFT distribution only arise from the absence/presence of different tree rooting strategies. Deeper roots clearly correlate with higher evapotranspiration and productivity during dry seasons. When shallow roots lead to decreasing evapotranspiration in dry seasons, then the water used for evapotranspiration by deep rooted sub-PFTs must come from deeper soil layers. Regardless of knowing the exact soil water balance we argue that this evidence is presented. The fact that deep rooted sub-PFTs are selected for in regions with deep soils and a dry season, even though deeper roots are more costly than shallow roots, proves that diverting carbon away from the leaves into a deeper root system can be beneficial for the overall carbon balance. Therefore, the differences in the results between LPJmL4.0-VR and LPJmL4.0-VR-base must be related to the new rooting scheme and not to soil hydraulics. Nevertheless, we prepared a new potential Appendix figure (Fig. A7) which we pasted right under this specific response. It shows the difference in monthly relative soil water saturation between LPJmL4.0-VR-base and LPJmL4.0-VR at STM KM67, for 2001-2010, for all 23 soil layers (so LPJmL4.0-VR-base minus LPJmL4.0-VR). Here blue colors depict a lower soil water content in LPJmL4.0-VR compared to LPJmL4.0-VR-base. The figure clearly shows how deep water is extracted by deep roots in the dry

season and how the soil water balance changes over the year. We refer to this new assisting appendix figure now in line 596-598 and hope this presents enough evidence.

[Figure]

**Figure A7: Difference in soil water reaction to seasonal precipitation between LPJmL4.0-VR-base and LPJmL4.0-VR at Fluxnet site STM KM67 a) Mean monthly precipitation input from CRU for 2001-2010. b) Difference in monthly relative soil water content between LPJmL4.0-VR-base and LPJmL4.0-VR forced with CRU climate for 2001-2010. The underlying model output variable "soil water content" of each model version is a number between 0 and 1 depicting the relative water saturation of the soil. Blue colors denote lower soil water content in LPJmL4.0-VR and red colors a lower soil water content in LPJmL4.0-VR-base.**

Second, I feel the authors need to come up with a stronger story and reduced set of results. The paper is currently very long and dense. The authors have made many comparisons using a variety of datasets. Consequently, there is a large number of figures (15) and tables (8). This makes it hard for us to know where to put our attention.

Thank you very much for this suggestion. Indeed the manuscript is very long as variable roots changed a variety of model results for the better. We agree that a clearer story helps to convey the most important messages. As we will now highlighted in the introduction in lines 101-103 we now focus on "the hypothesis that varying tree rooting depth is key to explain major patterns of evapotranspiration, productivity and the geographical distribution of tropical evergreen forests in South America" and structure the manuscript accordingly. We will shorten the manuscript according to this question and transfer several figures and tables to the supplement.
Finally, more work is needed to make the different results accessible and easy to interpret.
I found that each figure required a fair bit of work to interpret what is going on. Some simple changes could make it much easier for the reader, then we could spend less time deciphering your results and more time thinking about the science!

Thank you for raising this very important point. We will check every figure, insert more information like labels into the figures and simplify the captions and labels.

As examples,

Thank you very much for pointing out all these examples.

- In Fig 1: confusing caption. Simplify labels in legend.
We now simplified the caption and reduced the complexity of the legend labels.

- In Fig 6: label panels with dataset name, so that we don't need to refer to legend as much
We now label each panel as suggested in this figure which is now Fig.5.

- In Fig 9, uses different colours in the map and traces, otherwise these are easily confused. Label each subplot with site name.
We colored the markers more intensively and inserted the site name in each panel as suggested for this figure which is now Fig. 7. We applied the same approach in Fig. 8.

- In Fig 12, put labels on the columns (evergreen, deciduous) and rows (models), so that we can easily see what the different panels are without constantly referring to the caption.
We labelled all panels according to the suggestions for this figure which is now Fig. 10.

Some minor issues
- Some line breaks between paragraphs would make the text much easier to read
Thank you for raising this point. During shortening of the manuscript and strengthening the story line we will insert more line breaks.

- Eq 8: I've looked over this a few times and wonder if the n_est_tree at the end should be removed?
Thank you for this question. Yes this is true. The 2 times occurring n_est_tree can be cancelled from this equation. We kept it in for an easy comparison to the original equation (Eq. 7). We think this makes it easier for model developers to copy our approach.

- ln 413 – It didn't make much sense to me to compare your results to a modelled product of rooting depth.
Thank you for raising this point. We completely understand this criticism. Since empirical data on rooting depth in this region is very sparse, we wanted at least evaluate our results in the context of a totally different modelling approach. With regard to shortening the manuscript we now moved the comparison to the Appendix and avoid a detailed comparison of our results with Fan et al. (2017).

- I found the talk of offspring, saplings, and "growth" throughout the paper a bit misleading. My understanding of this version of LPJ is that each patch has a single functional type which has a density 'n' of average sized individuals. When new offspring are recruited, they don't grow from seed to adult, but rather enter fully formed at the average size (this occurs by increasing n, the number of individuals). The only time the individuals seemingly grow from small to large plants, is when starting from bare soil, i.e. during spin up. Yet, often the paper gave the impression that individuals could be born and grow.

Thank you very much for this criticism. We fully agree. Apparently, it is always a thin line between easy wording and correctness of model description. We now clarify that PFTs in LPJmL are average individuals in the methods now in line 237-239 and avoid any misleading formulations in the entire manuscript.

---

## Author Response (AR2)

Associate Editor Decision: Reconsider after major revisions (14 Oct 2020) by Martin De Kauwe

Dear Martin de Kauwe, dear referees,

We thank the reviewers for their time and effort to review our revised manuscript. We are very pleased that reviewer #1 was satisfied with our revision and recommended to accept our revised manuscript. We have inserted all of his minor points and suggestions in the 2nd revision (now manuscript version #5 in the BG file system; "bg-2020-97-manuscript-version5.pdf"). Regarding the continued criticism of reviewer #2 to our revised manuscript, we show in our response below that we already had in large part addressed his concern with our first revision of the manuscript and highlight this once more in our point-by-point response below. Furthermore, we now made additional changes to emphasize our storyline. We apologize that we did not upload a manuscript version showing all changes in track-changes mode with the first revision, which may have partly contributed to the difficulties for reviewer #2 not being able to see the in-depth revision that we had already undertaken. We have pasted the track changes version at the end of our response letter, which marks all the changes we made in the entire revision process.

In this author response, the editor comments are marked in *italic,* our response to the editor and reviewers is inserted in blue in the point-by-point response below.

Summary:

*Both of the original reviewers have read your revised manuscript and arrive at contrasting positions. R1 is satisfied with your revisions but highlights some minor points to address. R2 is less convinced by the scope of the revision and raises some very important issues. I think it is important to address all of their points. For me they make two key points:*

*1. A better explanation of the motivation - why do we need variable rooting depth? This feels fairly straightforward but is an important element of the narrative for any reader.*

Thank you for raising this point. In our view, the manuscript clearly introduces a hypothesis which we want to test. Testing this hypothesis is the motivation for implementing variable roots. Proving this hypothesis true is the reason why we need variable rooting depth. We have made additional changes to the current manuscript version #5 to clarify our logic. Here are our key points which can also be found in our detailed response to reviewer #2 below:

We would like to take the opportunity to explain our reasoning and refer also to our first author response letter. We show where in the revised manuscript we have made the requested changes:
Reviewer #2 claims that our "main motivation is to introduce variable rooting strategies" per se without wanting to "explain some phenomenon". We find that right in the beginning of the introduction the reader is introduced to a hypothesis and respective phenomenon, namely that "tree rooting depth is regarded as a crucial variable to explain the geographical distribution of main phenology strategies such as evergreen and deciduous, as well as the observed local to continental pattern of productivity, biomass storage, evapotranspiration (ET)" (line 39-42 last manuscript version #4). This is the hypothesis which aims at explaining the phenomenon why evergreen tropical forests have the observed geographical extent and why rates of ET and productivity are often high during the dry season. To further underline that this is the hypothesis we want to test, we now write "In this study we revisit the hypothesis that tree rooting depth is a crucial variable to explain the geographical distribution of main phenology strategies such as evergreen and deciduous, as well as the observed local to continental pattern of productivity, biomass storage, evapotranspiration (ET) and consequently moisture recycling" (line 39-42 current manuscript version #5).

Next, we introduce the reader to the fact that most models have a prescribed constant shallow rooting depth which is regarded as a reason why they have problems "reproducing the extent of South-America's tropical evergreen forests, as well as its seasonal productivity and ET especially in regions with seasonal rainfall" (line 50-52 last manuscript version #4, line 51-53 current manuscript version #5). This addresses "the inability of previous models to explain the wide distribution of evergreen types" and represents a strong argument to support the hypothesis. To underline the latter point we now write "In favor of the rooting depth hypothesis, most DGVMs in the past had problems reproducing the extent of South-America's tropical evergreen forests, as well as its seasonal productivity and ET especially in regions with seasonal rainfall" (line 51-53 manuscript version #5).

In the following paragraph, we introduce several examples of modelling approaches trying to solve this problem and mention their positive results (line 52-63 manuscript version #4; line 53-64 manuscript version #5), i.e. another strong argument in favor of the hypothesis. Next we introduce the conceptual shortcomings of all those previous modelling approaches which could undermine the results of the respective studies (line 64-92 last manuscript version #4; line 65-93 current manuscript version #5). This is a strong argument against the hypothesis and we therefore now write "… some assumptions of the underlying models might decrease the liability of their results and therefore pose arguments against the rooting depth hypothesis" (line 65-67 current manuscript version #5). We then continue to explain how our modelling approach overcomes these shortcomings (line 93-100 last manuscript version #4; line 94-101 current manuscript version #5). We finish the introduction with a clear statement that our study wants to "re-evaluate the hypothesis that varying tree rooting depth is key to explain major patterns of evapotranspiration, productivity and the geographical distribution of tropical evergreen forests in South America" (line 101-104 last manuscript version #4; line 102-105 current manuscript version #5). We give a clear summarizing answer regarding this re-evaluation of the hypothesis in the conclusion saying: "We here show for the first time that mean tree rooting depth across South-America can indeed explain the spatial distribution of tropical evergreen forests and their spatio-temporal pattern of ecosystem fluxes (ET and NEE) even when the competition of tree rooting strategies, carbon investment into gradually growing roots, and a spatially explicit soil depth are considered." (line 617-619 last manuscript version #4; line 583-585 current manuscript version #5). We furthermore now explicitly mention the hypothesis in the abstract (line 29-31 current manuscript version #5).

In summary we make it very clear that our "main motivation" is neither to introduce variable roots per se, nor to increase model quality per se (as suggested by reviewer #2), but to test this hypothesis. This is also in accordance with the requests from the first reviews to introduce a more biology oriented objective or research question that we want to answer with this study.

With this explanation and evidence from the revised manuscript we hope to have now made our reasoning clear which we had explained in our author response to reviewer #2 during the first round of revision. We hope that the introduction now clearly explains our justification of including variable roots to the reader and also explains the biology of the system.

*2. I felt the reviewers point about not understanding why variable root depth leads to predicting a wider distribution of evergreens was very important. It is important this is clearly explained in the revision.*

Our revised manuscript version 2 had already addressed this point in great detail, by addressing comments of reviewer #1 who had raised this issue in his/her first review. We had produced a detailed answer also in our first author response to reviewer #1. We explain again this point in our point-by-point explanation below. We have double-checked in our revised manuscript version #5, that this point is clearly explained and understandable to a wide audience. We have also answered new points raised by reviewer #2 of his second review in our response below and cross-checked this is clearly described and explained in the revised manuscript version #5. This also includes the explanation of the model differences in our method section. We hope to have now clarified the last remaining open questions in our revised manuscript version #5. Here are our key points which can also be found in our detailed response to reviewer #2 below:

*Our answer on main issue of not understanding why deeper roots lead to better reproducing measured rates of evapotranspiration and distributions of PFTs:*

We agree that this is an important point in improving the manuscript and so we take the opportunity to explain why variable rooting depth has an effect on the spatial distribution of evergreen trees.

An evergreen plant needs a minimum water supply all year round. When models use the standard maximum rooting depth of around 1 m they fail to reproduce evapotranspiration rates and the distribution of tropical evergreen rainforest. When we allow for variable rooting depth (even with investment trade-offs and root growth) results of evapotranspiration and evergreen distribution come close to observations. By weighting rooting depth with NPP contribution (Fig. 5) and plotting this rooting depth vs. an index of dry season strength (MCWD) and annual precipitation (Fig. 6), we show how the simulated vegetation adjusts to those climatological conditions. As deep roots are costly, any adjustment towards deeper roots must mean that deeper rooted PFTs are more productive, hence more competitive. And this adjustment makes only sense when the general growing strategy is "evergreen". So, the ultimate answer for "why are the results improving?" is, because the evergreen PFT has access to deep water in the dry season and gains a competitive advantage. We refer to lines 480-490, 503-516, 537-548 and 555-564 in the discussion section in the current manuscript version #5, where we discuss all these implications. Responding to the first review of reviewer #2, we put much emphasis on this explanation in our first manuscript revision. We have inserted background information on how water percolation is simulated in the LPJmL model (line 137-152 current manuscript version #5) as was requested by reviewer #2. Related to this, we also inserted a new Appendix figure (now Fig. A9 referred to in line 564 current manuscript version #5) both in the manuscript but also directly in the last response letter which irrefutable shows how the deep roots in LPJmL4.0-VR extract water from deep soil layers during the dry season in comparison to LPJmL4.0-VR-base. This level of evidence and explanation makes, in our view, a convincing set of arguments. We also refer to our response to the point below on a related topic. We would like to emphasize that we wrote a comprehensive paragraph specifically focusing on this topic in the last response letter to reviewer #2.

*Our answers on questions regarding differences between model versions related to doubting that rooting depth is the reason for reproducing measured rates of evapotranspiration and distributions of PFTs:*

We thank the reviewer for raising this point, which reviewer #1 also raised in his/her first review. Therefore, we had already explained this point in the revised manuscript version 2 in our previous resubmission. We would like to refer to our author response letter to reviewer #1 (p. 9 in bg-2020-97-AC1_version2.pdf), where he/she had commented on Fig. 12 which is now Fig. 10. We provide a high detail explanation, in response to the first review of reviewer#1, in the current manuscript version #5 in lines 503-516. It resolves all these new questions of reviewer #2. In summary our explanation in the revised manuscript comprises the following points:

LPJmL4.0 has 1) a constant shallow root distribution for both tropical tree PFTs and 2) no dominance dependent establishment of PFTs. Here the evergreen PFT is slightly more abundant in the wetter tropics and the deciduous slightly more abundant in the drier tropics (now Fig. 10g-h). Since there is no real mechanism rewarding one tree strategy over the other over time in the LPJmL4.0 model (including the spin-up simulation phase), both PFTs can be found about 50/50 in the Amazon region. LPJmL4-VR-base also has 1) constant shallow root distribution for both tropical tree PFTs, but 2) a dominance-dependent establishment of PFTs (LPJmL4.0-VR-base is equal to LPJmL4.0-VR except that it allows for shallow roots only as explained in line 287-292 current manuscript version #5). These two criteria reveal under which climate conditions, each PFT outcompetes the other over time, if trees could grow shallow roots only. Therefore, there is a clear dominance pattern of either PFT using LPJmL4.0-VR-base (even though this pattern is far away from matching evaluation data). Now in LPJmL4-VR, the addition of variable roots (which is the only difference to LPJmL4.0-VR-base), broadens the geographical extent where the evergreen PFT is clearly dominant. In this sense, LPJmL-VR-base does not go a "considerable way to explaining the wide distribution of evergreens", but only reveals the true dominance pattern of PFTs when performance is rewarded. Acceptable results are only achieved when adding variable roots, i.e. with LPJmL4.0-VR.
The dominance-dependent tree establishment and its consequences are mentioned at different occasions in the manuscript, e.g. in the discussion in line 503-513 (current manuscript version #5). We think we made sure that this point is now clearly explained in the revised manuscript.

*I am recommending further major revisions to allow you time to fully address the concerns of R2.*

Thank you for reconsidering our manuscript. Please find our detailed answers regarding the comments of reviewer #1 & #2 below.

**Referee #1: Anonymous**

The authors revised the manuscript substantially and it is now much clearer and more concise. All my comments and suggestions were adequately considered and included in the revised manuscript. I don't have additional major comments, only a few minor points and suggestions.

Thank you for your positive evaluation and important remarks. We have addressed all your points.

l. 35: often have access
Changed accordingly in line 35.
l. 66: maybe mention timing of what
We changed the word "timing" to "temporal growth" to be more specific now in line 67.
l. 131: but extend this scheme
Changed accordingly in line 131.
l. 133: computational demand. Further, I assume that this demand increase because deeper soil implies more soil layers that need to be updated. (in principle it would also be possible to keep the number of soil layers but change their thickness.
Changed accordingly now in line 134. You are right. In principle any soil layer thickness can be applied. Choosing thicker soil layers would decrease the computational demand, but would decrease the vertical resolution of root shapes as well as the temporal resolution of water percolation and temporal resolution of root growth. We chose the general soil layer partitioning of LPJmL4.0 of 1 m, in order to keep differences between model versions as small as possible. We now indicate this reason in line 132.
l. 194: when considering coarse roots
Changed accordingly now in line 195-196.
l. 371: explain the geographical pattern
Changed accordingly now in line 372.
l. 428: and more seasonal climates
Changed accordingly now in line 429.
l. 441: negative correlation
Changed accordingly now in line 442.
l. 456: other two models
Changed accordingly now in line 457.
l. 478: delete header, there is no 3.4.2
Thank you for pointing out this mistake. The whole section regarding simulated biomass has been moved to the Appendix.
l. 529: climatic envelope
Changed accordingly now in line 514.
l. 530: which coincides with

Changed accordingly now in line 515.

**Referee #2: Daniel Falster, daniel.falster@unsw.edu.au**

I reviewed a previous version of this paper, which introduces a scheme for variable rooting depths into the LPJmL4.0 DGVM. While generally enthusiastic about the idea, I found the previous version too long and unfocussed, which prevented any deep engagement and understanding. Both reviewers felt similarly and requested a more focussed narrative.

While the revised paper is a little clearer and has benefited from some minor improvements, my major criticisms have only been superficially addressed. The authors have clarified their goals only a little by adding a few lines of text in the introduction. Otherwise the introduction and methods seems almost identical, and we still have 12 figures, most with multiple panels.

We have pasted the track changes version of the manuscript version #5 at the end of our response letter, which marks all the changes we made in the entire revision process. We apologize for this confusion so that the reviewer could not retrace the modifications. We have compiled this now together with the minor changes from reviewer #1 which illustrates that the revision was indeed in-depth.
In the first revision, we reduced the number of figures from 15 to 12. Because we aim to enhance readability also for non-modelers, we want to emphasize that Fig. 1-3 are methodological figures which we regard as being essential for readers who are less familiar with modelling techniques to follow our logic. For the current manuscript version #5 we now additionally transferred Fig. 11-12 and the respective results and discussion section to the Appendix, which reduces the number of figures to 10 and shortens the manuscript. The remaining figures we regard as crucial results and very important for the storyline. Those figures show simulated rooting depths, evapotranspiration rates and forest type cover, which are essential variables to support the findings of our study.

The authors really need to do more to streamline the paper and tell a convincing story about the biology of the system.
The way it currently reads, the main motivation is to introduce variable rooting strategies. But this is the solution. The motivation should be to explain some phenomenon: I'd suggest focussing on the inability of previous models to explain the wide distribution of evergreen types in the Amazon. While the authors do mention this in paragraph two, this material seems to come second to the goal of modelling variable rooting depths, and then gets lost amongst the many plots and attention given to biomass and ET.

The significance of the problem relating to evergreens vs deciduous doesn't really hit until we reach Figure 10, which shows just how bad the standard LPJ model does in predicting the distribution of evergreen vs deciduous types. Perhaps these panels could be introduced in the introduction, to motivate the study? Then variable rooting depth introduced as a potential solution?

We would like to take the opportunity to explain our reasoning and refer also to our first author response letter. We show where in the revised manuscript we have made the requested changes:

Reviewer #2 claims that our "main motivation is to introduce variable rooting strategies" per se without wanting to "explain some phenomenon". We find that right in the beginning of the introduction the reader is introduced to the hypothesis and respective phenomenon, namely that "tree rooting depth is regarded as a crucial variable to explain the geographical distribution of main phenology strategies such as evergreen and deciduous, as well as the observed local to continental pattern of productivity, biomass storage, evapotranspiration (ET)" (line 39-42 last manuscript version #4). This is the hypothesis which aims at explaining the phenomenon why evergreen tropical forests have the observed geographical extent and why rates of ET and productivity are often high during the dry season. To further underline that this is the hypothesis we want to test, we now write "In this study we revisit the hypothesis that tree rooting depth is a crucial variable to explain the geographical distribution of main phenology strategies such as evergreen and deciduous, as well as the observed local to continental pattern of productivity, biomass storage, evapotranspiration (ET) and consequently moisture recycling" (line 39-42 current manuscript version #5).

Next, we introduce the reader to the fact that most models have a prescribed constant shallow rooting depth which is regarded as a reason why they have problems "reproducing the extent of South-America's tropical evergreen forests, as well as its seasonal productivity and ET especially in regions with seasonal rainfall" (line 50-52 last manuscript version #4, line 51-53 current manuscript version #5). This addresses "the inability of previous models to explain the wide distribution of evergreen types" and represents a strong argument to support the hypothesis. To underline the latter point we now write "In favor of the rooting depth hypothesis, most DGVMs in the past had problems reproducing the extent of South-America's tropical evergreen forests, as well as its seasonal productivity and ET especially in regions with seasonal rainfall" (line 51-53 current manuscript version #5).

In the following paragraph, we introduce several examples of modelling approaches trying to solve this problem and mention their positive results (line 52-63 last manuscript version #4; line 53-64 current manuscript version #5), i.e. another strong argument in favor of the hypothesis. Next we introduce the conceptual shortcomings of all those previous modelling approaches which could undermine the results of the respective studies (line 64-92 last manuscript version #4; line 65-93 current manuscript version #5). This is a strong argument against the hypothesis and we therefore now write "… some assumptions of the underlying models might decrease the liability of their results and therefore pose arguments against the rooting depth hypothesis" (line 65-67 current manuscript version #5). We then continue to explain how our modelling approach overcomes these shortcomings (line 93-100 last manuscript version #4; line 94-101 current manuscript version #5). We finish the introduction with a clear statement that our study wants to "re-evaluate the hypothesis that varying tree rooting depth is key to explain major patterns of evapotranspiration, productivity and the geographical distribution of tropical evergreen forests in South America" (line 101-104 last manuscript version #4; line 102-105 current manuscript version #5). We give a clear summarizing answer regarding this re-evaluation of the hypothesis in the conclusion saying: "We here show for the first time that mean tree rooting depth across South-America can indeed explain the spatial distribution of tropical evergreen forests and their spatio-temporal pattern of ecosystem fluxes (ET and NEE) even when the competition of tree rooting strategies, carbon investment into gradually growing roots, and a spatially explicit soil depth are considered." (line 617-619 last manuscript version #4; line 583-585 current manuscript version #5). We furthermore now explicitly mention the hypothesis in the abstract (line 29-31 current manuscript version #5).

In summary we make it very clear that our "main motivation" is neither to introduce variable roots per se, nor to increase model quality per se (as suggested by reviewer #2), but to test this hypothesis. This is also in accordance with the requests from the first reviews to introduce a more biology oriented objective or research question that we want to answer with this study. The storyline is oriented along this hypothesis and delivers an ecological derivation of our main conclusion, by a) first showing that in our modelling approach the rooting strategies align along the environmental gradients, which then leads to being able to reproduce b) observed rates of evapotranspiration productivity as well as c) observed distributions of plant functional types.

With this explanation and evidence from the revised manuscript we hope to have now made our reasoning clear which we had explained in our author response to reviewer #2 during the first round of revision. We hope that the introduction now clearly explains our justification of including variable roots to the reader and also explains the biology of the system.

After reading the paper several times, I still can't say why variable rooting depth leads the LPJmL4.0-VR version to predict wider distribution of evergreens. I see that it does, but don't understand why.

We agree that this is an important point in improving the manuscript and so we take the opportunity to explain why variable rooting depth has an effect on the spatial distribution of evergreen trees:

An evergreen plant needs a minimum water supply all year round. When models use the standard maximum rooting depth of around 1 m they fail to reproduce evapotranspiration rates and the distribution of tropical evergreen rainforest. When we allow for variable rooting depth (even with investment trade-offs and root growth) results of evapotranspiration and evergreen distribution come close to observations. By weighting rooting depth with NPP contribution (Fig. 5) and plotting this rooting depth vs. an index of dry season strength (MCWD) and annual precipitation (Fig. 6), we show how the simulated vegetation adjusts to those climatological conditions. As deep roots are costly, any adjustment towards deeper roots must mean that deeper rooted PFTs are more productive, hence more competitive. And this adjustment makes only sense when the general growing strategy is "evergreen". So, the ultimate answer for "why are the results improving?" is, because the evergreen PFT has access to deep water in the dry season and gains a competitive advantage. We refer to lines 480-490, 503-516, 537-548 and 555-564 in the discussion section in the current manuscript version #5, where we discuss all these implications. Responding to the first review of reviewer #2, we put much emphasis on this explanation in our first manuscript revision. We have inserted background information on how water percolation is simulated in the LPJmL model (line 137-152 current manuscript version #5) as was requested by reviewer #2. Related to this, we also inserted a new Appendix figure (now Fig. A9 referred to in line 564 current manuscript version #5) both in the manuscript but also directly in the last response letter which irrefutable shows how the deep roots in LPJmL4.0-VR extract water from deep soil layers during the dry season in comparison to LPJmL4.0-VR-base. This level of evidence and explanation makes, in our view, a convincing set of arguments. We also refer to our response to the point below on a related topic. We would like to emphasize that we wrote a comprehensive paragraph specifically focusing on this topic in the last response letter to reviewer #2.

That said, the results suggest it's not only about variable rooting depth. I was interested to see in Figure 10 that that the LPJmL4.0-VR-base model actually goes a considerable way to explaining the wide distribution of evergreens. So there seems to be two changes making a difference: from LPJmL4.0 -> LPJmL4.0-VR-base, then from LPJmL4.0-VR-base -> LPJmL4.0-VR. The authors need to explain the biological mechanism causing LPJmL4.0 and LPJmL4.0-VR-base to differ and perhaps temper their claims about variable rooting accordingly.

We thank the reviewer for raising this point, which reviewer #1 also raised in his/her first review. Therefore, we had already explained this point in the revised manuscript version 2 in our previous resubmission. We would like to refer to our author response letter to reviewer #1 (p. 9 in bg-2020-97-AC1_version2.pdf), where he/she had commented on Fig. 12 which is now Fig. 10. We provide a high detail explanation, in response to the first review of reviewer#1, in the current manuscript version #5 in lines 503-516. It resolves all these new questions of reviewer #2. In summary our explanation in the revised manuscript comprises the following points:

LPJmL4.0 has 1) a constant shallow root distribution for both tropical tree PFTs and 2) no dominance dependent establishment of PFTs. Here the evergreen PFT is slightly more abundant in the wetter tropics and the deciduous slightly more abundant in the drier tropics (now Fig. 10g-h). Since there is no real mechanism rewarding one tree strategy over the other over time in the LPJmL4.0 model (including the spin-up simulation phase), both PFTs can be found about 50/50 in the Amazon region. LPJmL4-VR-base also has 1) constant shallow root distribution for both tropical tree PFTs, but 2) a dominance-dependent establishment of PFTs (LPJmL4.0-VR-base is equal to LPJmL4.0-VR except that it allows for shallow roots only as explained in line 287-292 current manuscript version #5). These two criteria reveal under which climate conditions, each PFT outcompetes the other over time, if trees could grow shallow roots only. Therefore, there is a clear dominance pattern of either PFT using LPJmL4.0-VR-base (even though this pattern is far away from matching evaluation data). Now in LPJmL4-VR, the addition of variable roots (which is the only difference to LPJmL4.0-VR-base), broadens the geographical extent where the evergreen PFT is clearly dominant. In this sense, LPJmL-VR-base does not go a "considerable way to explaining the wide distribution of evergreens", but only reveals the true dominance pattern of PFTs when performance is rewarded. Acceptable results are only achieved when adding variable roots, i.e. with LPJmL4.0-VR.
The dominance-dependent tree establishment and its consequences are mentioned at different occasions in the manuscript, e.g. in the discussion in line 503-513 (current manuscript version #5). We think we made sure that this point is now clearly explained in the revised manuscript.

Unfortunately, the LPJmL4.0-VR-base model is never really introduced or explained in any detail, so we can only guess why it predicts a wider distribution of evergreens than LPJmL4.0. I also note strong differences between LPJmL4.0 and LPJmL4.0-VR-base in Tables A6 and A7. My naive expectation is that LPJmL4.0 and LPJmL4.0-VR-base should produce near identical results, as they mostly do in Tables A3-A5. Why is this the case, if there's no variance in rooting depth?

Here, we would like to stress that reviewer #2 is raising a new issue which we are happy to explain. Parts of these points are already addressed in our response above and had been answered in the first review. The other points are explained here:

Regarding the claim of reviewer #2 that LPJmL4.0-VR-base has never been introduced in the manuscript, we would like to refer to line 287-292 where LPJmL4-VR-base was perfectly introduced in the manuscript in section "2.4. Model versions and simulation protocol" since our first submission. Moreover, the features of LPJmL4.0-VR-base are again explained in the discussion of the revised manuscript version #5 now in line 560-564.

Reviewer #2 claims that one "can only guess why it [LPJmL4.0-VR-base] predicts a wider distribution of evergreens than LPJmL4.0" and that numbers in Tables A3-A7 do not add up with our logic. However, the numbers in Tables A3-A7 are perfectly in line with our argumentation and the coherences are explained throughout the manuscript. As mentioned in our points above there must be differences between LPJmL4.0 and LPJmL4.0-VR-base, because one model rewards PFT performance over time and the other one does not (as explained in detail in line 503-513 current manuscript version #5). Again (as explained in our points above), therefore LPJmL4.0 shows no clear dominance pattern of PFTs, whereas LPJmL4.0-VR-base does show clear dominance pattern. That is why in Table A6 (comparing regional PFT dominance to validation data) and Table A7 (comparing regional biomass to validation data) the differences between the models are large. The reason why Table A3-A5 show nearly identical results for LPJmL4.0 and LPJmL4.0-VR-base is, because here simulation results are compared to regional/local monthly rates of evapotranspiration and local monthly net ecosystem exchange evaluation data. These monthly variables are highly dependent on PFT water access and therefore rooting depth. Because LPJmL4.0 and LPJmL4.0-VR-base have the same constant rooting depth (unlike LPJmL4.0-VR with variable rooting depth), regardless of the underlying PFT type, similar simulation results are obtained. In this sense, reviewer #2 is correct that there is no variance in rooting depth, but this is anticipated by our study design to show the reader 1) the effect of dominance-dependent tree establishment and 2) the effect of introducing variable roots.

We hope that with the current revision and additional explanations we have finally addressed all concerns of reviewer #2 and have hopefully provided a coherent and understandable manuscript.

Yours sincerely,
Boris Sakschewski on behalf of all co-authors.

**Variable tree rooting strategies  are key to model distribution, productivity and evapotranspiration  of tropical evergreen forests**

Boris Sakschewski[1], Werner von Bloh[1], Markus Drüke[1], Anna A. Sörensson[2, 3], Romina Ruscica[2, 3], Fanny Langerwisch[4, 5], Maik Billing[1], Sarah Bereswill[6], Marina Hirota[7, 8], Rafael S. Oliveira[8], Jens Heinke[1], Kirsten Thonicke[1]

[1]Potsdam Institute for Climate Impact Research, Potsdam, 14473, Germany
[2]Universidad de Buenos Aires - Consejo Nacional de Investigaciones Científicas y Técnicas, Centro de Investigaciones del Mar y la Atmósfera (CIMA/UBA-CONICET), Buenos Aires, Argentina.
[3]Institut Franco-Argentin d'Etudes sur le Climat et ses Impacts, Unité Mixte Internationale (UMI-IFAECI/CNRS-CONICET-UBA), Argentina
[4]Czech University of Life Sciences Prague, Department of Water Resources and Environmental Modeling, 165 00 Praha 6 – Suchdol, Czech Republic
[5]Palacký University Olomouc, Department of Ecology and Environmental Sciences, 78371 Olomouc, Czech Republic
[6]University of Potsdam, Potsdam, 14469, Germany
[7]Federal University of Santa Catarina (UFSC), Campus Universitário Reitor João David Ferreira Lima Trindade – Florianópolis – SC, CEP: 88040-900, Santa Catarina, Brazil
[8]University of Campinas (UNICAMP) Cidade Universitária "Zeferino Vaz" CEP 13083-970, Campinas-SP, Sao Paulo, Brazil

*Correspondence to*: Boris Sakschewski (boris.sakschewski@pik-potsdam.de)

**Abstract.** ~~Tree water access via roots is crucial for forest functioning and therefore forests have developed a vast variety of rooting strategies across the globe. However, Dynamic Global Vegetation Models (DGVMs), which are increasingly used to simulate forest functioning, often condense this variety of tree rooting strategies into biome scale averages, potentially under- or overestimating forest response to intra- and inter annual variability in precipitation. Here we present a new~~ A variety of modelling studies have suggested tree rooting depth as a key variable to explain evapotranspiration rates, productivity and the geographical distribution of evergreen forests in tropical South America. However, none of those studies acknowledged resource investment, timing and physical constraints of tree rooting depth within a competitive environment, undermining the ecological realism of their results. Here we present an approach of implementing variable rooting strategies and dynamic root growth into the LPJmL4.0 DGVM and apply it to tropical and sub-tropical South-America under contemporary climate conditions. We show how competing rooting strategies which underlie the trade-off between above- and below-ground carbon investment lead to more  realistically simulated intra-annual productivity and evapotranspiration, and consequently forest cover and spatial biomass distribution. We find that climate and soil depth determine a spatially heterogeneous pattern of mean rooting depth and belowground biomass across the study region. Our findings support the hypothesis that the ability of evergreen trees to adjust their rooting systems to seasonally dry climates is crucial to explain the current dominance, productivity and evapotranspiration of evergreen forests in tropical South America.

**1 Introduction**

Tropical evergreen forest is the naturally dominant biome type in South-America over a large climatic range including regions with a marked dry season (Hirota et al., 2011; Xiao et al., 2006). To withstand seasonal shortages of precipitation and sustain productivity, trees with evergreen phenology often  have access to deep soil water via deep roots (Brum et al., 2019; Canadell et al., 1996; Johnson et al., 2018; Kim et al., 2012; Markewitz et al., 2010). Consequently, recent studies suggest a heterogeneous spatial pattern of maximum rooting depth across tropical forest biomes in South-America which differs over the order of magnitudes depending on local groundwater, soil and climate conditions (Canadell et al., 1996; Fan et al., 2017).   tree rooting depth is  a crucial variable to explain the geographical distribution of main phenology strategies such as "evergreen" and "deciduous", as well as the observed local to continental pattern of productivity, biomass storage, evapotranspiration (ET) and consequently moisture recycling (Fan et al., 2017; Jobbágy and Jackson, 2000; Kleidon and Heimann, 2000; Langan et al., 2017; Nepstad et al., 1994; Stahl et al., 2013). To test this hypothesis, dynamic global vegetation models (DGVMs) seem to be promising tools, as those models are suitable to project 
[revised manuscript text omitted]

---

## Author Response (AR3)

**Associate Editor Decision: Publish subject to minor revisions (review by editor)** (22 Feb 2021) by
Martin De Kauwe

Comments to the Author:

Dear Authors,

Two new reviewers have looked through your major revision, both have concluded that it is an improvement and makes a nice contribution to the literature. Nevertheless, both have highlighted shortcomings with respect to the presentation of figures & associated descriptions. I think they have both made some positive and constructive suggestions for how you might streamline the manuscript before publication. I am recommending minor revisions, which I will review. Can I ask you to seriously consider their various suggestions.

Best wishes,

Martin De Kauwe

Dear Editor Martin De Kauwe and referees, thank you very much for your detailed reviews and editorial comments. We have thoroughly addressed all points raised by the 2 new referees to further streamline the manuscript and put more emphasis on figures and associated descriptions.

Please find below our new specific responses to the comments of referee 3 & 4. Original referee comments are marked as black text. Our answers to each comment are marked as blue text. Please also find the track changes version of the manuscript at the end of this response letter. We sincerely hope the new vast adjustments of the manuscript fulfill the requirements of the new referees.

With best regards and on behalf of all co-authors,

Boris Sakschewski

**Referee #3: Anonymous**

This study by Sakschewski and colleagues presents an updated version of the DGVM LPJmL that is able to reproduce the observed distribution of tropical vegetation based on representation of competing rooting strategies between evergreen and deciduous trees. It is exciting to see that the sub-model version LPJmL4.0-VR seems to capture spatio-temporal patterns of ecosystem fluxes across South-America based on spatially explicit information on soil depth and thus considering variable rooting depth and differences in relative carbon investment into belowground biomass between coexisting plant species in competition for limiting resources. Nevertheless, in line with the comments provided by foregoing referees some weakness in delivering the findings presented in this study could still be resolved in order to more concisely present respective findings to the reader. For instance, one option would be to use the structure of the sub-sections presented in the discussion section, i.e. 4.1, 4.2, 4.3 to formulate respective hypothesis at the end of the introduction section.

We thank the reviewer for this positive evaluation. We hope that we have resolved weaknesses in delivering our findings to the reader by following your suggestions below and applying the logic to the whole manuscript. Therefore, we strongly reduced the size of the manuscript by placing much of the methods and results into the Appendix and adjusted the whole manuscript accordingly. Please also find the track-changes version of manuscript at the end of this author response letter.

For more specific examples of how to improve the presentation of the manuscript please see the following points and some minor suggestions below:

1. As has been criticised by referee #2 some aspects of the manuscript are presented quite lengthy and the language used to explain some of the main findings could still be improved, e.g. to further clarify some of the contrasting findings presented in L59-61 and L61-64.

Thank you for emphasizing the need to streamline some aspects of our manuscript further. It helped to improve our manuscript. The studies mentioned in the introduction formerly mentioned in line 59-61 and 61-64 (now line 42-53) essentially all agree that rooting depth must be an important explanatory variable for rates of evapotranspiration, productivity and vegetation distribution in the study region. We reckon that each study reflected more or less on different aspects of variable rooting systems and different additional factors in tropical biomes, but we do not see a contradiction between their findings or to our results. The general message is that models should consider different tree rooting strategies. We updated the manuscript to put more emphasis on this aspect and the explanation to underline our point now in the introduction, now in line 41-42. We had discussed this aspect in earlier versions of the manuscript, where we put those studies into context formerly in line 485-490 and 528-529. We kept them, now they can be found in line 343-346 and 386-387.

As suggested, we now strongly shortened the whole manuscript and tried our best to improve the language and explanations. With the improvements in the introduction we hope to now raise the expectation of what the manuscript will deliver in the introduction and discuss our contribution to this scientific topic in the discussion section. We hope this now is more consistent. You will find the full extent of the changes in the track-changes file at the end of this response letter.

2. In line with this criticism, the manuscript could be streamlined further by reducing the number of display items that are critical to understand the main findings without presenting the underlying method, which could be moved into the appendix, e.g. by reducing the figures to only Fig. 2/5/10 and moving the rest in the appendix.

Thank you for your suggestions on how to shorten the manuscript and number of figures to streamline it better. We reduced the display items following your suggestion and adjusted the manuscript text accordingly. We additionally keep Fig. 9 (now Fig. 3 on continental scale evapotranspiration) as our storyline is built around evapotranspiration as an important indicator of forest productivity and atmospheric moisture supply and serves to validate our results on a continental scale. In our opinion, focusing on vegetation distribution only, misses an important point of our study, especially with regard to your comment 3 which includes evapotranspiration into the set of study hypotheses. Reducing the display items from 10 to 4 implied a large adjustment of the whole manuscript. Moreover, as you suggested, we vastly shortened the methods where we explain our modelling approach and now provide a short general overview in line 114-147. The full methods including explanatory figures have been moved to Appendix A. Because the changes are substantial, we can only refer to the track changes version of our manuscript at the end of this response letter.

3. While I would disagree with the referee's comment that major criticisms have only superficially addressed in the revision, I wonder if the presentation of the manuscript could be made even more explicit by using respective sub-section headers (i.e. 4.1, 4.2, 4.3) in the discussion section for formulating specific hypothesis at the end of the introduction section?

Thank you for pointing out how we could transform our hypothesis at the end of the introduction. We now split our initial hypothesis, originally in line 102-105, into 3 sub-hypotheses according to the discussion headlines 4.1, 4.2 and 4.3. We now write in line 92-96:

"Given these new model developments we here re-evaluate the hypotheses that I) climate and soil depth determine dominant tree rooting strategies, II) tree rooting depth influences the distribution and dominance and III) diverse tree rooting strategies are key to explain rates of evapotranspiration and productivity of tropical evergreen forests in South America." We also refer to these new hypotheses in the conclusions.

We hope those changes satisfy your request.

L64: consider rephrasing to "and biomass in fire prone ecosystems".

Thank you for this suggestion. We changed it according to your suggestion, now in line 50-53.

L65: please clarify which effects (i.e. on what) you are talking about?

Thank you for pointing this out. We have now added "on ET and forest productivity" in line 53-55.

L594: consider rephrasing "this potential treasure".

We deleted the words now in line 456.

**Referee #4: Anonymous**

This paper is a major revision of a previous discussion paper, although this is the first time I review it. The study describes the implementation of variable rooting depth into the LPJmL dynamic vegetation model and evaluates the model against a variety of datasets for the Amazon basin. I find that the paper is very well written, the rationale, methods and results are all described very clearly. It is true that the paper has quite a lot of figures, but the large number of validation datasets are needed to prove that the model actually works, and more importantly, that it makes a difference to our ability to predict vegetation distribution and function across the Amazon (which it does). The authors show that variable rooting depth improves both the model's ability to predict carbon and water flux seasonality and the distribution of PFTs across the basin. Generally, variable rooting depth is a well known gap in vegetation models and the current study goes a long way to address that gap, making it very valuable for the modelling community.

Thank you for this positive evaluation of our manuscript. We appreciate that you value the complexity of our analysis and required text and figures to explain the importance of variable rooting depth for PFT distribution and ecosystem function in a biodiverse and complex biome such as the Amazon rainforest. Nevertheless, in order to satisfy the remarks of referee #3 we reduced the number of figures and detail and transfer a lot of information to the Appendix. We hope by following those demands you are still happy with the new manuscript version.

Below, just a handful of very minor comments:
L 207 What is k the growth rate of? And does it have any units? Is this expected to be fixed in time and space?

Thank you for pointing this out. k is a constant defining the growth rate of the standard logistic growth function and it has no unit. We now write in Appendix A in line 866-868 "…, $k$ is a dimensionless constant which defines the growth rate of the standard logistic growth function (set to 0.02), …". We also discovered a typo in equation 5. We now corrected this in equation A5 in line 865 to:

$$D = \frac{S}{1 + e^{-kSh} \cdot \left(\frac{S}{D_0} - 1\right)}$$

L 392 I'm not sure I understand the brackets in this sentence

We are sorry for that confusion. We tried to safe space by referring to the logical counterparts in brackets. In the process of shortening the manuscript the sentence was deleted.

L 406 not sure apparently is the right word here

Thank you for pointing this out. We replaced it with "Therefore" now in line 281.

Fig. 7 Does 'reference' here refer to observations?

Thank you for this question. The word reference refers to the respective evaluation data set which is composed of observation and remotely sensed data which we used in our study. We now explain the label "Reference" in the description of Fig. 3, B6, and B7 to avoid misunderstandings.

L 567 It is unclear here why it is assumed that without the limiting factor of soil depth trees would keep on growing roots much deeper

Thank you for raising this issue. We are sorry for having caused confusion here. We argue that simulated mean rooting depth would increase in areas where soil depth is limiting it, if at the same time climate would make it beneficial to grow deeper roots. Taking away local soil depth limits and instead applying a relatively large universal soil depth of e.g. 20 m, would thus potentially increase rooting depth and therefore rates of ET.

We explain in more detail what we mean regarding this topic, now in line 424-426 by stating: "Without limits to rooting depth in the form of local soil depth (e.g. by applying a universal soil depth of e.g. 20 m) and below-ground carbon investment, seasonally-dry climatological clusters would potentially shift towards deeper rooted sub-PFT dominance, consequently leading to an overestimation of ET rates."

[revised manuscript text omitted]